# A prolonged stress rat model recapitulates some PTSD-like changes in sleep and neuronal connectivity

Yun Lo[1], Pei-Lu Yi [2✉], Yi-Tse Hsiao [1], Tung-Yen Lee[3] & Fang-Chia Chang [1,3,4,5,6✉]

Chronic post-traumatic stress disorder (PTSD) exhibits psychological abnormalities during fear memory processing in rodent models. To simulate long-term impaired fear extinction in PTSD patients, we constructed a seven-day model with multiple prolonged stress (MPS) by modifying manipulation repetitions, intensity, and unpredictability of stressors. Behavioral and neural changes following MPS conveyed longitudinal PTSD-like effects in rats for 6 weeks. Extended fear memory was estimated through fear retrieval induced-freezing behavior and increased long-term serum corticosterone concentrations after MPS manipulation. Additionally, memory retrieval and behavioral anxiety tasks continued enhancing theta oscillation activity in the prefrontal cortex-basal lateral amygdala-ventral hippocampus pathway for an extended period. Moreover, MPS and remote fear retrieval stimuli disrupted sleep-wake activities to consolidate fear memory. Our prolonged fear memory, neuronal connectivity, anxiety, and sleep alteration results demonstrated integrated chronic PTSD symptoms in an MPS-induced rodent model.

[1] Department of Veterinary Medicine, School of Veterinary Medicine, National Taiwan University, Taipei 10617, Taiwan. [2] Department of Sport Management, College of Tourism, Leisure and Sports, Aletheia University, New Taipei City 25103, Taiwan. [3] Graduate Institute of Brain & Mind Sciences, College of Medicine, National Taiwan University, Taipei 110225, Taiwan. [4] Neurobiology & Cognitive Science Center, National Taiwan University, Taipei 10617, Taiwan. [5] Graduate Institute of Acupuncture Science, College of Chinese Medicine, China Medical University, Taichung 40402, Taiwan. [6] Department of Medicine, College of Medicine, China Medical University, Taichung 40402, Taiwan. ✉email: pyi67@hotmail.com; fchang@ntu.edu.tw

Post-traumatic stress disorder (PTSD) is a chronic psychological syndrome that develops after direct or indirect exposure to a traumatic event. Aversion stimuli prompt long-lasting symptoms, including fear-intrusive memory, trauma-related event avoidance, negative emotion, hypervigilance, and anxiety[1], severely impairing an individual's function. Many studies have established animal models by manipulating various traumatic stressors to induce PTSD-like symptoms[2–5]. For instance, restraint, electrical foot shock, and single prolonged stress (SPS) are physical single-manipulates stressors, whereas repeated multiple concurrent stressors (RMS) stimulation utilizes repeated stimulations to initiate chronic psychological and behavioral disabilities[5–8]. Studies have proven that restraint and inescapable footshock stress (IFS) can induce delayed avoidance, anxiety, and hyperarousal in rodents[8–16]. RMS is a ten-day constant chronic paradigm, combining physical, emotional, and social stressors, kindling neuronal loss and cognitive impairments in rodents, to uncover life stressor effects[5,17]. The comprehensive stressor SPS combines three different manipulations, including 2-hour restraint, 20 min of forced swimming, and ether anesthesia to strengthen stimulation intensity and provoke PTSD-like symptoms[18–20]. In addition, some studies used an electrical footshock as a Pavlovian fear conditioning inducer to assess sensitization toward new attacking stress after SPS[7,21,22]. Their results confirmed that SPS before the electrical shock decreased the stress threshold and failed to instill fear extinction retention[23]. Moreover, SPS can effectively initiate several PTSD-like criteria in molecules, neuronal transmission, and psychological components[24–26]. One main PTSD feature is its time-dependent negative feedback enhancement in the hypothalamic-pituitary-adrenal (HPA) axis[26,27]. This chronic HPA axis suppression impairs hippocampal activity and consequent memory extinction[23]. In contrast, fear memory processing after a traumatic event relies on hippocampus-amygdala-prefrontal cortex (HPC-AMY-PFC) circuit neuronal activity[28–31]. HPC's and PFC's traumatic stress-induced apoptosis and synaptic plasticity reduction damage neuronal structure and function, reducing top-down amygdala connections controlled by HPC-PFC circuits[30,32–35] and resulting in behavior deficits[25,36–40]. Furthermore, hyperarousal and sleep disruption in PTSD-like rodent models increased wakefulness and rapid-eye-movement (REM) sleep percentages following SPS initiation[41–43]. These effects demonstrate that exposed to multiple stressors before a traumatic event may be more susceptible to PTSD[20,44]. However, the modified SPS procedure divergence in some research generated variant behavioral and neuronal consequences during different periods, indicating that SPS' PTSD-like effects were time- and experience-dependent[20]. Further research is necessary to determine long-lasting PTSD-like consequences since clinical patients experience significant fear extinction impairment[45–48]. We established multiple stressors-induced long-term PTSD in the SPS process by increasing stressor repetitions and intensities. This method simulated rodent exposure to chronic, high-intensity stressors to stabilize PTSD-like abnormalities after stimulation.

The manipulation procedure in our modified model was prolonged from one to 7 days with random stressor sequences to increase stress unpredictability. We also reinforced stressor intensity by integrating four times the electric footshock stimuli as the primary stressor[49]. Since the single stress acquisition in the SPS was converted to multiple time stimuli, we named this modified process Multiple Prolonged Stress (MPS). Through this MPS model, we evaluated long-term PTSD criteria, including freezing behavior during conditioned fear retrieval, HPA-axis alteration, neuronal circuitry activity, anxiety, and sleep disruption. Furthermore, we demonstrated that MPS administration persisted in physiological and psychological alterations for 6 weeks, revealing multiple

stress stimulation efficiencies in manifesting long-lasting PTSD-like syndromes.

## Results

**MPS-induced prolonged fear expressions after memory retrieval.** We first examined freezing behavior for seven consecutive weeks to assess fear memory preservation post-MPS. Then, we compared MPS and SPS remote memory retrieval abilities with IFS manipulations (Fig. 1a, b). Next, we determined freezing duration (immobility percentage) during IFS stimuli to evaluate fear memory acquisition. Day 1 SPS pre-tone (before first tone cue) immobility percentages were <10% (0.97%), establishing that rats did not exhibit fear generalization toward the IFS context (Fig. 1c). IFS stimuli across MPS (Days 1, 3, 5, and 7) and SPS (Day 1) processes significantly increased immobility percentages (***$p < 0.001$: control vs. MPS, ###$p < 0.001$: control vs. SPS; Fig. 1c), and the MPS group's immobility increased alongside IFS frequency compared to the first IFS manipulation, indicating fear acquisition (&&$p < 0.01$, &&&$p < 0.001$: Day 1 vs. Days 3, 5, and 7; Fig. 1c).

Then, we detected immobility percentages during context and cue retrieval procedures, respectively. For context retrieval, immobility responses were significantly elevated during short- (2 weeks) and long-term (6 weeks) periods in the MPS group's recollection (***$p < 0.001$: control vs. MPS, ++$p < 0.01$, +++$p < 0.001$: MPS vs. SPS), though the long-term freezing effect declined when compared with the first context recall (&&&$p < 0.001$: Day 10 vs. Days 35 and 46; Fig. 1d). However, SPS only heightened immobility during the short-term period (##$p < 0.01$, ###$p < 0.001$: control vs. SPS) and reduced it during long-term recall (&&&$p < 0.001$: Day 10 vs. Days 35 and 46; Fig. 1d). In cue retrieval, both MPS and SPS groups displayed high immobility throughout the 6-week recall process (***$p < 0.001$, ###$p < 0.001$; Fig. 1e). Both context and cue retrieval data substantiate MPS's prolonged stress effects on fear memory recall. Yet, SPS only preserved cue-provoked fear retrieval during the remote period. These findings suggest that MPS has a stronger fear memory sustainability for contextually associated fear than SPS.

In addition, we confirmed MPS stress levels by measuring serum corticosterone concentrations, which remained at higher levels post-memory retrieval (combining context and cue retrieval) on Day 50 compared to the control (***$p < 0.001$; Fig. 1f). This data reveals that MPS-induced fear memory and abnormal physiology functions persist for at least 6 weeks.

**Fear retrieval post-MPS enhanced PFC, BLA, and vHPC neuronal activity.** We next evaluated whether fear memory retrieval (context and cue retrieval) post-MPS manipulation would excite neuronal activity in fear encoding regions. Theta oscillations modulate neuronal circuitry during fear memory processing[50]. Thus, we examined theta levels from the fear-memory encoding brain regions PFC, BLA, and vHPC over 50 days (Fig. 2a). In context retrieval, rats' local field potentiations (LFPs) were first recorded in their home cage for 10 min as the basal neuronal activity marker. They were then transferred to the footshock box for contextual fear memory retrieval (Fig. 2b). Average theta oscillation alterations from these regions were measured and designated for each experimental day (Fig. 2d, Supplementary Fig. 1a). We determined that theta power intensities in the MPS group's PFC, BLA, and vHPC surged from this shift (*$p < 0.05$, **$p < 0.01$, ***$p < 0.001$: home cage vs. context; Fig. 2d). The control group also expressed enhancement after exposed in context (*$p < 0.05$, ***$p < 0.001$: home cage vs. context; Fig. 2d, Supplementary Fig. 1a). Furthermore, the control group's theta powers in PFC presented a higher level than that of MPS group due to the basal oscillation intensity difference (#$p < 0.05$, ##$p < 0.01$; Fig. 2d). However, the MPS has greater power variation in PFC when

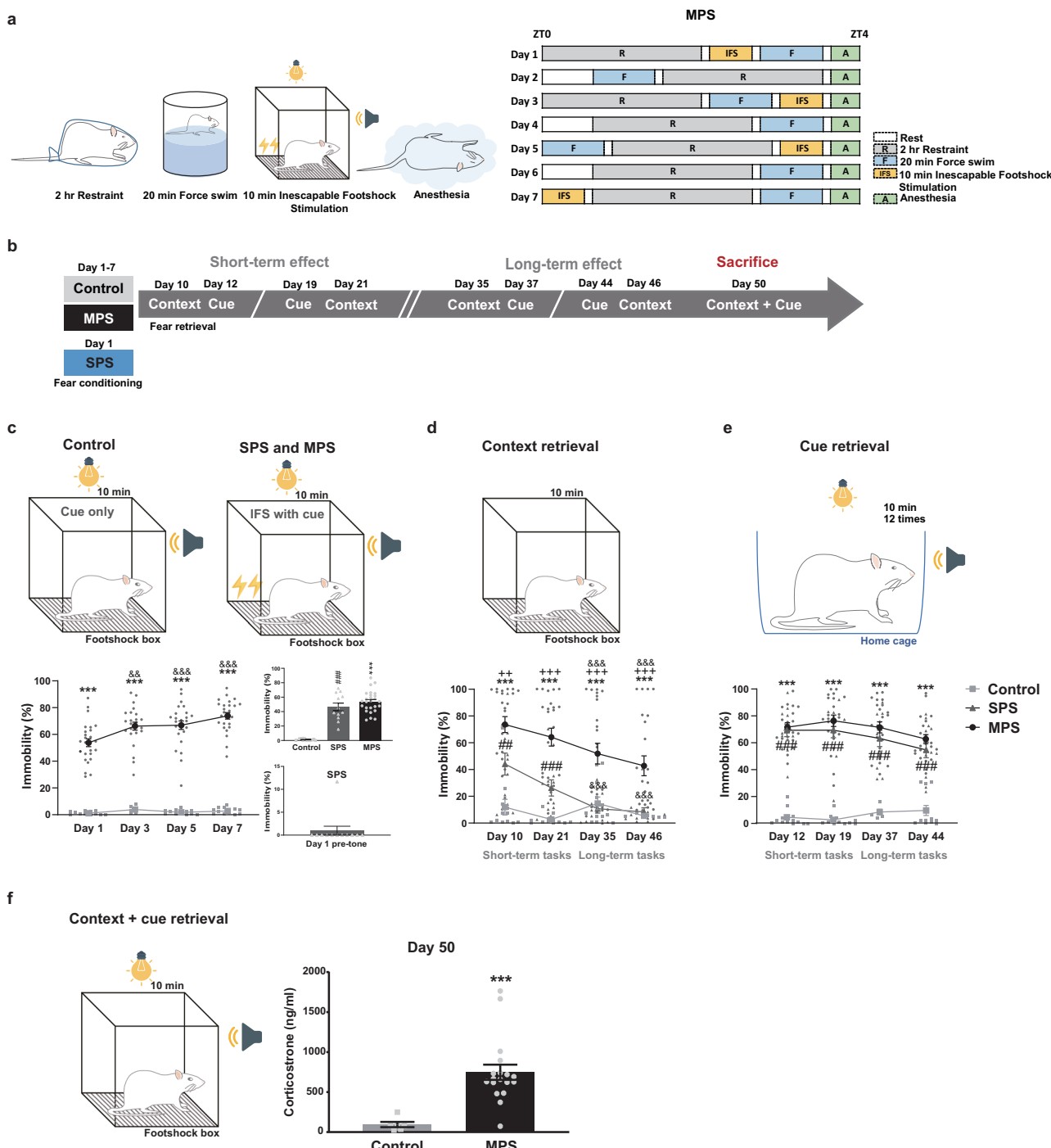

obtained from the home cage and context retrieval than that of control group, reflecting a direct effect of MPS on context memory retrieval (***$p < 0.001$: control vs. MPS; Supplementary Fig. 2a). The control theta power in vHPC and variation values obtained from BLA and vHPC during short-term retrieval were also higher than the MPS group (Day 21; ##$p < 0.01$ in Fig. 2d; ***$p < 0.001$ in Supplementary Fig. 2a), potentially reflecting spatial change's direct impact. This result suggests neuronal theta power escalation is context-dependent; however, we could not exclude the control group's elevated neuronal activity from environmental change. Nevertheless, the MPS group showed a more considerable theta power variance in BLA and vHPC than the control during long-term memory retrieval (#$p < 0.05$, ##$p < 0.01$, ###$p < 0.001$ in Fig. 2d; **$p < 0.01$, ***$p < 0.001$ in Supplementary Fig. 2a).

To investigate theta power variation after cue retrieval, we measured three brain regions' neuronal activity 3 s before (pre) and after (post) cue tone onset (present as 0) (Fig. 2c, e). The MPS-treated 6-s spectrogram displayed significantly intensified power 1-s post-cue onset (Fig. 2e, Supplementary Fig. 3a). Therefore, we also collected this duration to analyze power intensity within fear encoding-correlated regions. Control and MPS groups expressed similar PFC theta oscillation activities, with power escalating post-cue initiation during long-term cue retrieval (Day 44; ***$p < 0.001$: pre- vs. post-cue; Fig. 2f). MPS group's BLA and vHPC neuronal activity intensified after cue onset among the long-term cue retrieval process (Days 37 and 44), while the control group's enhanced during the short-term (Day 19; ***$p < 0.001$; Fig. 2f). Besides, the control group's theta power in PFC was significantly stronger than

**Fig. 1 The MPS protocol elicited a prolonged fear memory. a** The MPS manipulating sequence randomized four stressors: 2-h restraint, 20-min forced swim, 10-min IFS, and a short-term anesthesia within 7 days. The SPS protocol with IFS administrated four stressors on the first experimental day. **b** Control (gray box), MPS (black box), and SPS (blue box) groups were manipulated with context and cue retrieval processes, which are distinguished as short-term (Days 10-21) and long-term (Days 35-50) experimental protocols (see "Methods"). **c-e** Average MPS group immobility (%) increased during IFS procedure (**c**), context retrieval (**d**), and cue retrieval (**e**) within 10 min. **c** MPS group immobility (%) continuously increased during the IFS procedure (left). MPS-Day 1 and SPS manipulation elevated immobility percentage (right upper panel). SPS group immobility also increased on Day 1 and with low (0.97%) pre-tone freezing behavior (right lower panel; gray bar). GEE: Group x Time interaction = 29.358, $p < 0.001$; B = 41.170, 95% CI 35.423-46.916, $p < 0.001$. Control $n = 12$, MPS $n = 26$ on Day 1; control $n = 6$, MPS $n = 24$ on Day 3; control $n = 12$, MPS $n = 25$ on Day 5; control $n = 12$, MPS $n = 25$ on Day 7. Bonferroni's post hoc comparison: [***]$p < 0.001$: control vs. MPS; [&&]$p < 0.01$, [&&&]$p < 0.001$: Day 1 vs. Days 3, 5, and 7 for the MPS group. One-way ANOVA: $F_{(2,47)} = 59.766$, $p < 0.001$. Control $n = 12$, MPS $n = 26$, SPS $n = 12$. Bonferroni's post hoc comparison: [***]$p < 0.001$: control vs. MPS; [###]$p < 0.001$: control vs. SPS. **d** Context retrieval elevated MPS and SPS group immobility (%) and declined during long-term recall. GEE: Group x Time interaction=29.918, $p < 0.001$; B = 8.403, 95% CI 2.651-14.155, $p < 0.01$. Control $n = 12$, MPS $n = 23$, SPS $n = 12$ on Day 10; control $n = 12$, MPS $n = 23$, SPS $n = 12$ on Day 21; control $n = 12$, MPS $n = 23$, SPS $n = 12$ on Day 35; control $n = 6$, MPS $n = 25$, SPS $n = 12$ on Day 46. Bonferroni's post hoc comparison: [***]$p < 0.001$: control vs. MPS; [##]$p < 0.01$, [###]$p < 0.001$: control vs. SPS; [++]$p < 0.01$, [+++]$p < 0.001$: MPS vs. SPS; [&&&]$p < 0.001$: Day 10 vs. Days 21, 35, and 46 of SPS and MPS groups respectively. **e** Cue retrieval increased MPS and SPS group immobility (%) for 6 weeks. GEE: Group x Time interaction=9.984, $p = 0.125$; B = 54.542, 95% CI 43.654-65.429, $p < 0.01$. Control $n = 12$, MPS $n = 19$, SPS $n = 12$ on Day 12; control $n = 12$, MPS $n = 22$, SPS $n = 12$ on Day 19; control $n = 6$, MPS $n = 23$, SPS $n = 11$ on Day 37; control $n = 12$, MPS $n = 22$, SPS $n = 12$ on Day 44. Bonferroni's post hoc comparison: [***]$p < 0.001$: control vs. MPS; [###]$p < 0.001$: control vs. SPS. **f** Corticosterone concentrations heightened after contextual and cue memory retrieval on Day 50. Unpaired $t$-test: [***]$p < 0.001$: control vs. MPS (control $n = 6$, MPS $n = 18$). Values represent the mean ± SEM.

MPS group (Days 19, 37, and 44; [#]$p < 0.05$, [###]$p < 0.001$: control vs. MPS; Fig. 2f); and such enhancement could also be found in the pre- and post-cue onset neuronal power variations (Day 19; [***]$p < 0.001$; Supplementary Fig. 2c). Additionally, the BLA and vHPC power intensities and variations predicted that the instantaneous cue tone response in the control group would be higher than that of MPS during short-term cue retrieval (Day 19; [###]$p < 0.001$ in Fig. 2f; [***]$p < 0.001$ in Supplementary Fig. 2c). However, MPS theta powers and variation values were significantly elevated during remote memory recall compared to the control (Days 37 and 44; [#]$p < 0.05$, [##]$p < 0.01$, [###]$p < 0.001$ in Fig. 2f; [***]$p < 0.001$ in Supplementary Fig. 2c). This indicates that elevated MPS neuronal activity in long-term cue retrievals may be due to paired conditional stimuli (cue tones), representing long-term fear memory effects.

**Post-MPS fear retrieval strengthens PFC, BLA, and vHPC theta oscillation coherence and connectivity.** To further investigate neuronal interactions in fear retrieval, we analyzed theta oscillation coherence and directional connectivity within PFC-BLA, PFC-vHPC, and BLA-vHPC circuitries (Fig. 3a). We found that MPS protocol strengthened PFC-BLA coherence during context retrieval (Day 46; [*]$p < 0.05$: home cage vs. context; Fig. 3b, Supplementary Fig. 1b, c). This implies that fear memory processing efficiency increases after memory recall. Meanwhile, the environment shifts also increased PFC-BLA coherence in the control group during Day 46 ([***]$p < 0.001$; Fig. 3b). PFC-vHPC connectivity also modulated contextual memory retrieval by increasing coherence in control and MPS groups during the long-term period (Day 46; [***]$p < 0.001$; Fig. 3b). We observed similar effects between BLA and vHPC coherency (Days 35 and 46; [*]$p < 0.05$, [**]$p < 0.01$, [***]$p < 0.001$; Fig. 3b, Supplementary Fig. 1b, c). Apart from the environment shift-caused coherence change between brain regions, we found that the basal level connectivity in control group was generally larger than that of MPS group ([###]$p < 0.001$: control vs. MPS; Fig. 3b). Similar effect was also shown in the coherence of variation data, which indicated considerably higher variation in the control group compared to the MPS group after long-term contextual retrieval (Days 35 and 46) because of baseline signal intensity deviations between groups ([**]$p < 0.01$, [***]$p < 0.001$; Supplementary Fig. 2b). Overall, context retrieval intensified fear memory-related brain region integration

post-remote memory recall (Days 35 and 46; [*]$p < 0.05$, [**]$p < 0.01$, [***]$p < 0.001$; Fig. 3b, Supplementary Fig. 1b, c).

Cue retrieval promoted PFC-BLA theta coherence during long-term retrieval in control and MPS groups (Day 44; [*]$p < 0.05$, [***]$p < 0.001$: pre- vs. post-cue; Fig. 3c). Reduced PFC-vHPC coherence was only expressed in the MPS group (Day 37; [**]$p < 0.01$), indicating a decreased PFC-vHPC neuronal interaction following cue tone initiation in MPS rats (Fig. 3c). BLA-vHPC interaction consistently enhanced throughout every cue retrieval day in both control and MPS groups ([*]$p < 0.05$, [**]$p < 0.01$, [***]$p < 0.001$; Fig. 3c). Furthermore, MPS increased theta coherence post-cue tone onset in PFC-BLA (Day 44) and BLA-vHPC (Day 44) compared to the control group ([##]$p < 0.01$: control vs. MPS; Fig. 3c). Such enhancement also displayed in PFC-BLA and BLA-vHPC's coherence variation during long-term retrieval ([***]$p < 0.001$; Supplementary Fig. 2d). While cue tone-induced coherence activation effects within brain regions could not be ruled out, PFC-BLA (Day 19; [###]$p < 0.001$ in Fig. 3c; [*]$p < 0.05$ in Supplementary Fig. 2d) and PFC-vHPC (Day 37; [***]$p < 0.001$; Supplementary Fig. 2d) interactions substantially increased in control rats. These results imply that the cue stimulation during memory retrieval facilitates neuronal interaction (Fig. 3c, Supplementary Fig. 3b, c).

Next, we examined directional transmissions in fear-related brain regions by analyzing Granger causality (G.C.). Dominant neuronal transition is essential for fear memory consolidation and extinction shifts; therefore, we divided our results into two groups based on directional oscillations. PFC to BLA (PFC → BLA), PFC to vHPC (PFC→vHPC), and vHPC to BLA (vHPC→BLA) directions represent prevalent fear extinction processing, while BLA → PFC, vHPC→PFC, and BLA→vHPC moderate fear memory consolidation and retrieval processing. Upon short-term context retrieval (Day 21), we determined that control group causality theta wave propagation in the home cage and context retrievals presented a stronger fear extinction direction ([#]$p < 0.05$, [###]$p < 0.001$: forward transmission direction vs. reverse transmission direction; Fig. 4a). An increased bilateral causality direction was presented at PFC-vHPC (Day 35; [*]$p < 0.05$: home cage vs. context) after shifting control rats in context, which may represent the environment alteration's effect on enhancing PFC-vHPC bilateral interaction (Fig. 4a). Overall, the short-term context retrieval result illustrated neuronal interaction enhancement maybe responsible for safety signal processing in control rats (Fig. 4a). However, the MPS group

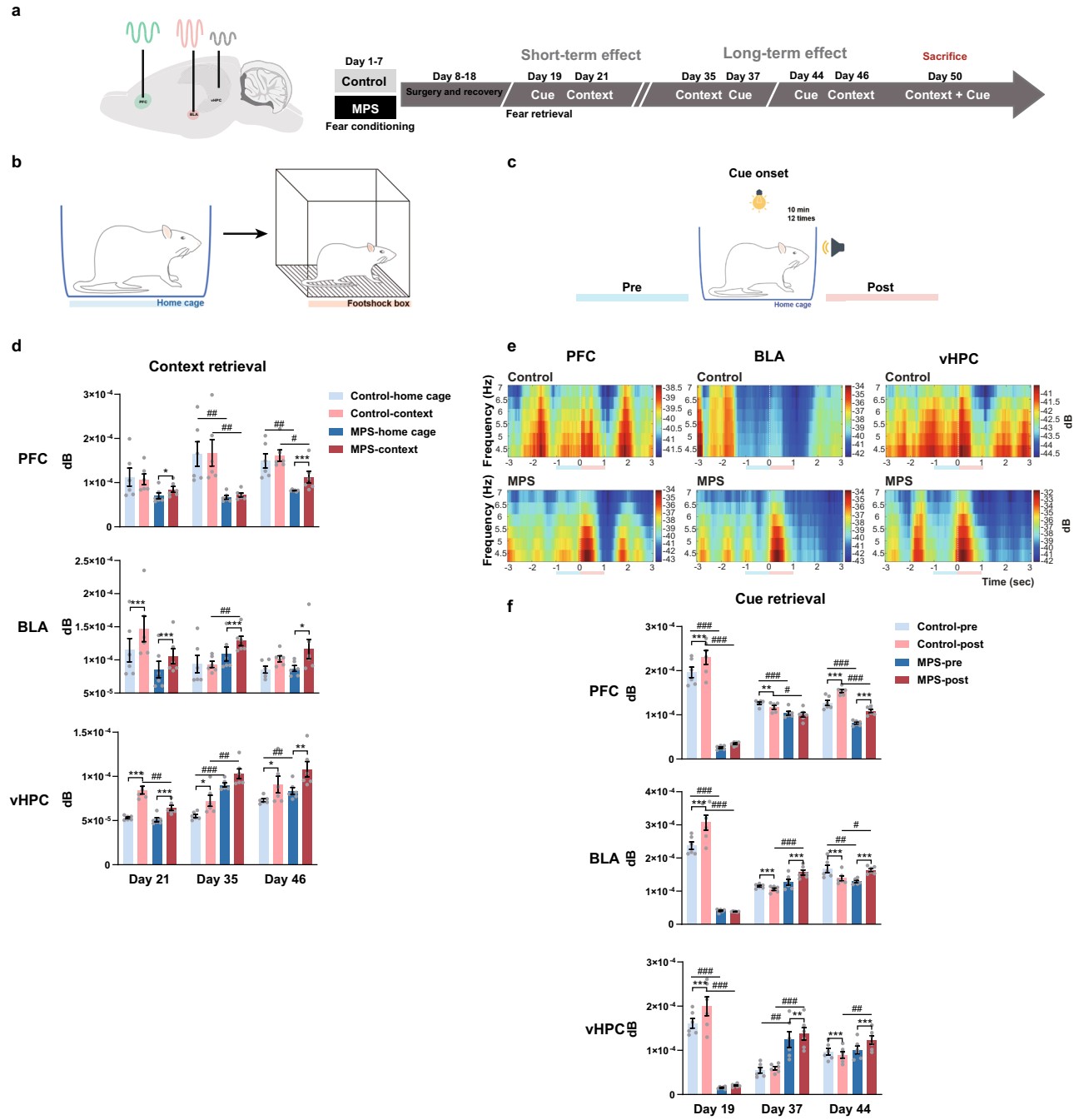

exhibited lower neuronal causality of the connections in PFC-BLA and PFC-vHPC, implying that neuronal randomness in these interactions interrupted signal transmission (Fig. 4a). Nonetheless, an enhanced PFC→vHPC (Day 21; $^\#p < 0.05$) causality was observed before environment changed. Moreover, transmission direction was dominant in the BLA to PFC connectivity after shifting MPS rats to context (BLA → PFC; Day 21; $^\#p < 0.05$), which suggested that context memory retrievals may promote fear consolidation processing during a short-term period other than returning to randomness in the remote period (Fig. 4a). Although cue retrieval dominated G.C. in the fear extinction direction of vHPC→BLA (Day 19) and PFC → BLA (Day 37), a higher transmission was also displayed from vHPC to PFC (vHPC→PFC; Day 37) in control group, which may suggest an elevated vHPC strength in memory retrieval ($^\#p < 0.05$, $^{\#\#}p < 0.01$: forward transmission direction vs. reverse transmission direction; Fig. 4b).

Despite directional strength dwindled on Day 37 following cue introduction, the MPS group's cue retrieval strengthened directional neuronal connections short- (Day 19) and long-term (Day 44; $^*p < 0.05$, $^{**}p < 0.01$: pre- vs. post-cue; Fig. 4b). We also revealed that the corresponding neuronal coherence during cue retrieval was enhanced in fear extinction processing on Days 19 and 37 in the MPS group ($^\#p < 0.05$, $^{\#\#}p < 0.01$; Fig. 4b). However, the dominant direction shifted to fear consolidation after remote cue retrieved on Day 44 ($^\#p < 0.05$, $^{\#\#}p < 0.01$; Fig. 4b). Our data illustrated that cue retrieval caused bidirectional neuronal transmission between PFC-BLA-vHPC connections. The remote cue retrieval in MPS group induced a signal transmission for fear consolidation. Moreover, recent context retrieval triggered a dominant neuronal transmission for fear consolidation. These results indicate that MPS may cause significant stress, which requires repeated extinction process to improve aberrant behavior.

**Fig. 2 Fear memory retrieval reinforced PFC, BLA, and vHPC theta powers. a** LFP recordings and fear memory retrieval protocols (see "Methods"). **b** LFPs were recorded in the home cage as the baseline and in the footshock box as the contextual retrieval for 10 min. **c** For cue retrieval, LFPs were recorded in the home cage with cue tones for 10 min, and neuronal activities were analyzed during the 1-s pre-cue tone (blue line) and the 1-s post-cue tone (red line). **d** Average PFC, BLA, vHPC theta powers were altered after transferring rats from the home cage to context during context retrieval. Two-way repeated measures ANOVA: environment x group interaction on Day 21 $F_{(1,10)} = 4.802$, $p = 0.053$, on Day 35 $F_{(1,10)} = 0.129$, $p = 0.727$, on Day 46 $F_{(1,10)} = 1.527$, $p = 0.245$; main effect of environment on Day 21 $F_{environment(1,10)} = 1.101$, $p = 0.319$, on Day 46 $F_{environment(1,10)} = 7.804$, $p < 0.05$; main effect of group on Day 35 $F_{group\ (1,10)} = 11.223$, $p < 0.05$, on Day 46 $F_{group(1,10)} = 13.795$, $p < 0.01$ in PFC. Environment x group interaction on Day 21 $F_{(1,10)} = 5.549$, $p < 0.05$, on Day 35 $F_{(1,10)} = 2.064$, $p = 0.181$, on Day 46 $F_{(1,10)} = 0.887$, $p = 0.368$; main effect of environment on Day 21 $F_{environment(1,10)} = 100.749$, $p < 0.0001$, on Day 35 $F_{environment(1,10)} = 1.843$, $p = 0.204$, on Day 46 $F_{environment(1,10)} = 10.984$, $p < 0.01$; main effect of group on Day 35 $F_{group(1,10)} = 4.948$, $p < 0.05$ in BLA. Environment x group interaction on Day 21 $F_{(1,10)} = 22.761$, $p < 0.001$, on Day 35 $F_{(1,10)} = 0.266$, $p = 0.617$, on Day 46 $F_{(1,10)} = 0.514$, $p = 0.490$; main effect of environment on Day 21 $F_{environment(1,10)} = 139.202$, $p < 0.0001$, on Day 35 $F_{environment(1,10)} = 10.905$, $p < 0.01$, on Day 46 $F_{environment(1,10)} = 18.615$, $p < 0.01$; main effect of group on Day 21 $F_{group(1,10)} = 9.769$, $p < 0.05$, on Day 35 $F_{group\ (1,10)} = 56.217$, $p < 0.0001$, on Day 46 $F_{group(1,10)} = 2.957$, $p = 0.116$ in vHPC. Control $n = 6$, MPS $n = 6$. Bonferroni's post hoc comparison: $^*p < 0.05$, $^{**}p < 0.01$, $^{***}p < 0.001$: home vs. context; $^\#p < 0.05$, $^{\#\#}p < 0.01$, $^{\#\#\#}p < 0.001$: control vs. MPS. **e** Theta power spectrograms during cue retrievals in the PFC, BLA, and vHPC indicated enhanced theta power during the 1-s after cue onset. The white dashed line represents cue tone onset; blue and red lines depict 1-s pre-cue and 1-s post-cue tone periods, respectively. **f** Average theta powers changed during the 1-s post-cue tone period compared to those obtained during the 1-s pre-cue tone period. Two-way repeated measures ANOVA: period x group interaction on Day 19 $F_{(1,10)} = 14.200$, $p < 0.01$, on Day 37 $F_{(1,10)} = 1.083$, $p = 0.323$, on Day 44 $F_{(1,10)} = 0.032$, $p = 0.862$; main effect of period on Day 19 $F_{period(1,10)} = 39.646$, $p < 0.0001$, on Day 37 $F_{period(1,10)} = 5.633$, $p < 0.05$, on Day 44 $F_{period(1,10)} = 185.645$, $p < 0.0001$; main effect of group on Day 19 $F_{group(1,10)} = 182.311$, $p < 0.0001$, on Day 37 $F_{group(1,10)} = 13.954$, $p < 0.05$, on Day 44 $F_{group(1,10)} = 70.602$, $p < 0.0001$ in PFC. Period x group interaction on Day 19 $F_{(1,10)} = 31.945$, $p < 0.0001$, on Day 37 $F_{(1,10)} = 215.244$, $p < 0.0001$, on Day 44 $F_{(1,10)} = 139.108$, $p < 0.0001$; main effect of period on Day 19 $F_{period(1,10)} = 27.794$, $p < 0.0001$, on Day 37 $F_{period(1,10)} = 58.858$, $p < 0.0001$, on Day 44 $F_{period(1,10)} = 1.733$, $p = 0.217$; main effect of group on Day 19 $F_{group(1,10)} = 193.934$, $p < 0.0001$, on Day 37 $F_{group(1,10)} = 13.337$, $p < 0.01$, on Day 44 $F_{group(1,10)} = 0.450$, $p = 0.517$ in BLA. Period x group interaction on Day 19 $F_{(1,10)} = 9.034$, $p < 0.05$, on Day 37 $F_{(1,10)} = 2.312$, $p = 0.159$, on Day 44 $F_{(1,10)} = 172.006$, $p < 0.0001$; main effect of period on Day 19 $F_{period(1,10)} = 15.024$, $p < 0.01$, on Day 37 $F_{period(1,10)} = 10.530$, $p < 0.01$, on Day 44 $F_{period(1,10)} = 44.881$, $p < 0.0001$; main effect of group on Day 19 $F_{group(1,10)} = 104.841$, $p < 0.0001$, on Day 37 $F_{group(1,10)} = 19.732$, $p < 0.001$, on Day 44 $F_{group(1,10)} = 2.561$, $p = 0.141$ in vHPC. Control $n = 6$, MPS $n = 6$. Bonferroni's post hoc comparison: $^{**}p < 0.01$, $^{***}p < 0.001$: 1-s pre-cue tone vs. 1-s post-cue tone; $^\#p < 0.05$, $^{\#\#}p < 0.01$, $^{\#\#\#}p < 0.001$: control vs. MPS. Values represent the mean ± SEM. The detail statistical values were presented in Supplementary Table. 1.

**MPS increased anxiety and neuronal activity in fear-processing circuitry.** Stress-induced flashbacks provoked AMY, PFC, and HPC activation, which share the same hyperresponsivity regions that cause abnormal behaviors as anxiety[51–53]. Hence, after fear retrievals, we evaluated anxiety (Fig. 5a) and related neuronal activities (Fig. 5b) from elevated plus maze (EPM) and open field test (OFT) tasks. We divided the experimental groups into short-term or long-term assessments to prevent rats from acclimating to behavioral tasks due to repeated pattern exposures (Fig. 5a, b). EPM behavioral results demonstrated that the MPS protocol significantly decreased the frequency and time of rats exploring in the open arms ($^*p < 0.05$, $^{**}p < 0.01$, $^{***}p < 0.001$), indicating an increased anxiety level after fear retrieval (Fig. 5c). Next, we assessed PFC, BLA, and vHPC neuronal power intensities as rats concurrently resided in each zone (Supplementary Fig. 4a). Corresponding neuronal powers diminished when rats were in open arms after conditional cue tone retrieval in the short-term assessment (Day 19), whereas they increased in long-term for context (Day 35) and cue retrievals (Day 44; $^{***}p < 0.001$; Fig. 5d). However, statistical analysis accuracy was limited due to the low open arms entries in MPS rats. Thus, we calculated neuronal activity strength in closed arms and found that theta power decreased after long-term context retrieval (Day 35; $^*p < 0.05$, $^{***}p < 0.001$). An opposite phenomenon was observed after cue retrieval, indicating conditional cue stimuli had to be stronger than contextual retrieval as they prompted stress-increasing neuronal activity in the EPM (Day 44; $^{***}p < 0.001$; Fig. 5d).

OFT results also demonstrated increased anxiety levels through low entry frequency and accumulated time in the inner zone ($^*p < 0.05$, $^{**}p < 0.01$; Fig. 5e). Neuronal power intensities were analyzed when rats remained in inner and outer zones (Fig. 5f, Supplementary Fig. 4b). PFC and BLA regions exhibited distinct presentations in the in the inner zone during short-term context retrieval (Day 21), before reducing after long-term context retrieval (Day 46; $^{***}p < 0.001$; Fig. 5f). Additionally, cue retrieval increased activity during the long-term period (Day 37;

$^{***}p < 0.001$). Still, these MPS group results may be biased due to rare entry into the inner zone. Consequently, our neuronal activity observations focused on the outer zone and confirmed enhanced theta powers after context and cue retrievals ($^{***}p < 0.001$; Fig. 5f). Together, the MPS protocol provoked extended anxiety, and raised neuronal activity in fear-associated regions (PFC, BLA, and vHPC) during behavior tasks, signifying rats exhibited hyper-responses after fear memory retrieval during the remote time segment.

**MPS protocol and fear retrieval elicited sleep alterations.** After a traumatic event, sleep-wake activity notably enhances fear memory consolidation[54,55]. Therefore, we recorded sleep alteration and architecture in bout numbers, bout duration (min), and vigilance transitions post-MPS and fear retrieval evaluation to determine whether aberrational sleep correlated with behavioral abnormalities (Fig. 6a, Tables 1, 2, 3). Since the MPS protocol took ~4-h, we established a 4-h sleep deprivation (SD) as the control to expound SD effects. Our results revealed that after 2 days of MPS delivery, non-rapid eye movement (NREM) sleep rose during zeitgeber time (ZT) 13-24 (dark period) and continued until Day 7, with the increase attributed to bout numbers and bout duration ($^*p < 0.05$, $^{**}p < 0.01$, $^{***}p < 0.001$: baseline vs. 4-h SD and Days 1–7; $^\#p < 0.05$, $^{\#\#}p < 0.01$, $^{\#\#\#}p < 0.001$: 4-h SD vs. Days 1–7; Fig. 6b, Table 1). However, as a result of REM sleep bout numbers and duration fluctuations, an initial REM sleep suppression was observed during ZT5-12 over the entire MPS process, which was later compensated at ZT13-24 ($^*p < 0.05$, $^{**}p < 0.01$, $^{***}p < 0.001$: baseline vs. 4-h SD and Days 1–7; $^\#p < 0.05$, $^{\#\#}p < 0.01$, $^{\#\#\#}p < 0.001$: 4-h SD vs. Days 1-7; Fig. 6b, Table 1). We then determined sleep variation following contextual or cue fear retrievals. Both retrieval processes did not alter NREM sleep (Fig. 6c, d). In contrast, REM sleep reduction was elicited for 12 h (ZT1-12) after long-term context (Day 35) and cue retrievals (Day 37), which were mediated by bout numbers or

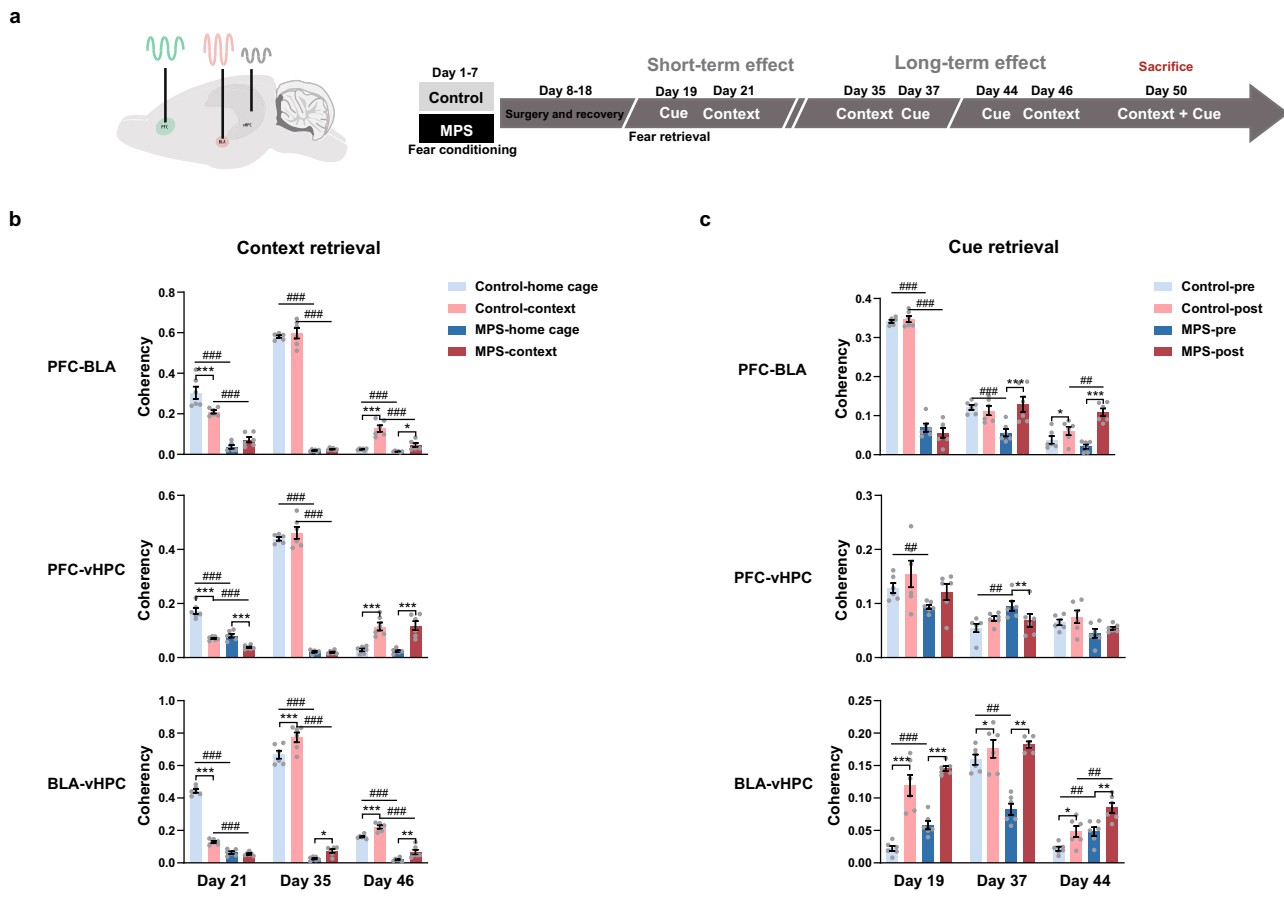

**Fig. 3 Fear memory retrieval enhanced PFC-BLA-vHPC circuitry coherency. a** LFP recordings and fear memory retrieval protocol (see "Methods").
**b**, **c** Theta coherence within PFC-BLA, PFC-vHPC, and BLA-vHPC circuitries during context retrieval (**b**) and cue retrieval (**c**). **b** Neuronal coherence was analyzed between LFPs obtained from the home cage (the baseline) and contextual retrieval within 10 min. Two-way repeated measures ANOVA: environment x group interaction on Day 21 $F_{(1, 10)} = 19.826$, $p < 0.001$, on Day 35 $F_{(1, 10)} = 0.223$, $p = 0.647$, on Day 46 $F_{(1, 10)} = 16.131$, $p < 0.01$; main effect of environment on Day 21 $F_{environment(1, 10)} = 4.170$, $p = 0.068$, on Day 46 $F_{environment(1, 10)} = 59.034$, $p < 0.0001$; main effect of group on Day 21 $F_{group(1, 10)} = 101.392$, $p < 0.0001$, on Day 35 $F_{group(1, 10)} = 1313.358$, $p < 0.0001$, on Day 46 $F_{group(1, 10)} = 22.104$, $p < 0.001$ in PFC-BLA. Environment x group interaction on Day 21 $F_{(1, 10)} = 21.211$, $p < 0.001$, on Day 35 $F_{(1, 10)} = 1.786$, $p = 0.211$, on Day 46 $F_{(1, 10)} = 0.182$, $p = 0.679$; main effect of environment on Day 21 $F_{environment(1, 10)} = 131.485$, $p < 0.0001$, on Day 46 $F_{environment(1, 10)} = 94.912$, $p < 0.0001$; main effect of group on Day 21 $F_{group(1, 10)} = 66.890$, $p < 0.0001$, on Day 35 $F_{group(1, 10)} = 968.437$, $p < 0.0001$ in PFC-vHPC. Environment x group interaction on Day 21 $F_{(1, 10)} = 596.370$, $p < 0.0001$, on Day 35 $F_{(1, 10)} = 7.200$, $p < 0.05$, on Day 46 $F_{(1, 10)} = 0.491$, $p = 0.500$; main effect of environment on Day 21 $F_{environment(1, 10)} = 662.074$, $p < 0.0001$, on Day 35 $F_{environment(1, 10)} = 50.214$, $p < 0.0001$, on Day 46 $F_{environment(1, 10)} = 33.200$, $p < 0.0001$; main effect of group on Day 21 $F_{group(1, 10)} = 1353.968$, $p < 0.0001$, on Day 35 $F_{group(1, 10)} = 620.400$, $p < 0.0001$, on Day 46 $F_{group(1, 10)} = 216.516$, $p < 0.0001$ in BLA-vHPC. Control $n = 6$, MPS $n = 6$. Bonferroni's post hoc comparison: *$p < 0.05$, **$p < 0.01$, ***$p < 0.001$: home vs. context; ###$p < 0.001$: control vs. MPS. **c** Cue retrieval enhanced the neuronal coherence during the 1-s post-cue tone compared to the 1-s pre-cue tone. Two-way repeated measures ANOVA: period x group interaction on Day 19 $F_{(1, 10)} = 2.373$, $p = 0.154$, on Day 37 $F_{(1, 10)} = 19.142$, $p < 0.001$, on Day 44 $F_{(1, 10)} = 34.364$, $p < 0.0001$; main effect of period on Day 37 $F_{period(1, 10)} = 12.141$, $p < 0.01$, on Day 44 $F_{period(1, 10)} = 99.491$, $p < 0.0001$; main effect of group on Day 19 $F_{group(1, 10)} = 598.203$, $p < 0.0001$, on Day 37 $F_{group(1, 10)} = 2.446$, $p = 0.149$, on Day 44 $F_{group(1, 10)} = 1.627$, $p = 0.231$ in PFC-BLA. Period x group interaction on Day 19 $F_{(1, 10)} = 0.002$, $p = 0.962$, on Day 37 $F_{(1, 10)} = 11.122$, $p < 0.01$; main effect of period on Day 37 $F_{period(1, 10)} = 0.510$, $p = 0.492$; main effect of group on Day 19 $F_{group(1, 10)} = 6.428$, $p < 0.05$, on Day 37 $F_{group(1, 10)} = 3.168$, $p = 0.105$ in PFC-vHPC. Period x group interaction on Day 19 $F_{(1, 10)} = 0.268$, $p = 0.616$, on Day 37 $F_{(1, 10)} = 69.187$, $p < 0.0001$, on Day 44 $F_{(1, 10)} = 0.633$, $p = 0.445$; main effect of period on Day 19 $F_{period(1, 10)} = 83.048$, $p < 0.0001$, on Day 37 $F_{period(1, 10)} = 136.010$, $p < 0.0001$, on Day 44 $F_{period(1, 10)} = 24.256$, $p < 0.001$; main effect of group on Day 19 $F_{group(1, 10)} = 14.535$, $p < 0.01$, on Day 37 $F_{group(1, 10)} = 8.161$, $p < 0.05$, on Day 44 $F_{group(1, 10)} = 17.805$, $p < 0.01$ in BLA-vHPC. Control $n = 6$, MPS $n = 6$. Bonferroni's post hoc comparison: *$p < 0.05$, **$p < 0.01$, ***$p < 0.001$: 1-s pre-cue tone vs. 1-s post-cue tone; ##$p < 0.01$, ###$p < 0.001$: control vs. MPS. Values represent the mean ± SEM. The detail statistical values were presented in Supplementary Table. 2.

bout duration changes (*$p < 0.05$, **$p < 0.01$, ***$p < 0.001$: baseline vs. fear retrieval days; Fig. 6c, d, Tables 2, 3). These results suggest that MPS and long-term memory retrieval coordinated sleep architecture and curtailed REM sleep, which resembled the immediate stress response of high-intensity fear acquisition and MPS-induced retrieval.

**MPS and fear retrieval augmented SWA and spindle power during NREM sleep.** Delta oscillation synchronization during NREM sleep contributes to fear memory consolidation; thus, we evaluated slow-wave activity (SWA) and sleep spindle intensity to correspond with behavioral alteration during fear memory recalls. MPS increased SWA for 20 h, including the immediate response from ZT5-12, which reached significance at Day 3, and ZT13-24 during Days 3–5 (*$p < 0.05$, ***$p < 0.001$: baseline vs. 4-h SD and Days 1-7; #$p < 0.05$: 4-h SD vs. Days 1–7; Fig. 7a). NREM sleep oscillation activity also mediates memory reconsolidation after fear retrieval. In addition, short- and long-term contextual and cue

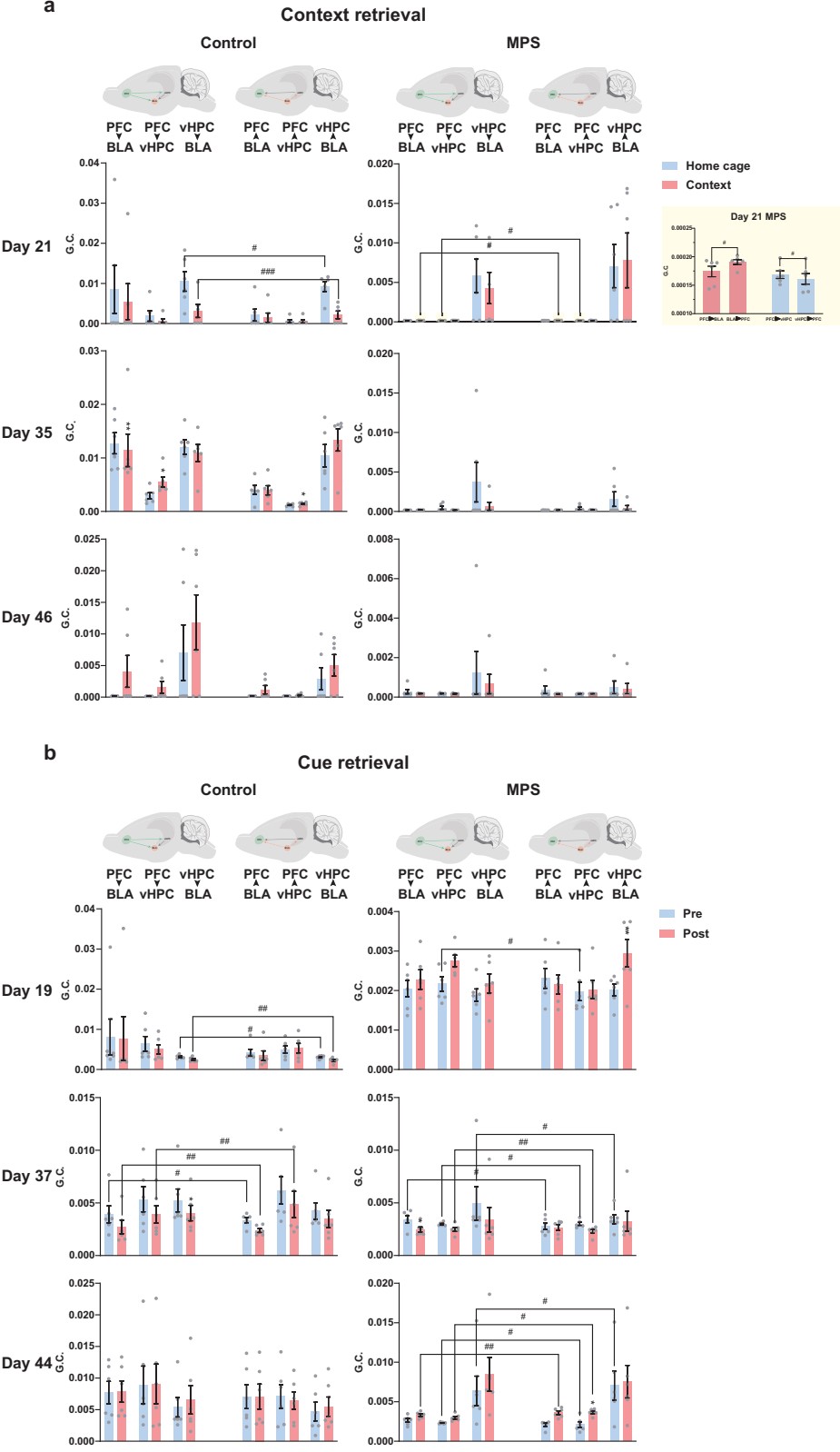

recalls ($^{***}p < 0.001$; Fig. 7b, c) intensified the SWA, which indicates a comprehensive augmented delta slow-wave during NREM sleep for 24 h compared with the baseline.

To further elucidate the sporadic sleep spindle for consolidating memories after MPS and fear retrieval, we assessed 12–15 Hz oscillation powers during NREM sleep. We found that MPS

increased sleep spindle power compared to the baseline and 4-h SD groups ($^{***}p < 0.001$; $^{###}p < 0.001$; Fig. 7d). Higher sleep spindle powers transpired after context and cue memory retrievals ($^{*}p < 0.05$, $^{***}p < 0.001$; Fig. 7e, f). MPS manipulation and fear recall enhanced SWA and sleep spindle synchronizations during NREM sleep, strengthening fear memory reconsolidation.

**Fig. 4 Fear memory retrieval enhanced PFC-BLA-vHPC circuitry connectivity. a, b** Granger causality within PFC-BLA, PFC-vHPC, and BLA-vHPC circuitries during context retrieval (**a**) and cue retrieval (**b**). **a** Contextual retrieval altered the Granger causality of neuronal connectivity and signal transmission direction when rats were transferred from home cage to contextual cage. The yellow background bar graph enlarged the significant differences in Day 21 MPS PFC-BLA and PFC-vHPC interactions. Two-way repeated measures ANOVA: environment x direction interaction on Day 21 BLA-vHPC $F_{(1,5)} = 0.033$, $p = 0.863$, on Day 35 PFC-BLA $F_{(1,5)} = 0.282$, $p = 0.618$, on Day 35 PFC-vHPC $F_{(1,5)} = 4.319$, $p = 0.092$; main effect of environment on Day 35 PFC-BLA $F_{environment(1,5)} = 12.612$, $p < 0.05$, on Day 35 PFC-vHPC $F_{environment(1,5)} = 26.073$, $p < 0.01$; main effect of direction on Day 21 BLA-vHPC $F_{direction(1,5)} = 26.652$, $p < 0.01$ in control group. Environment x direction interaction on Day 21 PFC-BLA $F_{(1,5)} = 1.134$, $p = 0.336$, on Day 21 PFC-vHPC $F_{(1,5)} = 0.321$, $p = 0.596$; main effect of direction on Day 21 PFC-BLA $F_{direction(1,5)} = 11.044$, $p < 0.05$, on Day 21 PFC-vHPC $F_{direction(1,5)} = 5.039$, $p = 0.075$ in MPS group. Control $n = 6$, MPS $n = 6$. Bonferroni's post hoc comparison: $^*p < 0.05$, $^{**}p < 0.01$: home vs. context; $^\#p < 0.05$, $^{\#\#\#}p < 0.001$: forward transmission direction vs. reverse transmission direction. **b** Cue retrieval changed the Granger causality when compared between 1-s post-cue tone and 1-s pre-cue tone. Period x direction interaction on Day 19 BLA-vHPC $F_{(1,5)} = 0.830$, $p = 0.404$, on Day 37 PFC-BLA $F_{(1,5)} = 0.113$, $p = 0.751$, on Day 37 PFC-vHPC $F_{(1,5)} = 0.042$, $p = 0.846$, on Day 37 BLA-vHPC $F_{(1,5)} = 0.169$, $p = 0.698$; main effect of period on Day 37 BLA-vHPC $F_{period(1,5)} = 2.227$, $p = 0.196$; main effect of direction on Day 19 BLA-vHPC $F_{direction(1,5)} = 29.845$, $p < 0.01$, on Day 37 PFC-BLA $F_{direction(1,5)} = 57.217$, $p < 0.001$, on Day 37 PFC-vHPC $F_{direction(1,5)} = 14.736$, $p < 0.05$ in control group. Period x direction interaction on Day 19 PFC-vHPC $F_{(1,5)} = 1.406$, $p = 0.289$, on Day 19 BLA-vHPC $F_{(1,5)} = 4.282$, $p = 0.093$, on Day 37 PFC-BLA $F_{(1,5)} = 32.303$, $p < 0.01$, on Day 37 PFC-vHPC $F_{(1,5)} = 1.587$, $p = 0.263$, on Day 37 BLA-vHPC $F_{(1,5)} = 1.661$, $p = 0.254$, on Day 44 PFC-BLA $F_{(1,5)} = 4.255$, $p = 0.094$, on Day 44 PFC-vHPC $F_{(1,5)} = 2.785$, $p = 0.156$, on Day 44 BLA-vHPC $F_{(1,5)} = 4.877$, $p = 0.078$; main effect of period on Day 19 BLA-vHPC $F_{period(1,5)} = 7.132$, $p < 0.05$, on Day 37 PFC-BLA $F_{period(1,5)} = 1.102$, $p = 0.342$, on Day 44 PFC-vHPC $F_{period(1,5)} = 3.501$, $p = 0.120$; main effect of direction on Day 19 PFC-vHPC $F_{direction(1,5)} = 2.660$, $p = 0.164$, on Day 37 PFC-BLA $F_{direction(1,5)} = 2.021$, $p = 0.214$, on Day 37 PFC-vHPC $F_{direction(1,5)} = 29.987$, $p < 0.01$, on Day 37 BLA-vHPC $F_{direction(1,5)} = 20.390$, $p < 0.01$, on Day 44 PFC-BLA $F_{direction(1,5)} = 21.055$, $p < 0.01$, on Day 44 PFC-vHPC $F_{direction(1,5)} = 18.820$, $p < 0.01$, on Day 44 BLA-vHPC $F_{direction(1,5)} = 15.132$, $p < 0.05$ in MPS group. Control $n = 6$, MPS n = 6. Bonferroni's post hoc comparison: $^*p < 0.05$, $^{**}p < 0.01$: 1-s pre-cue tone vs. 1-s post-cue tone; $^\#p < 0.05$, $^{\#\#}p < 0.01$: forward transmission direction vs. reverse transmission direction. Values represent the mean ± SEM. The detail statistical values were presented in Supplementary Table. 3.

**MPS and fear retrieval enhanced theta power during REM sleep for fear memory consolidation**. Neural replay during REM sleep is crucial for fear memory consolidation and reconsolidation after memory recalls. MPS results exhibited a high REM sleep theta power lasting ~24 h compared to that after 4-h SD ($^\#p < 0.05$, $^{\#\#}p < 0.01$, $^{\#\#\#}p < 0.001$). This, in turn, reached higher statistical significance during the light period (ZT5-12) on Days 3-6 and the dark period (ZT13-24) on Day 5 compared with the baseline ($^*p < 0.05$; Fig. 7g). During contextual retrieval, theta power increments only occurred during the long-term effect on Day 35 (Fig. 7h). A similar long-term impact also developed after cue retrievals ($^*p < 0.05$, $^{***}p < 0.001$; Fig. 7i), emphasizing that theta oscillation intensity in long-term fear reconsolidation was more dominant than REM sleep quantity.

**MPS and fear retrieval increased innate anxiety**. Theta activity analysis during wakefulness determined anxiety levels. Waking theta powers post-MPS were mainly gained during ZT13-24, in which the rats were more awake ($^*p < 0.05$, $^{***}p < 0.001$: baseline vs. 4-h SD and Days 1–7; $^\#p < 0.05$, $^{\#\#\#}p < 0.001$: 4-h SD vs. Days 1–7; Fig. 7j). Additionally, context retrieval consistently aroused waking theta powers for 24 h, contrasting that of the baseline ($^*p < 0.05$, $^{**}p < 0.01$, $^{***}p < 0.001$; Fig. 7k). A similar effect was also noted after cue retrieval as increased waking theta intensities persisted throughout the 7 weeks ($^*p < 0.05$, $^{**}p < 0.01$, $^{***}p < 0.001$; Fig. 7l). Fear acquisition and retrieval differences may be due to SD during the MPS period. Nevertheless, an overall increase in waking theta power illustrated that MPS stimulation and fear retrieval elevates rodent anxiety levels.

## Discussion

This study evaluated MPS effects on memory processing efficiency, neuronal circuitry, anxiety, and sleep-wake activity in an enduring modified PTSD-like rodent model. MPS prolonged stress response by increasing stressor intensity and stochasticity during fear acquisition. We determined that 6 weeks after MPS, remote fear memory could be retrieved after context and cue recall. In contrast, SPS reveals a memory preservation constrained within 2 weeks after context recall, suggesting that MPS substantially affects fear memory persistence. Moreover, MPS-enhanced neuronal oscillations and directional transitioned randomness within PFC-BLA-vHPC circuitries consisted throughout remote memory retrievals, reflecting a neuronal response to learned fear memory. Memory retrieval tasks also increased innate anxiety and sleep disruptions throughout 6 weeks process. REM sleep disruption and power oscillation alterations during NREM and REM sleep post-MPS may prolong fear memory post-remote memory retrieval. Our results specify that MPS' role in fear memory processing simulates PTSD-like symptoms.

Clinical studies report psychological effects after extended exposure to aversive stimulation and that PTSD symptomatology development highly correlates with traumatic event experiences and post-event neuronal abnormalities[56,57]. Correspondingly, PTSD-like pathological progression exhibited in animal models demonstrates that stressor manipulation provokes similar fear expression, neuroendocrine system alterations, anxiety, and sleep disorders when animals retrieve fear memory, revealing these stimulators are potentially adaptable for clinical use[6,7,43]. We emulated SPS model stimuli for 7 days to decrease stress tolerance, used the IFS as the major stressor to imitate the chronic traumatic stress, and assessed the long-term PTSD-related irregularity functions through consequent symptomatic indicators. Contextual and cue retrieval (the conditioned stimulators) data demonstrated increased immobilization. Similarly, the footshock stimulation and SPS raised rodent freezing behavior during fear memory retrieval, representing preserved emotional memories expressed towards learned events[7,58]. Long-term observation data revealed that SPS-manipulated rats could not maintain contextual fear memory after 2 weeks, and immobilization during cued memory retrieval declined after 7 weeks (Fig. 1d, e). Additional studies noted that PFC, BLA, and HPC reactivated fear memory retrieving neurons to assist remote fear memory attenuation[59–62]. Moreover, retrieval (reconsolidation) generated labile in consolidated memory led to significant experience updating for recent (2–7 days) but not remote memories (14–28 days), indicating older memories are more resistant to disruption[63–65]. Our experimental design repeatedly exposed rats to the same conditioned context and cue tones (sound-light compounded) and constantly reactivating consolidated remote memory to

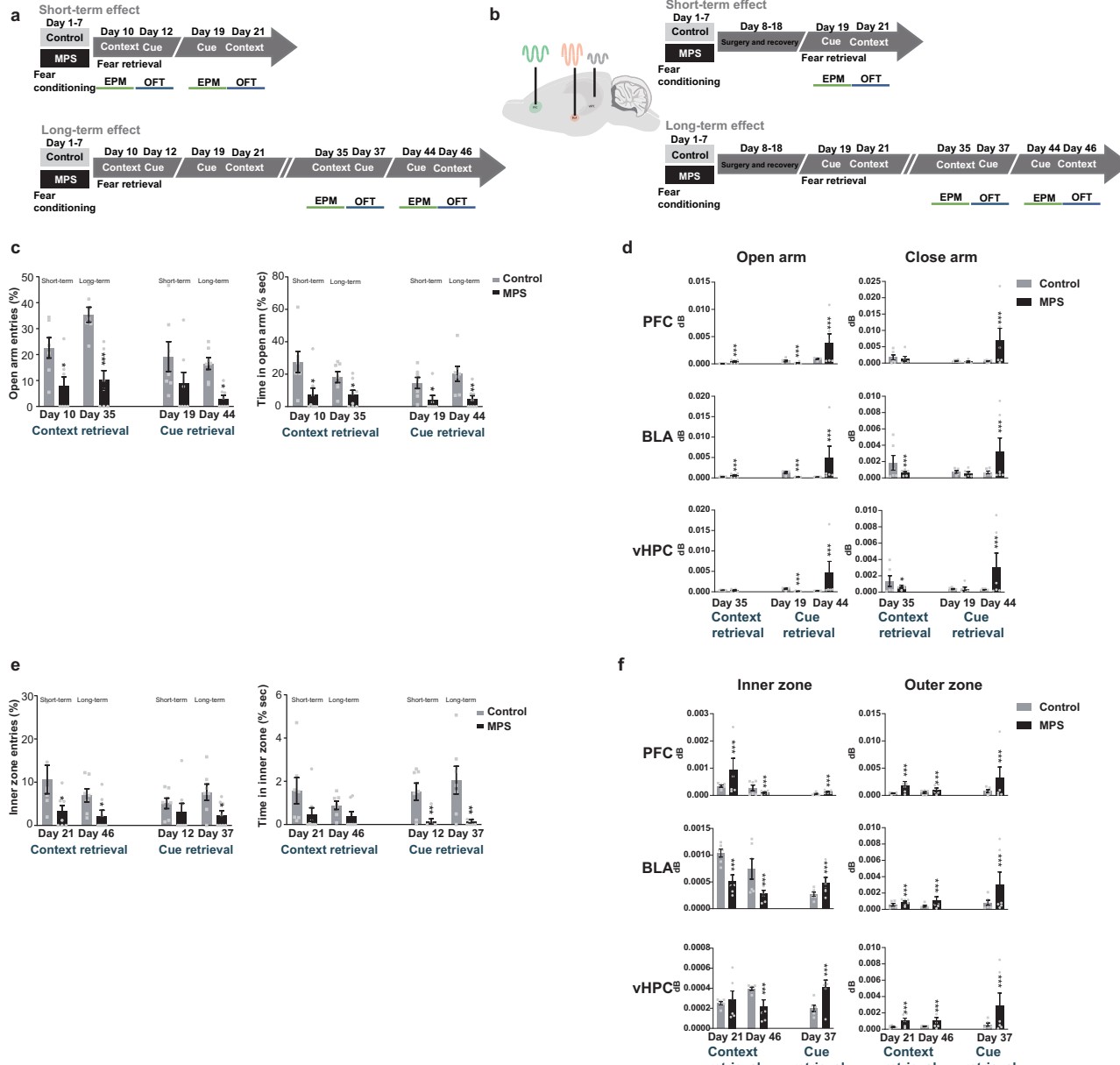

detect any remaining fear for 6 weeks. We further examined the immobility (%) of single-time remote memory retrieval tasks to exclude interference from repeated memory recall-induced neuronal reactivation. Rats were only exposed to the conditioned context (Day 46) or cue tones (Day 44) one-time post-MPS manipulation. Our results displayed no differences between single and repeated retrieval examination (Supplementary Fig. 5), indicating that repeated context and cue tone exposures did not generate reminder or extinction effects on MPS rats. Although retrieval during a recent period may enhance memory retention, the high immobility response presented within 6 weeks post-MPS was not disturbed by repeated memory recall, indicating MPS severe affects fear persistence. These results corroborate that MPS enhances fear preservation, which learned-dependent memory retrieves during the remote period post-MPS. PTSD development is memory-dependent and requires emotional memory consolidation towards an aversive event[66]. Thus, fear memory retrieval in our experiments specifically recalled IFS-dependent conditioned stimuli (footshock context and compounded sound-light cue tone). The MPS process manipulated multiple stressors

to induce PTSD-like symptoms susceptibility after a main traumatic event (IFS stimulation), which modeled human PTSD risk factors before a traumatic experiences[20,67]. Prior stress exposure enhanced the fear learning tasks also described in the stress-induced enhancement of fear learning (SEFL) model[68]. SEFL utilized 15 unpredictable and inescapable electric footshock stimuli within a 90-min experimental period before a single shock stimulation (1 mA, 1 s for each shock) to increase the rat's sensitivity toward a subsequent mild stressor (single shock) and induced long-term PTSD-like symptoms[68,69]. Previous studies verified that pre-exposure to repeated shocks enhances fear learning ability to the following fear-conditioned paired context or cue tone, which might be due to increased anxiety after shock stress[68,70]. These results complement our findings that multiple stressors manipulation before the main shock (IFS) intensified contextual and cue-paired fear learning efficacy in the rats[71,72]. Moreover, fear memory retention requires functional processing in multiple brain regions to enhance neuronal plasticity and synchronization in the PFC, BLA, and vHPC[28–31,50]. Therefore, we further evaluated the interaction within infralimbic (IL) of the

**Fig. 5 Fear memory retrieval promoted anxiety and neuronal activity. a** Anxiety levels in short- and long-term alterations were assessed through EPM (green line) and OFT (blue line) (see "Methods"). **b** LFPs were simultaneously acquired when rats were performed EPM (green line) and OFT (blue line) (see "Methods"). **c, d** Behavioral alterations and correlated neuronal activities during the EPM 5-min period. **c** Open arm entry frequency (left) and accumulated time (right) (%) decreases were promoted by context (Days 10 and 35 short- and long-term recall) and cue retrievals (Days 19 and 44 short- and long-term recall). Two-way repeated measures ANOVA: time x group interaction $F_{(1,13)} = 2.881$, $p = 0.113$; main effect of group $F_{group(1,13)} = 26.550$, $p < 0.0001$ in context recall open arm entry frequencies (%) (left) on Days 10 and 35. Time x group interaction $F_{(1,13)} = 2.261$, $p = 0.157$; main effect of group $F_{group(1,13)} = 8.318$, $p < 0.05$ in context recall open arm accumulated time (right) (%) on Days 10 and 35. Time x group interaction $F_{(2,1572)} = 0.169$, $p = 0.688$; main effect of group $F_{group(1,13)} = 11.528$, $p < 0.01$ in cue recall open arm entry frequencies (%) (left) on Days 19 and 44. Time x group interaction $F_{(1,13)} = 0.451$, $p = 0.514$; main effect of group $F_{group(1,13)} = 26.969$, $p < 0.0001$ in cue recall open arm accumulated time (right) (%) on Days 19 and 44. Control $n = 7$, MPS $n = 8$. Bonferroni's post hoc comparison: $^*p < 0.05$, $^{**}p < 0.01$, $^{***}p < 0.001$: control vs. MPS. **d** Average theta powers in each exploration zone was altered after memory retrievals. Control $n = 6$, MPS $n = 6$. Unpaired $t$-test: $^*p < 0.05$, $^{**}p < 0.01$, $^{***}p < 0.001$: control vs. MPS. **e, f** Behavioral alterations and correlated neuronal activities during 10-min OFT. **e** Inner zone entry frequency (left) and accumulated time (right) (%) were reduced by context retrieval (Day 21 of short-term recall) and cue retrieval (Days 12 and 37 of short- and long-term recall). Two-way repeated measures ANOVA: time x group interaction $F_{(1,13)} = 0.489$, $p = 0.497$; main effect of group $F_{group(1,13)} = 8.476$, $p < 0.05$ in context recall inner zone entry frequency (%) (left) on Days 21 and 46. Time x group interaction $F_{(1,13)} = 0.752$, $p = 0.402$; main effect of group $F_{group(1,13)} = 19.788$, $p < 0.001$ on context recall inner zone accumulated time (%) (right) on Days 21 and 46. Time x group interaction $F_{(1,13)} = 1.794$, $p = 0.203$; main effect of group $F_{group(1,13)} = 3.849$, $p = 0.072$ in cue recall inner zone entry frequency (%) (left) on Days 12 and 37. Time x group interaction $F_{(1,13)} = 0.476$, $p = 0.503$; main effect of group $F_{group(1,13)} = 22.947$, $p < 0.0001$ on cue recall inner zone accumulated time (%) (right) on Days 12 and 37. Control $n = 7$, MPS $n = 8$. Bonferroni's post hoc comparison: $^*p < 0.05$, $^{**}p < 0.01$: control vs. MPS. **f** Average theta powers in each exploration zone was altered after memory retrievals. Control $n = 6$, MPS $n = 6$. Unpaired $t$-test: $^{***}p < 0.001$: control vs. MPS. Values represent the mean ± SEM. The detail statistical values were presented in Supplementary Table. 4.

medial prefrontal cortex, BLA, and CA1 of vHPC (vCA1). Local theta spectral power was strengthened in fear processing regions during fear retrieval in control and MPS groups, implying that neuronal activity reacts from neutral or emotion-paired stimuli. PFC-BLA-vHPC regions' tri-direction consistency also demonstrates a high coherence during contextual and cued retrievals, revealing that conditional stimuli reinforce neuronal interaction. Although the control group's brain activity enhancement identifies potential neuronal responses factors to variated stimuli, related neurons' dominant transmission connectivity characterizes fear processing phases[30,73–77]. For instance, trauma-aroused amygdala activity moderates BLA → PFC connectivity during fear memory consolidation and impairs fear extinction[73]. However, IL PFC → BLA synchronization top-down control drives fear extinction after training[30]. Furthermore, vHPC involvement is necessary for cross-correlation with other nuclei to process fear memory. For example, PFC-vHPC bidirectional conversion bolsters the brain-derived neurotrophic factor (BDNF) to regulate memory extinction[78]. At the same time, enhanced vHPC→PFC connectivity triggers inhibition by activating PFC interneurons and consequently inducing fear relapse[79]. Moreover, vHPC and BLA interactions coordinating fear encoding and retrieval can intervene and counter the innate anxiety effectuated from photoactivated monosynaptic BLA→vCA1 connectivity[76,77,80]. Our Granger causality results within PFC-BLA-vHPC regions expressed increased randomness during the contextual retrieval, while the leading fear extinction direction was apparent in control group, and the recent context recall caused fear preservation. Furthermore, cue retrieval during the remote time point exhibited the enhancement of directional connectivity transmission from BLA to HPC and from vHPC to PFC in the MPS group. This directional enhancement denotes the tremendous MPS intensity required an extended extinction processing period and frequency for extinguishing emotional reactions from neutral tone stimulation during fear retrieval.

Anxiety after trauma is particularly apparant in chronic PTSD. Studies have verified that enhanced PFC, BLA, and vHPC activities control anxiety levels[81,82], corroborating our EPM and OFT results. Additionally, SPS induced anxiety in rodents over several weeks[46,83]. We assessed the anxious behavior following stressor manipulation for 7 weeks to evaluate SPS and MPS group long-term distinctions. Collected EPM and OFT task data displayed similar effects during the seven-week process (Supplementary

Fig. 6). After context memory retrieval, anxiety levels increased in both SPS and MPS groups ($^{###}p < 0.001$: control vs. SPS; $^{***}p < 0.001$: control vs. MPS; Supplementary Fig. 6b, c). However, post-cue retrievals results did not show any significant differences between the control and SPS groups, indicating that SPS cue tones triggered minor anxiety levels ($^*p < 0.05$, $^{**}p < 0.01$: control vs. MPS; $^+p < 0.05$, $^{++}p < 0.01$: SPS vs. MPS; Supplementary Fig. 6b, c). Overall, the MPS protocol consistently exhibited high anxiety and increased PFC-BLA-vHPC neuronal activities.

Along with behavioral tasks, waking theta powers' strengthened electroencephalography (EEG) demonstrated elevated innate anxiety levels after fear retrieval, reporting a prolonged anxiety effect in this MPS rodent model. Our SPS with IFS group findings also displayed higher waking theta oscillation powers following SPS administration and long-term fear memory retrieval ($^\#p < 0.05$: 4-h SD vs. SPS; $^{**}p < 0.01$, $^{***}p < 0.001$: baseline vs. fear memory retrieval days; Supplementary Figs. 7a, 8i–l). These variations substantiate that SPS and MPS inflict long-lasting anxiety. Additionally, sleep alteration coordinates with the fear consolidation efficiency after a traumatic event[84–86], as total sleep deprivation impairs negative memory consolidation and the following behavioral expression[85]. Increasing spindles during NREM sleep facilitates hippocampus-dependent memory consolidation by phase-locking to other associated nuclei[87]. In this study, we reported a compensatory NREM sleep effect after IFS during ZT1-4. Although no difference was observed during the memory retrieval period, 12-15 Hz sleep spindle power facilitation is prevalent after MPS or memory retrieval, indicating spindle oscillation intensity prominence during negative memory consolidation. Similar effects were also detected in the SPS pairing with IFS rats. SWAs significantly increased after SPS manipulation and fear memory retrieval without altering NREM sleep duration ($^*p < 0.05$, $^{**}p < 0.01$, $^{***}p < 0.001$: baseline vs. 4-h SD, MPS-Day1, and SPS; $^\#p < 0.05$, $^{###}p < 0.001$: 4-h SD vs. MPS-Day1 and SPS in Supplementary Figs. 7, 8a; $^{***}p < 0.001$: baseline vs. fear memory retrieval days in Supplementary Fig. 8b–d). Furthermore, REM sleep and its corresponding theta powers are essential for post-shock memory consolidation[88]. In the SPS model, REM sleep increased after stress administration and intensified fear expression during memory recall[41–43]. On the contrary, our data demonstrated immediately suppressed REM sleep after MPS manipulations and fear memory retrieval, while the

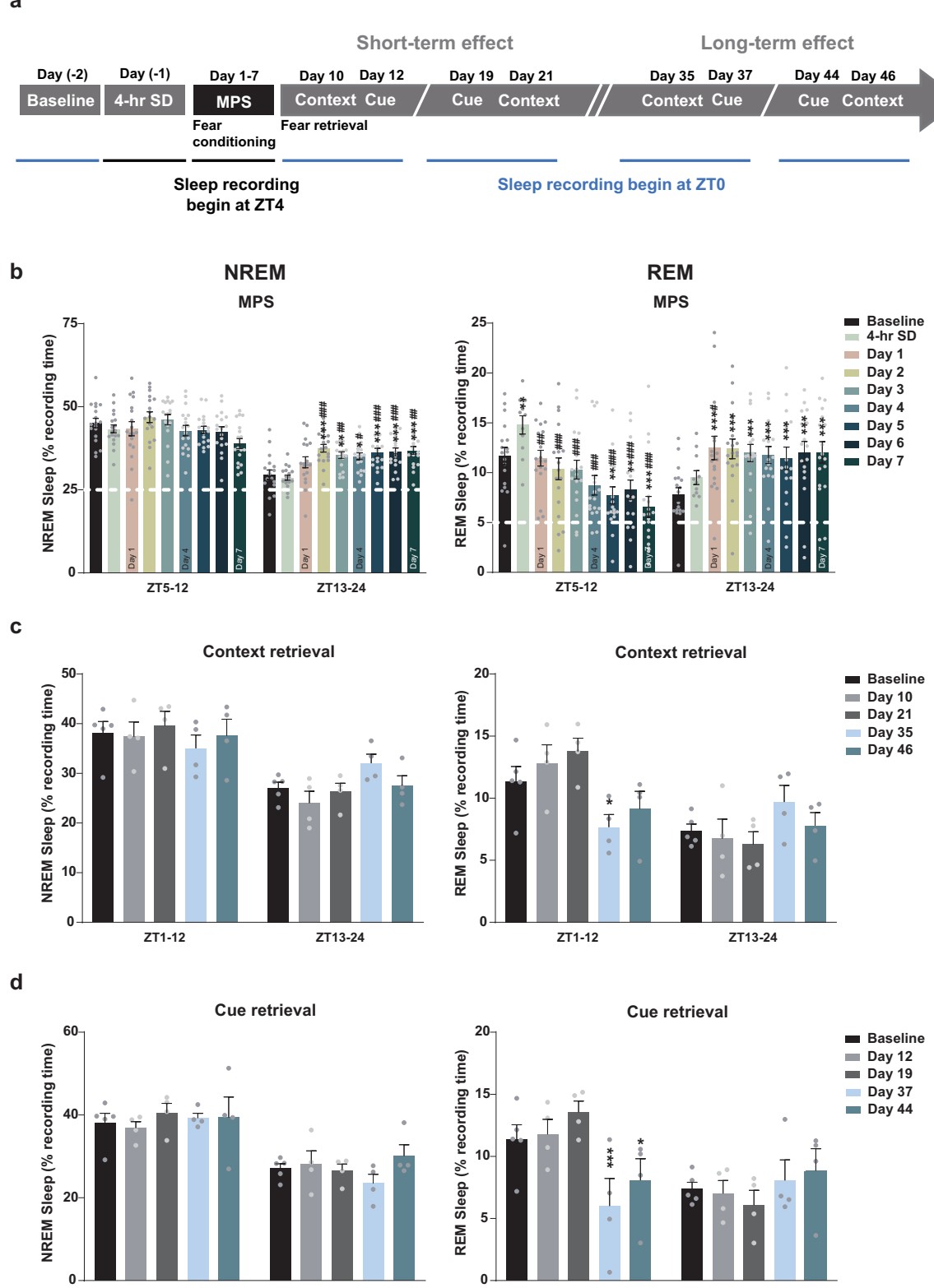

freezing behaviors continued to increase during long-term memory retrieval. The SPS group with IFS also demonstrated a significant depletion in REM sleep after the shock and a compensated REM sleep enhancement during the dark period ($^*p < 0.05$, $^{**}p < 0.01$, $^{***}p < 0.001$: baseline vs. 4-h SD, MPS-Day1, and SPS; $^\#p < 0.05$, $^{\#\#\#}p < 0.001$: 4-h SD vs. MPS-Day1 and SPS; Supplementary Fig. 7a, b). This result indicates that the REM sleep suppression percentage

may be due to electrical footshock stress experienced during this protocol. However, the absence of sleep alteration after SPS manipulation days (Days 2-6) and fear retrieval (contextual and cued) was markedly different from the MPS (Supplementary Fig. 7c–e). These divergent results indicate that increasing SPS repetition prompts longer sleep time disruption in MPS group. Although SPS manipulation did not alter the sleep

**Fig. 6 MPS and fear memory retrieval altered the sleep-wake activity. a** Sleep-wake activities were recorded for 7 weeks beginning at ZT4 and continued for 20 h during the 4-h SD (Day (-1)) and 7-day MPS protocol (**b**), and were acquired from ZT0 and lasted for 24 h during baseline recording (Day (-2)) and after fear memory retrievals (**c**, **d**) (see "Methods"). **b** MPS altered average NREM (left) and REM (right) sleep times (%) during light and dark periods. MPS manipulation during ZT1-4 primarily decreased REM sleep by 8 h (ZT5-12) and compensatively increased NREM sleep during the dark period (ZT13-24). Baseline $n = 18$, 4-h SD $n = 11$, Days 1–7 $n = 18$. One-way ANOVA: $F_{(8,1851)} = 9.840$, $p < 0.0001$ during ZT13-24 NREM sleep; $F_{(8,1231)} = 16.013$, $p < 0.0001$ and $F_{(8,1851)} = 8.350$, $p < 0.0001$ during ZT5-12 and ZT13-24 REM sleep, respectively. Tukey's multiple comparison: $^*p < 0.05$, $^{**}p < 0.01$, $^{***}p < 0.001$: baseline vs. 4-h SD and Days 1–7; $^\#p < 0.05$, $^{\#\#}p < 0.01$, $^{\#\#\#}p < 0.001$: 4-h SD vs. Days 1–7. **c**, **d** Average NREM sleep (left) and REM (right) sleep times (%) were altered after context (**c**) and cue retrieval (**d**). Long-term contextual and cue retrieval primarily affected the REM sleep during the light period (ZT1-12). Baseline $n = 5$ and memory retrieval days $n = 4$. One-way ANOVA: $F_{(4,247)} = 8.275$, $p < 0.0001$ during ZT1-12 REM sleep in (**c**); $F_{(4,247)} = 12.456$, $p < 0.0001$ during ZT1-12 REM sleep in (**d**); Tukey's multiple comparison: $^*p < 0.05$, $^{***}p < 0.001$: baseline vs. fear retrieval days. Values represent the mean ± SEM.

**Table 1 MPS effects on sleep architecture alteration.**

**MPS**

| ZT | Treatment | L:D cycle | Bouts | | | Duration | | | Transition |
|---|---|---|---|---|---|---|---|---|---|
| | | | Wake | NREM | REM | Wake | NREM | REM | |
| 5-12 | Baseline | L | 9.5 ± 0.2 | 14.4 ± 0.3 | 4.9 ± 0.2 | 3.4 ± 0.4 | 1.7 ± 0.03 | 1.4 ± 0.04 | 42.8 ± 1.0 |
| | 4-h SD | | 8.5 ± 0.4 | 13.8 ± 0.4 | 5.1 ± 0.3 | 3.9 ± 0.7 | 1.8 ± 0.1 | 1.7 ± 0.07* | 38.9 ± 1.2 |
| | Day 1 | | 8.5 ± 0.2 | 13.1 ± 0.4 | 4.3 ± 0.2 | 4.2 ± 0.6 | 1.9 ± 0.1 | 1.4 ± 0.1## | 37.6 ± 1.1* |
| | Day 2 | | 7.8 ± 0.3*** | 13.8 ± 0.3 | 4.5 ± 0.2 | 3.4 ± 0.3 | 1.9 ± 0.1 | 1.4 ± 0.1## | 38.6 ± 1.0 |
| | Day 3 | | 8.7 ± 0.3 | 13.2 ± 0.4 | 3.9 ± 0.2 | 4.1 ± 0.6 | 2.0 ± 0.1* | 1.4 ± 0.1## | 36.3 ± 1.0*** |
| | Day 4 | | 8.8 ± 0.2 | 13.7 ± 0.3 | 4.1 ± 0.2 | 4.2 ± 0.5 | 1.7 ± 0.04 | 1.3 ± 0.1### | 39.8 ± 1.0 |
| | Day 5 | | 8.9 ± 0.2 | 13.8 ± 0.4 | 3.6 ± 0.2***### | 3.8 ± 0.3 | 1.8 ± 0.04 | 1.3 ± 0.1### | 39.7 ± 1.0 |
| | Day 6 | | 8.9 ± 0.3 | 14.3 ± 0.4 | 4.0 ± 0.2 | 4.6 ± 0.5 | 1.6 ± 0.03 | 1.2 ± 0.1### | 41.7 ± 1.1 |
| | Day 7 | | 8.6 ± 0.3 | 13.4 ± 0.4 | 3.3 ± 0.2***### | 4.7 ± 0.4 | 1.6 ± 0.04 | 1.2 ± 0.1*### | 38.8 ± 1.0 |
| 13-24 | Baseline | D | 8.2 ± 0.2 | 11.5 ± 0.3 | 4.0 ± 0.2 | 6.4 ± 0.6 | 1.5 ± 0.04 | 1.0 ± 0.03 | 35.6 ± 1.0 |
| | 4-h SD | | 8.1 ± 0.3 | 11.3 ± 0.4 | 3.8 ± 0.3 | 6.5 ± 0.8 | 1.4 ± 0.04 | 1.2 ± 0.1** | 32.7 ± 1.2 |
| | Day 1 | | 7.9 ± 0.2 | 11.8 ± 0.3 | 5.4 ± 0.2***### | 5.9 ± 0.6 | 1.5 ± 0.03 | 1.3 ± 0.03***## | 35.6 ± 1.0 |
| | Day 2 | | 7.3 ± 0.3 | 13.1 ± 0.3* | 5.3 ± 0.2***### | 4.9 ± 0.4 | 1.6 ± 0.02# | 1.2 ± 0.03*** | 37.8 ± 1.0# |
| | Day 3 | | 8.8 ± 0.2 | 12.5 ± 0.3 | 5.0 ± 0.2*# | 5.2 ± 0.6 | 1.5 ± 0.03 | 1.3 ± 0.03*** | 35.5 ± 1.0 |
| | Day 4 | | 8.7 ± 0.2 | 12.7 ± 0.3 | 5.2 ± 0.2***### | 5.7 ± 0.6 | 1.5 ± 0.03 | 1.2 ± 0.03** | 38.0 ± 1.9# |
| | Day 5 | | 9.2 ± 0.2 | 13.7 ± 0.3***### | 5.4 ± 0.2***### | 4.1 ± 0.3* | 1.5 ± 0.03 | 1.2 ± 0.03* | 39.7 ± 1.0### |
| | Day 6 | | 8.4 ± 0.3 | 12.7 ± 0.4 | 5.4 ± 0.2***### | 5.1 ± 0.4 | 1.5 ± 0.03 | 1.1 ± 0.03 | 39.3 ± 1.1### |
| | Day 7 | | 8.9 ± 0.2 | 12.9 ± 0.3 | 5.1 ± 0.2*## | 5.5 ± 0.5 | 1/5 ± 0.03 | 1.2 ± 0.03*** | 39.1 ± 1.0## |

Bout numbers, average duration (min), and stage transition parameters were presented as the mean ± SEM. One-way ANOVA: $F_{(8,1231)} = 6.117$, $p < 0.0001$ and $F_{(8,1231)} = 6.116$, $p < 0.0001$ for ZT5-12 REM bout number and duration; $F_{(8,1851)} = 6.966$, $p < 0.0001$ and $F_{(8,1851)} = 5.727$, $p < 0.0001$ for ZT13-24 REM bout number and duration; $F_{(8,1851)} = 4.765$, $p < 0.0001$ and $F_{(8,1851)} = 1.904$, $p = 0.055$ for ZT13-24 NREM bout number and duration; $F_{(8,1851)} = 4.411$, $p < 0.0001$ for ZT13-24 transitions stages. Baseline $n = 18$, 4-h SD $n = 18$, Days 1–7 $n = 18$. Tukey's multiple comparison: $^*p < 0.05$, $^{**}p < 0.01$, $^{***}p < 0.001$: baseline vs. 4-h SD and baseline vs. MPS protocol (Days 1-7); $^\#p < 0.05$, $^{\#\#}p < 0.01$, $^{\#\#\#}p < 0.001$: 4-h SD vs. MPS protocol (Days 1-7).

**Table 2 Context retrieval effects on sleep architecture alteration.**

**Context retrieval**

| ZT | Treatment | L:D cycle | Bouts | | | Duration | | | Transition |
|---|---|---|---|---|---|---|---|---|---|
| | | | Wake | NREM | REM | Wake | NREM | REM | |
| 1-12 | Baseline | L | 8.8 ± 0.5 | 13.6 ± 0.7 | 4.9 ± 0.4 | 4.4 ± 0.6 | 1.6 ± 0.1 | 1.4 ± 0.1 | 41.8 ± 1.9 |
| | Day 10 | | 9.3 ± 0.5 | 13.0 ± 0.7 | 5.2 ± 0.5 | 4.1 ± 0.7 | 1.6 ± 0.1 | 1.3 ± 0.1 | 42.6 ± 2.3 |
| | Day 21 | | 8.6 ± 0.4 | 12.7 ± 0.5 | 4.9 ± 0.4 | 3.7 ± 0.5 | 1.8 ± 0.1 | 1.7 ± 0.1 | 42.1 ± 1.7 |
| | Day 35 | | 9.2 ± 0.6 | 12.9 ± 0.8 | 3.3 ± 0.3* | 5.5 ± 1.1 | 1.5 ± 0.1 | 1.3 ± 0.1 | 42.9 ± 2.2 |
| | Day 46 | | 9.9 ± 0.5 | 13.8 ± 0.6 | 3.7 ± 0.3 | 3.9 ± 0.6 | 1.5 ± 0.1 | 1.4 ± 0.1 | 46.1 ± 1.8 |
| 13-24 | Baseline | D | 7.5 ± 0.5 | 10.5 ± 0.8 | 3.5 ± 0.4 | 9.8 ± 1.9 | 1.4 ± 0.1 | 1.0 ± 0.1 | 33.3 ± 2.1 |
| | Day 10 | | 6.5 ± 0.5 | 8.9 ± 0.7 | 3.3 ± 0.4 | 10.2 ± 1.8 | 1.5 ± 0.1 | 1.0 ± 0.1 | 31.3 ± 2.4 |
| | Day 21 | | 6.5 ± 0.4 | 8.8 ± 0.6 | 3.0 ± 0.4 | 8.5 ± 1.4 | 1.7 ± 0.1** | 1.0 ± 0.1 | 30.4 ± 1.9 |
| | Day 35 | | 8.0 ± 0.4 | 11.2 ± 0.5 | 4.2 ± 0.4 | 4.8 ± 0.5 | 1.5 ± 0.1 | 1.2 ± 0.1 | 40.3 ± 1.8 |
| | Day 46 | | 7.5 ± 0.5 | 10.4 ± 0.7 | 3.6 ± 0.4 | 8.2 ± 1.7 | 1.5 ± 0.1 | 1.1 ± 0.1 | 33.5 ± 2.1 |

Bout numbers, average duration (min), and stage transition parameters were presented as the mean ± SEM. Baseline $n = 5$ and memory retrieval days $n = 4$. One-way ANOVA: $F_{(4,247)} = 5.004$, $p = 0.001$ for ZT1-12 REM bout number; $F_{(4,247)} = 3.207$, $p = 0.014$ for ZT13-24 NREM bout duration. Tukey's multiple comparison: $^*p < 0.05$, $^{**}p < 0.01$: baseline vs. context retrieval days.

percentage post-SPS and fear retrieval, theta oscillation power during REM sleep was elevated, analogous with the MPS group ($^{***}p < 0.001$: baseline vs. SPS; $^\#p < 0.05$: 4-h SD vs. SPS and $^{**}p < 0.01$, $^{***}p < 0.001$: baseline vs. fear memory retrieval days; Supplementary Fig. 8e–h). This result suggests that REM sleep quality, represented as theta power, is more influential in consolidating emotional memory than REM sleep quantity. Theta activity during REM sleep increased during MPS and long-term memory retrieval periods, and is crucial for maintaining fear memory. To verify fear memory persistence in stress events, we

**Table 3 Cue retrieval effects on sleep architecture alteration.**

**Cue retrieval**

| ZT | Treatment | L:D cycle | Bouts | | | Duration | | | Transition |
|---|---|---|---|---|---|---|---|---|---|
| | | | **Wake** | **NREM** | **REM** | **Wake** | **NREM** | **REM** | |
| 1-12 | Baseline | L | 8.8 ± 0.5 | 13.6 ± 0.7 | 4.9 ± 0.4 | 4.4 ± 0.6 | 1.6 ± 0.1 | 1.4 ± 0.1 | 41.8 ± 1.9 |
| | Day 12 | | 9.2 ± 0.5 | 13.3 ± 0.7 | 4.5 ± 0.3 | 4.8 ± 1.0 | 1.6 ± 0.1 | 1.5 ± 0.1 | 45.4 ± 2.1 |
| | Day 19 | | 10.1 ± 0.4 | 14.9 ± 0.6 | 5.4 ± 0.4 | 2.9 ± 0.3 | 1.5 ± 0.1 | 1.5 ± 0.1 | 45.8 ± 2.0 |
| | Day 37 | | 10.1 ± 0.5 | 14.1 ± 0.7 | 2.7 ± 0.4*** | 3.9 ± 0.6 | 1.6 ± 0.1 | 1.1 ± 0.1** | 42.3 ± 1.9 |
| | Day 44 | | 9.2 ± 0.6 | 13.8 ± 0.8 | 3.4 ± 0.3* | 5.8 ± 1.6 | 1.6 ± 0.1 | 1.4 ± 0.1 | 44.7 ± 2.2 |
| 13-24 | Baseline | D | 7.5 ± 0.5 | 10.5 ± 0.8 | 3.5 ± 0.4 | 9.8 ± 1.9 | 1.4 ± 0.1 | 1.0 ± 0.1 | 33.3 ± 2.1 |
| | Day 12 | | 5.3 ± 0.5* | 8.1 ± 0.8 | 2.8 ± 0.5 | 11.7 ± 2.1 | 1.3 ± 0.1 | 0.8 ± 0.1 | 29.1 ± 2.5 |
| | Day 19 | | 6.7 ± 0.4 | 10.1 ± 0.6 | 3.1 ± 0.4 | 6.8 ± 0.7 | 1.5 ± 0.1 | 1.0 ± 0.1 | 33.4 ± 1.7 |
| | Day 37 | | 6.9 ± 0.6 | 9.4 ± 0.8 | 3.9 ± 0.5 | 11.1 ± 2.1 | 1.4 ± 0.1 | 1.0 ± 0.1 | 30.0 ± 2.4 |
| | Day 44 | | 7.4 ± 0.5 | 10.8 ± 0.6 | 4.2 ± 0.5 | 7.6 ± 1.7 | 1.5 ± 0.1 | 1.0 ± 0.1 | 36.6 ± 2.0 |

Bout numbers, average duration (min), and stage transition parameters were presented as the mean ± SEM. Baseline $n = 5$ and memory retrieval days $n = 4$. One-way ANOVA: $F_{(4,247)} = 8.769$, $p < 0.0001$ and $F_{(4,247)} = 5.094$, $p = 0.001$ for ZT1-12 REM bout number and duration. Tukey's multiple comparison: $^{*}p < 0.05$, $^{**}p < 0.01$, $^{***}p < 0.001$: baseline vs. cue retrieval days.

assessed corticosterone levels at the end of the 7-week procedure as it referenced HPA axis activation. Seven days after stimuli, the SPS model confirmed that glucocorticoid negative feedback from the HPA axis is related to the increased glucocorticoid receptor (GR) and decreased mineralocorticoid receptor (MR) at the hippocampus[23,89]. However, our data indicated significantly increased corticosterone levels after fear retrieval, distinguishing how contextual and cued retrieval affect stress response.

Additionally, our MPS model produced prolonged and comprehensive abnormalities in behavior expression, neuronal synchronization, and the neuroendocrine system; nonetheless, the extinguished fear behavior was not shown during this 7-week observation period. Therefore, our MPS model's fear extinction demands further discussion on the observation period or desensitization training frequency. Furthermore, the considerable basal neuronal oscillation power variations in the PFC, BLA, and vHPC regions between control and MPS group were discovered in short-term memory retrieval (Days 21 and 19; Fig. 2d, f), and was specifically stronger at PFC region throughout the long-term period (Fig. 2d, f). One possible reason might be that the stress, caused by experiment manipulations (e.g., changing rats from animal housing room to insulation LFP recording space and electrodes implantation), increases the control rat's oscillation power during short-term memory retrievals. Another possibility is that PFC basal oscillation variation may result from the nuclei's functional specificity. The PFC activation is crucial for limbic system connection and processes the safe signal[90]. Our results exhibited that control rats enhanced their theta power intensity, specifically in the medial prefrontal cortex (mPFC) region, and its coherency from mPFC to BLA and vHPC transiting the safety direction, which may refer to the continuous dominancy of PFC when environment shifting. Moreover, other research has also indicated that a population of PFC neurons is less responsive when repeated exposure to similar stimulation that caused a slow adaptive effect[91], which multiple times of context or cue memory recall may reduce the reaction of PFC in the MPS rats and resulted in signal discrepancies contrast to the control group. Thus, to assess fear memory recall-provoked neuronal reactions, we compared pre- and post-retrieving (home vs. context and pre-cue vs. post-cue) theta powers, each group's connectivity, and control and MPS group's subtractive variations. Although a significant neuronal activation during remote memory retrieval after MPS manipulation was observed, context and cue tone stimuli also induced a control group response. These results suggest that environmental change and auditory-visual flash-boosted neuroexcitation effects could not be excluded during memory retrieval; therefore, a control group comparison was required. Furthermore,

increasing surgery consistency and electrodes stability throughout this process is imperative for steady signal collection in future experiments. Additionally, neuronal interaction within the MPS model's fear-related brain regions only partially recognized negative memory processing immediate reactions. Hence, future studies are needed to clarify underlying synaptic transmissions within PFC-BLA-vHPC circuitries and neurotransmitter variation mechanisms post-MPS manipulation. Nevertheless, this study established neuronal activity alterations in fear-encoding brain regions and dominant pathways signal transitions mediating the fear memory processing in the remote time point.

Our findings suggest the MPS manipulation prolongs PTSD-like symptoms by 6 weeks, including fear memory flashback, neuroendocrine system abnormality, enhanced fear-related regions interaction, increased anxiety, and sleep alteration. In conclusion, this research provides a long-term model for simulating chronic PTSD syndromes.

## Methods

**Subjects**. Male Sprague-Dawley rats (300-400 g; BioLASCO, Taiwan) were used in this study. Each animal was hosted in a single $28 \times 22 \times 40$ $cm^3$ cage with the controlled 12 h of light and 12 h of dark circadian rhythms, $22.0 \pm 2$ °C room temperature, and $50 \pm 5\%$ humidity. Food and drinking water were available *ad libitum*. Experiments and health care adhered to all protocols set by the instructions of Institutional Animal Care and Use Committee (IACUC) of National Taiwan University (Approval number: NTU104-EL-00101).

**Stereotaxic surgery**. Rats were anesthetized through Zoletil 50 (47 mg/kg; Virbac, Carros, France) mixed with xylazine (6.8 mg/kg; Sigma-Aldrich, St Louis, MO, USA) intraperitoneal injections (i.p). To record sleep-wake activity, three EEG electrodes (E363/20; Plastics One, Roanoke, VA, USA) were implanted on the frontal cortex (AP, +2 mm; ML, +2.5 mm coordinates from bregma), parietal cortex (AP, -5.5 mm; ML, +3 mm coordinates from bregma), and cerebellum (AP, -11 mm; ML, 0 mm coordinates from bregma). Two electromyogram (EMG) electrodes (E363/76/47 mm; Plastics One) were subcutaneously embedded in the neck. To stabilize skull electrodes, we fixed two anchor screws on opposite hemispheres relative to those EEG screws. Then, all the wires were collected into a pedestal (E363/20; Plastics One) and cemented onto the skull with dental acrylic (GC Corporation, Tokyo, JP).

For LFP recording, six electrodes were surgically implanted into three specific brain regions with two ground wires entangled on anchor screws. Each brain region was implanted with two electrodes clustered in one guiding cannula (26 gauge, O.D.: 0.46 mm, I.D.: 0.24 mm; Plastics One): the PFC (AP, +3.2 mm; ML, +0.5 mm; DV, -3.2 mm coordinates from bregma), BLA (AP, -2.8 mm; ML, +5 mm; DV, -8.2 mm coordinates from bregma), and vHPC (AP, -5.6 mm; ML, +4.8 mm; DV, -3.4 mm coordinates from bregma). Two anchor screws were fixed on the frontal (AP, +2 mm; ML, +2.5 mm coordinates from bregma) and parietal cortices (AP, -5.5 mm; ML, +3 mm coordinates from bregma). Then, electrodes were collected into the microwire connector and stabilized with dental acrylic onto the skull. After surgery, penicillin (5000 IU; Sigma-Aldrich) was i.p injected to prevent infection, and 0.15 mg/ml of ibuprofen (YUNGSHIN PHARM, TXG,

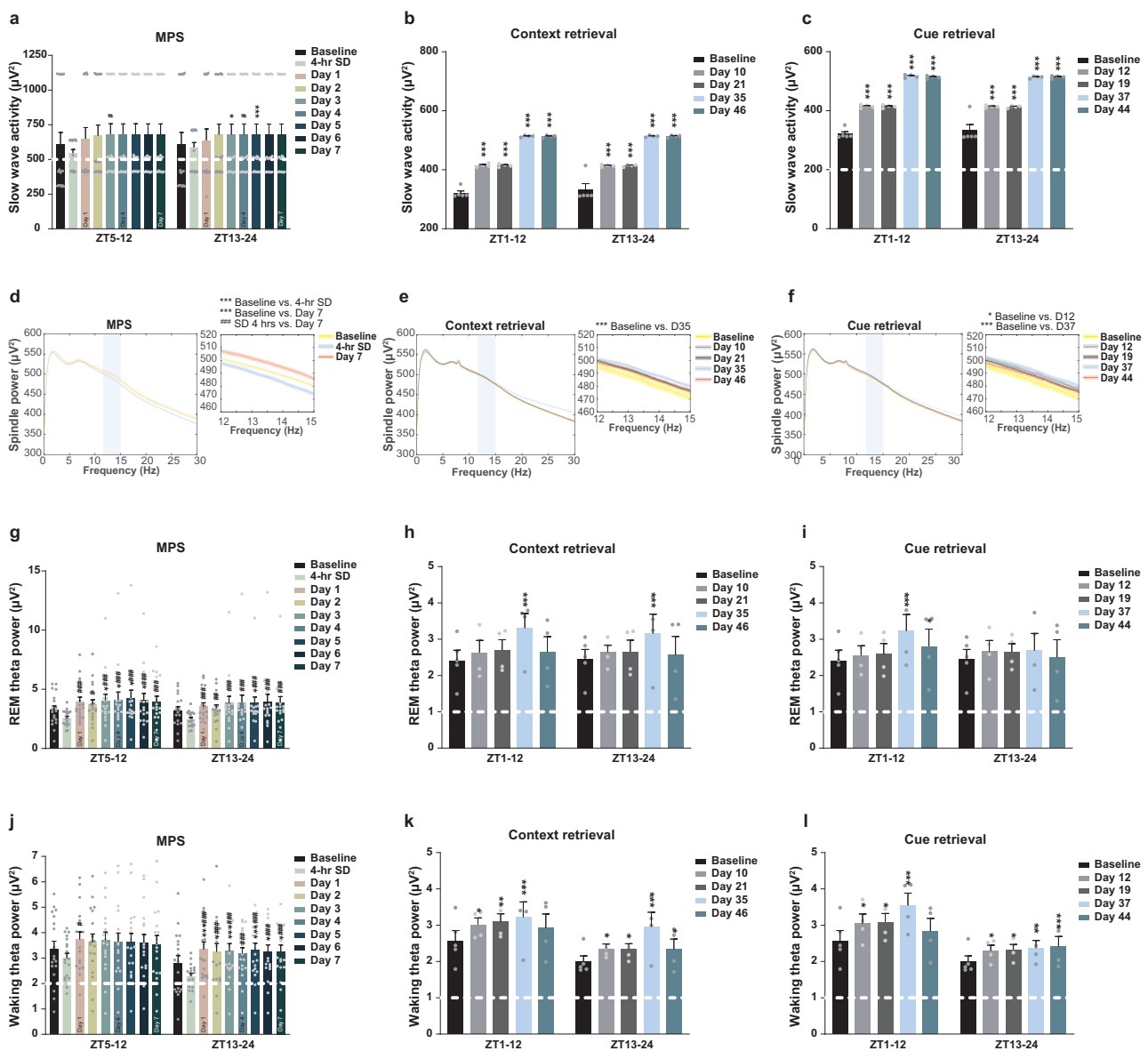

Taiwan) was provided in drinking water for 1 week to relieve pain. All the experiments began 1 week after surgery recovery.

**Multiple prolonged stress model**. Our MPS process consists of four different stressors manipulated over 7 days, with daily 4-h persistence after light onset. Rats randomly received a 2-h restraint, 20 min of forced swimming, 12 IFS within 10 min, and short-term isoflurane anesthetization for seven consecutive days (Fig. 1a). During restraint, rats were trapped the in a transparent piping plastic bag with small, poked air holes to prevent electrode damage from the restrainer and increase the breathability. Forced swimming was conducted in a 20 cm diameter cylinder with a 50 cm depth in 26-28 °C water. Rats were blow-dried before the following protocol implementation. In addition to being designed the primary stressor to trigger PTSD-like behavior, the IFS was linked with particular forward fear conditioning context and cues to ensure fear memory manifestation throughout the experimental phase. Four IFS procedures were implemented on odd days during the MPS process. IFS was conducted in a 28 ×22 x 40 cm³ footshock box with a stainless-steel rods board on the floor and 2% vinegar at the bottom of the box (background brightness: 3.5 lux) for conditioned contextual stimulation. Additionally, to pair the electric shocks with compounded light and sound cue tones as conditioned stimulations, we positioned the light source (LED light bulb, white light, 1200 lm, 10 W; Everlight Electronics Co., Taipei, Taiwan), and sound source (SRS-XB13 speaker; Sony, Tokyo, Japan) 80 cm right above and directly above the footshock box, respectively. During IFS, a 1-s light (34 lux) and sound (2,000 Hz, 80 dB) compounded cue was delivered immediately before every electric stimulus as conditioned stimuli for further fear memory retrieval tasks. Twelve

electrical stimulations (5 mA, 133 V, 0.5 s for each stimulus; World Precision Instruments, FL, USA) were randomly distributed over 10 min. Each shock was administered at a random time point, and all the interstimulus intervals were uncertain in each IFS implementation process (all rats experienced the same IFS treatment on the same day) as an unconditioned stimulus. Control group rats were well-habituated through gently handling using a towel for 10 min during the first seven-day procedure without MPS manipulation. To compare the immobility with the MPS group, the control group was placed with the same footshock box settings with 2% vinegar and 12 compounded light-sound cue tones (without IFS) for 10 min on odd days during the seven-day treatment (four times total).

**Single-prolonged stress model with IFS**. Rats were subjected to SPS protocols with IFS stimuli to compare MPS and SPS long-lasting effects. SPS combined with IFS was a one-time stimulation that lasted 4 h (beginning from light onset). Manipulation mirrored the third day of the MPS procedure, which included 2-h of restraint, 20-min forced swimming, 10-min 12 IFS stimuli (5 mA, 133 V, 0.5 s for each stimulus), and isoflurane anesthetization 15 min later. As a context environment, IFS was conducted in a 28 × 22 × 40 cm³ footshock box with a stainless-steel rods board on the floor and 2% of vinegar at the bottom of the box (background brightness: 3.5 lux). Each shock was paired with a 1-s pre-performed auditory (2000 Hz, 80 dB, 1 s) and light (34 lux, 1 s) cues as condition stimuli for following persisted fear memory assessment.

**Fear memory retrieval**. Contextual and cue fear memory retrievals were detected in the MPS, SPS, and a seven-day habituated (control) groups. Contextual retrieval-

**Fig. 7 MPS and fear memory retrieval facilitated SWAs, spindle power, REM sleep, and waking theta power. a-c** Average SWAs during the light and dark periods were obtained during NREM sleep. SWAs were enhanced after MPS (**a**), short- and long-term fear memory retrievals (**b**, **c**). One-way ANOVA: $F_{(8,1228)} = 2.659$, $p < 0.05$ and $F_{(8,1834)} = 2.668$, $p < 0.001$ during ZT5-12 and ZT13-24 in (**a**); $F_{(4,247)} = 2019.891$, $p < 0.001$ and $F_{(4,244)} = 806.013$, $p < 0.0001$ during ZT1-12 and ZT13-24 in (**b**); $F_{(4,245)} = 1978.855$, $p < 0.0001$ and $F_{(4,239)} = 788.768$, $p < 0.0001$ during ZT1-12 and ZT13-24 in (**c**). Tukey's multiple comparison: $^*p < 0.05$, $^{***}p < 0.001$: baseline vs. 4-h SD, baseline vs. MPS protocol (Days 1-7), and baseline vs. fear memory retrieval days; $^\#p < 0.05$: 4-h SD vs. MPS protocol (Days 1-7). **d-f** 0–30 Hz spectral power averages from the NREM sleep stage during 20 h after MPS (**d**), 24 h after context retrieval (**e**), and cue retrieval (**f**). Gray block: 12-15 Hz segment sleep spindle frequency. Sleep spindle power was intensified by MPS protocol (**d**) and fear memory retrievals (**e**, **f**). One-way ANOVA: $F_{(2,579)} = 135.733$, $p < 0.0001$ in (**d**); $F_{(4,7715)} = 21.109$, $p < 0.011$ in (**e**); $F_{(4,7715)} = 34.199$, $p < 0.017$ in (**f**). Tukey's multiple comparison: $^*p < 0.05$, $^{***}p < 0.001$: baseline vs. 4-h SD, baseline vs. MPS protocol (Days 1-7), and baseline vs. fear memory retrieval days; $^{\#\#\#}p < 0.001$: 4-h SD vs. MPS protocol (Days 1-7). **g-i** Average REM sleep theta powers were collected after MPS protocol (**g**), context retrieval (**h**), and cue retrieval (**i**) periods. MPS increased theta intensity during REM sleep (**g**), long-term context retrieval (**h**), and long-term cue retrieval (**i**). One-way ANOVA: $F_{(8,1113)} = 7.261$, $p < 0.0001$ and $F_{(8,1711)} = 7.108$, $p < 0.0001$ during ZT5-12 and ZT13-24 in (**g**); $F_{(4,235)} = 9.718$, $p < 0.0001$ and $F_{(4,206)} = 5.918$, $p < 0.0001$ during ZT1-12 and ZT13-24 in (**h**); $F_{(4,231)} = 8.847$, $p < 0.0001$ and $F_{(4,193)} = 1.159$, $p = 0.330$ during ZT1-12 and ZT13-24 in (**i**). Tukey's multiple comparison: $^*p < 0.05$, $^{***}p < 0.001$: baseline vs. 4-h SD, baseline vs. MPS protocol (Days 1-7), and baseline vs. fear memory retrieval days; $^\#p < 0.05$, $^{\#\#}p < 0.01$, $^{\#\#\#}p < 0.001$: 4-h SD vs. MPS protocol (Days 1-7). **j-l** MPS and fear memory retrievals elevated theta activity during wakefulness. **j** Average theta powers enhanced during wakefulness (the active period) post-MPS administration. One-way ANOVA: $F_{(8,1231)} = 2.440$, $p < 0.0001$ and $F_{(8,1850)} = 10.246$, $p < 0.0001$ during ZT5-12 and ZT13-24. Tukey's multiple comparison: $^*p < 0.05$, $^{**}p < 0.01$: baseline vs. 4-h SD and baseline vs. MPS protocol (Days 1-7); $^\#p < 0.05$, $^{\#\#\#}p < 0.001$: 4-h SD vs. MPS protocol (Days 1-7). **k**, **l** Average theta powers during wakefulness post-fear memory retrievals. Context retrieval (**k**) and cue retrieval (**l**) increased theta power intensities for 24 h during short- and long-term periods. One-way ANOVA: $F_{(4,247)} = 5.540$, $p < 0.0001$ during ZT1-12 and $F_{(4,246)} = 16.054$, $p < 0.0001$ during ZT13-24 in (**k**), and $F_{(4,247)} = 9.975$, $p < 0.0001$ during ZT1-12 and $F_{(4,243)} = 5.077$, $p < 0.0001$ during ZT13-24 in (**l**). Tukey's multiple comparison: $^*p < 0.05$, $^{**}p < 0.01$, $^{***}p < 0.001$: baseline vs. MPS protocol (Days 1-7). Baseline $n = 18$, 4-h SD $n = 11$, Days 1–7 $n = 18$ in (**a**), (**d**), (**g**), and (**j**); baseline $n = 5$ and memory retrieval days $n = 4$ in (**b**), (**c**), (**e**), (**f**), (**h**), (**i**), (**k**), and (**l**). Values represent the mean ± SEM.

induced freezing behavior was determined when rats re-entered the IFS box (with stainless-steel rods board and 2% vinegar) for 10 min. In addition, we validated the rats' immobility behavior in their home cages during cue retrieval to exclude novel context effects. Twelve compounded light and sound cues were randomly generated over 10 min. The same IFS lighting and audio equipment conditions were used in the cue retrieval test; thus, we placed a light bulb directly above the animal's home cage to provide the same brightness level (34 lux). We also positioned a speaker atop the home cage to produce the same sound cue volume (2000 Hz, 80 dB) to recall the fear memory. Moreover, all rats needed to be awake during both fear retrieval processes to prevent sleep-caused deviation from affecting immobility calculation.

**Experimental protocols.** In Experiment 1, the behavioral alterations post-MPS were assessed through fear retrieval, EPM, and OFT (Figs. 1b, 5a). All the manipulations were initiated at light-onset and continued for 3 h. During the short-term procedure, context retrieval was administered on Days 10 and 21, and cued retrieval was performed on Days 12 and 19. Anxiety level after different retrieval arrangements was immediately assessed through EPM on Days 10 and 19, and OFT on Days 12 and 21. Each behavior task was segmented at a seven-day interval to prevent rats from acclimating to the environment. The long-term procedure was examined 4 weeks post-MPS stimulation, with context retrieval on Days 35 and 46 and cued retrieval on Days 37 and 44. Long-term innate anxiety was determined through EPM on Days 35 and 44 and OFT on Days 37 and 46. Lastly, the rats were euthanized 5 min after receiving context and cued retrievals simultaneously on Day 50.

In Experiment 2, the LFP in the emotionally conductive brain regions during memory retrieval and the behavior tasks were recorded to further confirm negative emotion persistence (Figs. 2a, 3a, 5b). Fear memory retrieval and anxiety detection task manipulation procedures were the same as in Experimental 1.

Due to sleep deprivation defect caused by behavioral tasks in Experiment 1, Experiment 3 focused on sleep-wake activity variation without interruption during the light period. To compare typical and MPS-treated sleep-wake activity variations and exclude the difference between groups, we recorded a 24-h baseline (without disturbance) and 4-h SD 2 days before MPS manipulation as the control and sham control, respectively (Fig. 6a). The same rat group was subjected to the MPS protocol for 7 days. After 4 h of treatment, each day's sleep-wake activities were recorded. Fear memory retrieval was implemented during the last 30 min of the dark period. Sleep-wake activities were recorded for 24 h beginning at light-onset (Fig. 6a). Thus, the retrieval processes were identical to Experiment 1 to analyze sleep disruption after fear memory retrieval.

**Single-prolonged stress (SPS) with IFS experimental protocol.** Experimental protocols for evaluating immobility and behavior tasks during fear memory retrieval mirrored Experiment 1.

To evaluate regular and SPS-treated sleep-wake activity differences, we recorded a 24-h baseline and 4-h sleep deprivation (4-h SD) 2 days before SPS manipulation

as the control and sham control, respectively. Sleep-wake activity alterations were recorded post-SPS procedure on Day 1 (starting 4-h after light-onset). The 24-h recordings were acquired from Days 2, 4, and 6 to observe the sleep variations. Protocol for sleep recording manipulation during fear memory retrieval matched those in Experiment 3.

**Sleep-wake activity analysis.** Sleep-wake activities were recorded and visually analyzed using the custom ICELUS software (M. R. Opp) and written in LabView (National Instruments). The recorded signal gained was 10,000 with the 0.1–40 Hz EEG and 13–10,000 Hz EMG filtered from the bandpass amplifier model V75-01 (Coulbourn Instruments, MA, USA), then collected at a 128 Hz sampling rate and visually scored for 12 s per epoch.

Wakefulness, NREM, and REM sleep vigilance states were differentiated by comparing EEGs, EMGs, delta (0.5–4.0 Hz), and theta (6.0–9.0 Hz) waves alterations. During wakefulness, high-frequency and low amplitude EEG waves and high EMG activities were reported. In contrast, NREM sleep was dominant with the low-frequency and high amplitude delta waves and low-EMG signals. In addition, during REM sleep, hippocampal theta oscillation was dominant, exhibiting the lowest muscle tone. Sleep-wake activity quantities were determined by quantifying percentages from three stages in each hour. Furthermore, bout number, bout duration (min), and stages transitions were also statistically collected to evaluate sleep architecture alterations.

Sleep quality was estimated through SWAs and calculated from NREM sleep delta powers. Theta powers were collected from wakefulness and REM sleep through ICELUS and assessed for anxiety levels and consolidation ability, respectively. Moreover, we collected the frequency 12–15 Hz spindle powers from 0–30 Hz EEGs during 24-h NREM sleep to classify stress influence on rhythmic enhancement in fear memory consolidation. We then analyzed the gathered data with MatLab (MathWorks, MA, USA) Chronux algorithms. Spectral power parameters were a time-bandwidth product of 3 with 5 tapers and a 0–30 Hz or 12–15 Hz frequency band.

**Immobility behavior analysis.** To assess the fear memory persistence post-MPS, we measured rats' immobility duration (freezing behavior accumulation) during IFS, context, and cue retrieval processes. Recorded behavior was manually analyzed to prevent a variant background brightness bias during the experiment, potentially affecting the commercial software accuracy, which analyzed the dynamic contrast between background and animal. Additionally, the freezing behavior was blind and scored by a single observer viewing the recorded video at a 30 Hz framing rate to avoid the artificial analysis variation between different experimenters. Freezing behavior criteria refers to the threshold that the rats performed absence of the body movements, such as whisker and nose movements (excluding necessary respiration) persisting for at least 3 s or more, which would then be scored as accumulated for immobilization duration[92,93]. Data regarding sleeping behavior (rat presents with eyes closed and motionless for over 12 s) would be excluded to prevent resting deviation. Then, each rat's accumulated immobilized time was divided by the 10-min

assessing period to calculate the immobility percentage.

$$Immobility(\%) = \left(\frac{freezing\ time}{total\ time}\right) \times 100 \qquad (1)$$

**Behavior tasks**. The EPM is a $112.5 \times 112.5 \times 100\ cm^3$ black background cross-arm maze in which two of the opposite arms were covered with 40 cm high enclosed walls to construct an avoidance area. During the task, rats were placed in the center zone, then allowed to explore for 5 min. Recorded behavior was analyzed through EthoVision XT (Noldus, WUR, Netherlands) to assess the entry frequency and duration (sec) in the open and closed arms. Alteration percentage were calculated by:

$$Alteration(\%) = \left(\frac{open\ arm}{open\ arm + closed\ arm + center\ zone}\right) \times 100 \qquad (2)$$

OFT involved a $100 \times 100 \times 50\ cm^3$ black background square box with a $50 \times 50\ cm^2$ inner zone. Rats were placed in the inner zone and allowed to explore the environment for 10 min. Recorded videos were analyzed through EthoVision XT to determine inner and outer zone entry frequency and duration (sec). Alteration percentages were calculated by:

$$Alteration(\%) = \left(\frac{inner\ zone}{inner + outer\ zones}\right) \times 100 \qquad (3)$$

**Local field potentiation (LFP) analysis**. LFP oscillations were obtained through the OmniPlex System (Plexon, TX, USA). Signals were amplified with a 10,000 gain with the PBX Preamplifier (Plexon), conversed through a DigiAmp A/D device (Plexon) with a 2000 Hz sampling rate, and recorded with OmniPlex Software (Plexon). Finally, LFP waves were analyzed with the Chronux tool package in MatLab. We extracted 4-7 Hz theta oscillations from all the recorded data to analyze the spectral powers, neuronal coherence, and G.C. LFPs were acquired for 10 min from both the home cage and context retrieval period for context retrieval. Next, they were sectioned into ten 1-min time frames to exclude the artifact signal when comparing mean values. Cue retrieval results were compared using average values between 1-s pre- and post-cue tone onset (3 s each for power spectrograms). We excluded the time section with noise to prevent recorded data bias and subtracted the results with the mean value in each process to calculate variation. Context and cue retrieval used a 0.5 ms moving window (0.05 ms overlap), time-bandwidth of 3 with 5 tapers, 4-7 Hz frequency band, and Jackknife error bar parameters to estimate the spectral power and spectrograms, respectively. Cross-spectrum coherence within PFC-BLA, PFC-vHPC, and BLA-vHPC circuitries were calculated using the coherencyc (Chronux) function with the same parameters as the spectral power during fear retrieval. The variation of Δ power spectrum and coherency in contextual retrieval was calculated by:

$$\Delta\ Power\ Variation = Context - Home\ Cage \qquad (4)$$

The subtracted value between the control and MPS groups in cue retrieval was compared and calculated by:

$$Subtracted\ Value = (1s\ post\ cue) - (1s\ pre\ cue) \qquad (5)$$

The G.C. analyzed bidirectional connectivity between the PFC-BLA, PFC-vHPC, and BLA-vHPC in the open-source MatLab toolbox. Extracted values were calculated from the home cage and the context retrieval within the 10-min domain and 1-s pre- and post-cue tones during cue retrieval. We also compared directional intensity differences between forward and reversed transitions in paired brain regions. In behavioral tasks, each brain region's power intensity was plotted in accordance with traveling location and equal time points recorded from the EthoVision XT. Moreover, we determined mean theta powers in different task exploration zones to clarify behavior and neuronal activity variation. All the analyzed data were arranged with Adobe Illustrator (Adobe, CA, US).

**Enzyme-linked immunosorbent assay (ELISA)**. Blood samples were collected 5 min after fear memory retrieval and remained at room temperature for 10 min to clot. Then, the serum was centrifuged at 13,000 rpm for 20 min at 4 °C. All the experimental protocols followed the instructions included with the commercial ELISA kit (Enzo Life Sciences, NY, USA) with a 27 pg/ml sensitivity (range between 32 and 20,000 pg/ml). To land within the standard sensitivity, serum samples were 1:100 diluted with the sample dilution buffer provided by ELISA kit until the concentration was between 4000 and 10,000 pg/ml. The diluted sample was processed following the kit's instruction and quantified by a microplate reader (Multiskan EX, Thermo Electron Corp., Waltham, MA, USA) to determine corticosterone concentrations.

**Experimental schematic creation**. All the experimental schematics in the article (Figs. 1, 2a–c, 3a, 4a, 4b, 5a, 5b, 6a, Supplementary Figs. 5a, 6a, 7a) were original created through Microsoft PowerPoint (Microsoft, WA, USA) and Adobe Illustrator (Adobe). The light bulb icon in Figs. 1a, 1c, 1e, 1f, 2c and Supplementary Fig. 6a was made by *Vectors Market* from www.flaticon.com.

**Statistics and reproducibility**. All the statistical data are presented as the mean ± Standard Error of Means (SEMs) and analyzed by SPSS (IBM, New York, USA). The group allocation in this study was not randomized. We controlled the covariates in the experimental protocol by comparing the differences between treatment and control groups and comparing the pre-treated and post-treated effects in the same group. In fear memory retrieval, behavioral tasks, and sleep-wake activity analysis results, we used at least two repeated trails to verify the same MPS protocol, and the reproduced results presented the same trend that all replication attempts were successful.

The one-way analysis of variance (ANOVA) with Bonferroni pairwise post hoc comparison was used in first day of immobility (%) analyzed between control, SPS and MPS-Day 1 groups (Fig. 1c). The generalized estimating equations (GEEs) with exchangeable correlation structure and Bonferroni pairwise comparison was applied to estimate other immobility behavior (including group x time factors) owing to unequal data collections within groups (Fig. 1c–e). Recorded LFP-measured brain activity data and Granger causality (G.C.) were analyzed through two-way repeated measure ANOVA to include the between groups and pre- and post-treated fear memory retrieval difference factors throughout the retrieval process (Figs. 2–4). The two-way repeated measure ANOVA and Bonferroni pairwise post hoc comparison were applied to analyze the group and time differences in behavior tasks (EPM and OFT; Fig. 5c, e). The student's unpaired t-test was used to compare control and MPS group corticosterone level differences and theta power variation during behavior tasks (Figs. 1f, 5d, f). The one-way ANOVA was used to assess the sleep-wake alterations, followed by Tukey's comparison post hoc analysis (Figs. 6, 7). The α level of $p < 0.05$ refers to significant differences between groups.

In supplementary data, pre- and post-treated fear memory retrieval power and coherency spectrum differences were compared using the paired t-test (home cage vs. context in contextual recall and pre-cue vs. post-cue in cued recall; Supplementary Figs. 1, 3). To compare Δ power spectrum and coherency variations between the control and MPS groups, the two-way repeated measure ANOVA and Bonferroni post hoc comparison were used to include group and time difference factors (Supplementary Fig. 2). The GEEs with exchangeable correlation structure and Bonferroni pairwise comparison was applied to estimate the immobility (%) between repeated and single retrieval process manipulation groups (Supplementary Fig. 5). The two-way repeated measure ANOVA with Bonferroni post hoc analysis evaluated group and time differences in the behavior tasks (EPM and OFT; Supplementary Fig. 6). One-way ANOVA with Tukey's comparison post hoc analysis was applied to analyze 1-day sleep-wake activity post-manipulation in control, SPS, and MPS groups (Supplementary Figs. 7, 8). Sleep-wake activity alterations post-SPS manipulation and fear retrievals were also assessed by one-way ANOVA and followed by Tukey's comparison post hoc analysis (Supplementary Figs. 7, 8). $p < 0.05$ α level refers to a significant difference between groups. The details of statistical analysis information were provided in Supplementary Tables. 1–6.

**Reporting summary**. Further information on research design is available in the Nature Portfolio Reporting Summary linked to this article.

## Data availability

The data in the main figures and tables that support the findings of this study are available from the corresponding author upon reasonable request. Supplementary information and Supplementary Data 1 are available at communications biology's website.

## Code availability

The codes used in this study for analyzing LFP data and sleep spindle power are available from the corresponding author upon reasonable request.

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

## Acknowledgements

This work was supported by the Ministry of Science and Technology, ROC, grant 105-2320-B-002-059-MY3, 111-2320-B-002-040-, and National Taiwan University grant 112L892301. In addition, this article was subsidized for English editing by National Taiwan University under the Excellence Improvement Program for Doctoral Students (grant number 108-2926-I-002-002-MY4), sponsored by National Science and Technology Council, Taiwan.

## Author contributions

Y.L., F.-C.C., and P.-L.Y. designed the experiments. Y.L. and T.-Y.L. conducted the experiments. Y.L. and Y.-T.H. analyzed the data. Y.L. wrote the paper and created the experimental schematics in the article. F.-C.C. and P.-L.Y. reviewed and edited the paper.

## Competing interests

The authors declare no competing interests.
