## [Peer Review File · Communications Biology]

Reviewers' comments:

Reviewer #1 (Remarks to the Author):

This study is fundamentally an extension of the single prolonged stress (SPS) model of PTSD, where the SPS is repeated daily for one week with the addition of a cued fear conditioning paradigm. Multiple measures were evaluated at 6 weeks post-stress, including fear potentiated startle, fear-induced plasma cortisol, anxiety-like behavior, and sleep. The MPS elicited prolonged contextual and auditory fear memories that did not extinguish, enhanced anxiety, enhanced theta coherence across multiple circuits, and decreased REM sleep. Though the experiments represent a significant amount of work, it is not clear what benefits this study offers beyond the SPS model that it was based on. I believe all published studies of SPS examine measures either 24h or 7d post-stress; thus, it is not clear if the MPS procedure induces more persistent changes than SPS. The primary question of readers, I believe, will be “should I be using MPS instead of SPS, especially since SPS is easier?” The current manuscript does not shed light on this question. Given that there are already many models of chronic stress in rodents, this manuscript should do a better job explaining why another model is needed, and their data should show that this model is superior to existing models. Otherwise, the MPS model is unlikely to gain traction.

My major concerns are as follows:

1) There are other models of repeated, complex stressors that are not considered here. Most notably, the multiple acute stress model developed by Tallie Baram, which they have used both as a single stressor, and repeated over days (see Hokenson et al, *Bio. Protoc.*, 2020 for an example). Thus, this multiple stress model is not entirely new and is better conceptualized as an extension of the SPS model. However, it is not entirely clear what the authors why the authors felt this was necessary. Were there specific hypotheses about what would differ between the SPS and the MPS? Those hypotheses should be elaborated. The discussion also needs to do more to clearly delineate what is new about the present results. Why is the MPS procedure “better” than the SPS model?

Additionally, it seems like there are additional important controls missing. There were two different control groups used: one that received a 4hr sleep deprivation (but it is not clear that they were also exposed to the light-tone compound cue and the contexts) across MPS days, and one that was exposed to the light-tone compound cue and context across the MPS days. Since the important contrast between the current experiments and previously published work is the contrast between SPS and MPS, it seems like a control that received a single day of the MPS procedure (in essence, SPS with auditory fear conditioning) is needed. Having an SPS control would also have extended the novelty of the current study, as I don't believe any reports of SPS have looked beyond 7 days post-stress.

2) The pre-tone freezing for Day 1 of the MPS procedure and all auditory fear memory test days needs to be reported. Without these data, the reader cannot tell whether the freezing observed is due to fear of the tone, or generalized fear to the novel context.

3) It is strange to call the fear conditioning “IFS” (inescapable footshock stimulation”) instead of auditory fear conditioning. Also, the methods are unclear as to what was done. Did the shock coterminate with the compound light-tone stimulus or did it immediately follow?

4) The manuscript is, at times, difficult to read/interpret, because it is clearly written by a non-native speaker of English. Sometimes, the language incorrectly communicates what was done. For example, it is stated that the study modifies “the SPS process by increasing the stressor length and intensity”. However, it doesn't modify stressor length—it increases stressor repetitions.

Minor Points

The acronym "IFS" needs to be defined in the text the first time it is used (not defined in the text). It is not clear whether serum or plasma was collected. Serum requires that the blood be permitted to clot before spinning. Plasma requires the use of an anticoagulant. The dilution of the plasma/serum should be specified, as the ELISA manufacturer specifies a range. Also, it should be noted whether the reported values are dilution-corrected. Please clarify the methods.

Figure 1B and the methods should better describe what procedures were administered to the control groups. Were they at least handled during the days of the MPS procedure?

In Figure 5, the color black sometimes refers to baseline, and other times refers to Day 7 of MPS. This color should be used for only one time point across all panels.

Line 266 Please add a citation for the following statement "After a traumatic event, fear memory consolidation is intensively conducted 266 with sleep-wake activity"

Reviewer #2 (Remarks to the Author):

The manuscript by Lo and colleagues investigated the impact of multiple prolonged stress (MPS) on physiological and psychological alterations by analyzing long-lasting PTSD-like syndromes. They first reported that the MPS paradigm, lasting for long manipulation period (7 days) and manipulated by strong intensity of stressors, induced fear extinction deficits, which was correlated with theta oscillation coherence in PFC-BLA-vHPC pathway, and disrupted sleep-wake activities.

Single prolonged stress (SPS) has been mostly implicated to model PTSD syndrome in rat model in over 200 studies but also known to have limitation, such as transient effect which makes difficult to study long-lasting PTSD-like syndrome (Souza, et al. Front Pharmacol, 2017). The current study showed that MPS paradigm overcame this limitation, showing the prolonged effect to sustain fear memory, altered neural circuits, and sleep activities. However, there are several important issues that would first need to be addressed to make the authors' findings more conclusive.

Major comments:

The paper itself is well and carefully designed but the major concern is statistical analysis. Selective comparison of data may produce bias to data interpretation. The authors should reconsider the statistical analysis following as below.

1. Figure 2 & 3 and Supplementary Figure 1-3.:

All the data from in vivo electrophysiology has been compared only by t-test but not by ANOVA, even though there are several factors contributing to result. According to the behavior data in Figure 1, MPS group shows fear memory retrieval compared to control. To argue the impact of fear memory retrieval in MPS model, the data should be compared between MPS and control, otherwise the argument in this paper might be diluted.

For example, the power or coherency of theta oscillations were compared only before and after the context or cue retrieval but not between groups. Overall, the theta power, coherence and causality by

fear memory retrieval look lower in MPS group than control. Also, the theta power in MPS groups looks lower in short-term and higher in long-term effect than control. Moreover, the normalization from baseline can be considered for group comparison.

2. Figure 5 & 6.: The data was analyzed by one-way ANOVA with post-hoc test. But p values by post-hoc test are shown but p-value from ANOVA is not written.

Minor comments:

Figure 2e. Theta power spectrograms. The authors used the heatmap graph to plot the power of oscillation along time (3s before and 3s after cue) and theta frequency. However, it is difficult to understand this plot, since the color code seems reverted to the power of oscillation in scale bar and there is no variation of oscillatory power along frequency range (no variation of color code along y-axis). It is better to be checked.

We thank the editor and reviewers for reviewing this manuscript and providing constructive suggestions. According to the editor and reviewer's comments, this manuscript has been revised and additional experiments were added. These additional results were provided in the main text and supplementary text. The revised contents are written in red font with yellow highlight. The responses to each comment are answered point-by-point as follows.

Reviewer comments:

Reviewer: 1

Comments to the Corresponding Author:

This study is fundamentally an extension of the single prolonged stress (SPS) model of PTSD, where the SPS is repeated daily for one week with the addition of a cued fear conditioning paradigm. Multiple measures were evaluated at 6 weeks post-stress, including fear potentiated startle, fear-induced plasma cortisol, anxiety-like behavior, and sleep. The MPS elicited prolonged contextual and auditory fear memories that did not extinguish, enhanced anxiety, enhanced theta coherence across multiple circuits, and decreased REM sleep. Though the experiments represent a significant amount of work, it is not clear what benefits this study offers beyond the SPS model that it was based on. I believe all published studies of SPS examine measures either 24h or 7d post-stress; thus, it is not clear if the MPS procedure induces more persistent changes than SPS. The primary question of readers, I believe, will be “should I be using MPS instead of SPS, especially since SPS is easier?”

The current manuscript does not shed light on this question. Given that there are already many models of chronic stress in rodents, this manuscript should do a better job explaining why another model is needed, and their data should show that this model is superior to existing models. Otherwise, the MPS model is unlikely to gain traction.

Response: We sincerely appreciate reviewer's helpful criticism and recommendations for polishing our manuscript. We understood your concern regarding the benefits of this study offered beyond the SPS model; therefore,

we added the comparison experiments between SPS and MPS to elaborate the significance of MPS. We also added current models of chronic stress in rodents into the Introduction and Discussion sections, and explained why the MPS model is better than others as the PTSD rodent model. In this revised version we have added supplementary experiments and the revisions to each question in the text. The detail responses for individual question or concern are explained as follows.

Major Comments:

1. There are other models of repeated, complex stressors that are not considered here. Most notably, the multiple acute stress model developed by Tallie Baram, which they have used both as a single stressor, and repeated over days (see Hokenson et al, Bio. Protoc., 2020 for an example). Thus, this multiple stress model is not entirely new and is better conceptualized as an extension of the SPS model. However, it is not entirely clear what the authors why the authors felt this was necessary. Were there specific hypotheses about what would differ between the SPS and the MPS? Those hypotheses should be elaborated. The discussion also needs to do more to clearly delineate what is new about the present results. Why is the MPS procedure “better” than the SPS model?
Additionally, it seems like there are additional important controls missing. There were two different control groups used: one that received a 4hr sleep deprivation (but it is not clear that they were also exposed to the light-tone compound cue and the contexts) across MPS days, and one that was exposed to the light-tone compound cue and context across the MPS days. Since the important contrast between the current experiments and previously published work is the contrast between SPS and MPS, it seems like a control that received a single day of the MPS procedure (in essence, SPS with auditory fear conditioning) is needed. Having an SPS control would also have extended the novelty of the current study, as I don't believe any reports of SPS have looked beyond 7 days post-stress.

Response: We agree with reviewer's comments that we should elaborate more on the hypotheses of the MPS model and compare the advantages of MPS with that of SPS model. Therefore, we rewrote the content in the Introduction and Discussion sections, and added experiments to verify the differences in the long-term stress effects between MPS and SPS. The replies corresponding to each section are described as follows.

(1) We appreciate reviewer providing helpful references about multiple acute stress as a chronic repeated stress model. We are sorry about omitting the associated chronic manipulated stress model in our research review. We added the statements as: "For instance, restraint, electrical foot shock, and single prolonged stress (SPS) are single-manipulated physical stressors commonly used in rodents, whereas the repeated multiple concurrent stressors (RMS) utilizes repeated-stimulation to initiate chronic psychological and behavioral disabilities⁵⁻⁸" and "The RMS is a constant chronic paradigm, which combines physical, emotional, and social stressors for 10 days, and causes neuronal loss and cognitive impairments in rodents, to uncover the effects of life stresses^{5,17}" in the Introduction section on page 4, lines 58-61 and lines 64-67.

Additional references added are as follows.

Hokenson, R. *et al.* Multiple Simultaneous Acute Stresses in Mice: Single or Repeated Induction. *BIO-Protoc.* **10**, (2020).

Libovner, Y. *et al.* Repeated Exposure to Multiple Concurrent Stresses Induce Circuit Specific Loss of Inputs to the Posterior Parietal Cortex. *J. Neurosci.* **40**, 1849–1861 (2020).

(2) According to the reviewer's suggestion regarding the purpose of using MPS, we reconsidered and elaborated the hypothesis and necessity of the MPS model by indicating the individual difference and transient (short-term) effect of the original SPS model. The MPS model has simulated the features of repetitive high-intensity and chronic stress stimulation on rats, which displays superior effects to express the long-term symptoms. We added our

main purpose in the text as follows: “However, the single manipulation of SPS still has a limitation on the individual difference to generate constated symptoms in rodents, and the following long-lasting PTSD-like effects need to be further determined since fear extinction impairment is significantly observed in clinical cases ^{20,45–48}. In order to simulate the paradigm in which animals were exposed to a chronic, high-intensity stressor to cause PTSD and to stabilize the extent of PTSD-like abnormalities after stimulation, we modified the SPS process by increasing the stressor repetitions and intensity to establish the multiple stressors-induced long-term PTSD in rodents” on page 6, lines 91-99.

(3) We apologize for not clearly explaining the control group in the method. The control group in experiments 1 and 2 had the same manipulations with handling habituations and placing into the footshock box with the exposure of cues (both light and sound cues) during the first seven days of the MPS period. While in the experiment 3, we used MPS rats for comparison the alterations of sleep-wake activity before and after the MPS manipulation. Since the MPS protocol lasted for 4 hours and caused sleep deprivation (SD), the result of 4-hr SD was used as the control for comparison. The sleep-wake activities obtained from the baseline and 4-hr SD were recorded two days before the MPS. The advantage of using the same group of rats is to prevent the between-group variation generated by the signal power intensity. To let readers better understand the contrast control groups in experimental designs, we changed the icons in Figures 1b, 2a, 3a, 5a-b, and 6a. We also added the elaborative explanations on page 30, lines 553-556, page 31, lines 563-568, page 31, lines 571-572, page 33, lines 604-610, and page 57, lines 955-959.

(4) To directly assess the behavior difference after SPS and MPS manipulation, we added two additional experiments for evaluating the long-term effect of SPS on behavior tasks and sleep-wake activity. We used the protocol of day-1 MPS as the SPS pairing with fear conditioning control. A total of 12 rats were used in the experiment and divided into two subgroups

for different detection purposes. A group of SPS-manipulated rats (n=6) was utilized for anxiety behavior observation by using elevated plus maze (EPM) and open-field test (OFT). Another SPS group (n=6) was applied for sleep-wake activity recording. Both groups analyzed the immobility (%) during fear conditioning and memory retrieval (n=12). The results showed that the immobility (%) during contextual retrieval in SPS only persisted for one week. In contrast, the cue retrieval lasted for seven weeks and declined slowly. However, the long-term anxious behavior was more significant after context retrieval in the SPS group. The MPS group consistently presented high anxiety levels after context or cue recalls. The disrupted sleep time (%) was only shown after SPS manipulation, while the MPS affected the amount of sleep after stress manipulation and the fear retrieval days. These results indicated that the MPS could induce more stable and long-lasting PTSD-like symptoms than SPS manipulation.

The complementary methods are presented in the Supplementary Methods on page 8, lines 142-165, and page 10, lines 174-180. The graphical data are demonstrated in Supplementary Figures 5 (page 16, lines 230-268), 6 (page 19, lines 270-292), and 7 (page 21, lines 294-325). And the corresponding result explanations are displayed in the Discussion section on page 22, lines 402-405, page 24, lines 447-458, page 25, lines 462-466, page 26, lines 477-480, and page 26, lines 486-498 in the main text.

2. The pre-tone freezing for Day 1 of the MPS procedure and all auditory fear memory test days needs to be reported. Without these data, the reader cannot tell whether the freezing observed is due to fear of the tone, or generalized fear to the novel context.

Response: We thank reviewer's suggestion. To specify the freezing behavior responded to fear-conditioned cues and to exclude the effect of fear generalization when the rats exposing to a novel environment, we compare the

total immobile percentage during 10 mins between the control (only exposes to footshock box with cue tone stimuli) and the MPS group. The result from day 1 of the control group can represent the overall effects of fear generalization in response to the novel context. And we also determined the pre-tone immobilization on day 1 of MPS. The result showed a low freezing percentage (0.97 %) during the pre-tone period, which may exclude the generalized fear for novel context in the stress-manipulated group. This data is presented in the Supplementary Figure 5b and described in the Supplementary Figure legend on page 16, lines 235-237. Furthermore, we conducted our retrieval procedure in rat's origin home cages to exclude the freezing behavior due to the novel context during cue memory retrieval. The method is written in the Methods section on page 31, lines 574-576.

3. It is strange to call the fear conditioning "IFS" (inescapable footshock stimulation") instead of auditory fear conditioning. Also, the methods are unclear as to what was done. Did the shock coterminate with the compound light-tone stimulus or did it immediately follow?

Response: We apologize for not explicitly providing the specific procedure and depiction of the methods. For the statement about footshock, we understood the reviewer's opinion regarding the change from IFS to fear condition. However, we applied the IFS procedure four times in the MPS protocol as a primary stressor to induce PTSD-like behavior in rats. To effectively quantify the consequence of fear memory sustaining time, we paired the footshock with a combination of visual and auditory cue tones for the assessment of subsequent memory retrievals. The combined light and sound tones were immediately followed the footshock stimulation. Since we recognized this stimulation as major stress, we labeled it as IFS. To help clarify the experimental design of IFS, we added the descriptions on page 6, lines 101-105, and page 30, lines 561-563.

4. The manuscript is, at times, difficult to read/interpret, because it is clearly written by a non-native speaker of English. Sometimes, the language

incorrectly communicates what was done. For example, it is stated that the study modifies “the SPS process by increasing the stressor length and intensity”. However, it doesn’t modify stressor length—it increases stressor repetitions.

Response: We are sorry for not interpreting the methods and findings precisely. We have corrected the statement “modify stressor length” into “increases stressor repetitions” on page 3, line 38, and on page 6, lines 97-98.

Minor Comments:

1. The acronym “IFS” needs to be defined in the text the first time it is used (not defined in the text).

Response: We apologize for not being noticed in providing the acronym definition. We added the “inescapable footshock stress (IFS)” definition in the main text on page 4, lines 62-63.

2. It is not clear whether serum or plasma was collected. Serum requires that the blood be permitted to clot before spinning. Plasma requires the use of an anticoagulant. The dilution of the plasma/serum should be specified, as the ELISA manufacturer specifies a range. Also, it should be noted whether the reported values are dilution-corrected. Please clarify the methods.

Response: We apologize for not providing a clear protocol for sample collecting, dilution, and the commercial kit’s specificity. Therefore, we added further descriptions in the Supplementary Methods section as follows: “The blood samples were collected 5 mins after fear memory retrieval and staying in the room temperature for 10 mins for clot. Then, the serum was prepared by 13,000 rpm centrifuged for 20 mins at 4°C. All the experimental protocol was followed by the instruction of a commercial ELISA kit (Enzo Life Sciences, NY, USA) with the sensitivity of 27 pg/ml (range between 32 and 20,000 pg/ml). To fit into the sensitivity standard range, the serum samples were diluted 1:100 with the sample dilution buffer provided by the kit before the subsequent procedure, which the averaged concentration was at the range between 4,000 and 10,000 pg/ml. And the diluted sample was processed by the kit’s instruction”

and described on page 8, lines 129-138 of the Supplementary information. We also replaced all the statement of “plasma corticosterone concentrations” into “serum corticosterone concentrations” in the main text on page 3, lines 40-41 and page 8, lines 128-129.

3. Figure 1B and the methods should better describe what procedures were administered to the control groups. Were they at least handled during the days of the MPS procedure?

Response: We are sorry for not clearly described the experimental procedure on control group. We added the expanded information of the control group manipulation during the first seven days (days of the MPS procedure) as follows: “The rats in control group were well habituated by gently handling using a towel for 10 mins during the first seven-day procedure without MPS manipulation. In order to compare the immobility with the MPS group, the control group was further placed in the footshock box with 2% vinegar and given 12 cue tones (without IFS) for 10 mins on the odd days of the seven-day treatment (a total of four times)” on page 31, lines 563-568 and “both contextual and cue fear memory retrievals were detected after the MPS (MPS group) and a seven-day habituated protocol (control group)” on page 31, lines 571-572. For better understanding the comparison, we also changed the icons in Figures 1b, 2a, 3a, and 5a-b.

4. In Figure 5, the color black sometimes refers to baseline, and other times refers to Day 7 of MPS. This color should be used for only one time point across all panels.

Response: We appreciate your advice on keeping colors the same when showing graphics. To make the differences between the baseline and day 7 in the sleep-wake activity more obvious, we adjusted the color bars to black (baseline) and dark green (day 7) in Figures 6 and 7. Also, the comparison between the control and MPS groups refers to gray and black bars, respectively, shown in Figures 1 (page 48, line 851) and 5, and Supplementary Figures 2, 5, and 6.

5. Line 266 Please add a citation for the following statement “After a traumatic event, fear memory consolidation is intensively conducted with sleep-wake activity”

Response: We thank reviewer for your kind reminder, we have added corresponded studies (Pace-Schott et al., 2015 and Popa et al., 2010) of this statement on page 16, lines 294-295.

Pace-Schott, E. F., Germain, A. & Milad, M. R. Effects of sleep on memory for conditioned fear and fear extinction. *Psychol. Bull.* 141, 835–857 (2015).

Popa, D., Duvarci, S., Popescu, A. T., Lena, C. & Pare, D. Coherent amygdalocortical theta promotes fear memory consolidation during paradoxical sleep. *Proc. Natl. Acad. Sci.* 107, 6516–6519 (2010).

Reviewer: 2

Comments to the Corresponding Author:

The manuscript by Lo and colleagues investigated the impact of multiple prolonged stress (MPS) on physiological and psychological alterations by analyzing long-lasting PTSD-like syndromes. They first reported that the MPS paradigm, lasting for long manipulation period (7 days) and manipulated by strong intensity of stressors, induced fear extinction deficits, which was correlated with theta oscillation coherence in PFC-BLA-vHPC pathway, and disrupted sleep-wake activities.

Single prolonged stress (SPS) has been mostly implicated to model PTSD syndrome in rat model in over 200 studies but also known to have limitation, such as transient effect which makes difficult to study long-lasting PTSD-like syndrome (Souza, et al. Front Pharmacol, 2017). The current study showed that MPS paradigm overcame this limitation, showing the prolonged effect to sustain fear memory, altered neural circuits, and sleep activities. However, there are several important issues that would first need to be addressed to make the authors' findings more conclusive.

Major Comments:

The paper itself is well and carefully designed but the major concern is statistical analysis. Selective comparison of data may produce bias to data interpretation. The authors should reconsider the statistical analysis following as below.

Response: We appreciate reviewer's comments and suggestions for strengthening our study. To address the concerns for our study, we re-analyzed data and revised some of the contexts in our manuscript. The responses to each question and the additional contents are presented below.

1. Figure 2 & 3 and Supplementary Figure 1-3.:

All the data from in vivo electrophysiology has been compared only by t-test but not by ANOVA, even though there are several factors contributing to

result. According to the behavior data in Figure 1, MPS group shows fear memory retrieval compared to control. To argue the impact of fear memory retrieval in MPS model, the data should be compared between MPS and control, otherwise the argument in this paper might be diluted.

For example, the power or coherency of theta oscillations were compared only before and after the context or cue retrieval but not between groups. Overall, the theta power, coherence and causality by fear memory retrieval look lower in MPS group than control. Also, the theta power in MPS groups looks lower in short-term and higher in long-term effect than control. Moreover, the normalization from baseline can be considered for group comparison.

Response: We really thank for reviewer's helpful suggestions on statistical analysis. The replies corresponding to each question are described as follows. (1) We agree that the pre-treated and post-treated power and coherency theta oscillations (Figures 2 and 3) should consider several factors during the comparison. However, the large data of power differences between the control and MPS groups may vague the significant tendency of the result, if we calculated these groups in the same parameter. Thus, we analyzed the effect in each recorded brain area using one-way ANOVA with Tukey's post-hoc comparison to include the factor of immediate reaction to the retrieval and the prolonged effect towards days in the control and MPS groups. The change of method was shown on page 34, lines 620-626. The renewed graphical significances are shown in Figure 2 and 3, and the descriptions of results were rewritten on page 9, lines 147-152, page 10, lines 164-167, page 11, lines 191-200, page 12, lines 206-207. Nonetheless, to respectively present the distinctions before and after the fear retrieval, the trend of theta spectrogram in the Supplementary Figures 1 and 3 were still analyzed by the paired *t*-test. This statistical analysis method is presented in the Supplementary Methods on page 10, lines 167-174.

(2) The comparison of causality within the circuitry of PFC-BLA-vHPC is more focused on determining the dominant direction of signal transitions, either from the fear consolidation directions or fear extinction directions. Therefore, we thought the paired *t*-test would be more appropriate for identifying the strength

of shifted oscillations following fear retrieval and the prevailing functional direction. Additionally, we changed the causality Figure's order from the Supplementary information to Figure 4 in the main text. We also re-organized the illustration in the Results section on page 13, lines 238-242.

(3) We appreciate the reviewer's suggestion for using baseline normalization to examine the differences between the control and MPS groups rather than relying on pre- and post-treatment data to infer effects. The between-group difference in power intensity is too large to directly examine, as was already indicated. Although we made an effort to compare the normalized data between the control and MPS groups, the oscillation power findings showed no differences between the groups (all the data are close to zero), and the standard error of the mean is very high. Hence, we calculated the variations by subtracting the pre-treated from the post-treated data. We then used unpaired *t*-test to compare the discrepancy between the control and MPS groups. The analyzing method is added in Supplementary information on page 7, lines 112-115, and page 10, 169-172. A complemented Figure is displayed in Supplementary Figure 2. The results that correspond to it are described on page 9, lines 149-152 and 155-157, page 10, lines 167-171, page 11, lines 188-191 and 200-206 in the main text.

2. Figure 5 & 6.: The data was analyzed by one-way ANOVA with post-hoc test. But p values by post-hoc test are shown but p-value from ANOVA is not written.

Response: We appreciate reviewer's kind reminder. We did not show the p values in the main text due to the word limitation in the first version. To better portray the findings in the result section and avoid obstructing the narrative flow of the main text, we added the p-value of ANOVA in the legends of Figures 2 (page 50, lines 871-876, and page 51, lines 883-887), 3 (page 52, lines 898-903 and 907-911), 6 (page 58, lines 964-966 and 971-973), and 7 (page 59, lines 982-986, page 60, lines 994-995 and 1002-1006, and page 61, lines 1012-1013 and 1018-1021) and in the Tables 1-3 (page 64, lines 1029-1032; page 65, lines 1038-1039; page 66, lines 1044-1045). We also added the statistical p-value in the legends of Supplementary Figures 5 (page 16, lines 231-268), 6

(page 20, lines 271-292), and 7 (page 21, lines 295-325) in the Supplementary information.

Minor Comments:

1. Figure 2e. Theta power spectrograms. The authors used the heatmap graph to plot the power of oscillation along time (3s before and 3s after cue) and theta frequency. However, it is difficult to understand this plot, since the color code seems reverted to the power of oscillation in scale bar and there is no variation of oscillatory power along frequency range (no variation of color code along y-axis). It is better to be checked.

Response: We apologize for our negligence on the coding of power spectrograms. The origin parameters of time-bandwidth and number of tapers is 25-30, which cannot display the variation of our oscillatory power within the range of 4-7 Hz. Therefore, we reassessed the tapers of theta power spectrograms, and altered the parameter into 3-5 for calculating the theta power spectrogram and coherency variation during fear memory retrieval. This adjustment can be found in the Supplementary Methods on page 6, lines 106-107. The renewed data are presented in the Figures 2e-f, 3c, and Supplementary Figure 3. The corresponding descriptions are written in the Results section of main text on page 9, lines 147-152, page 10, lines 164-167, page 11, lines 191-200, page 12, lines 206-207.

Reviewers' comments:

Reviewer #1 (Remarks to the Author):

I appreciate that the authors have extensively revised their manuscript. I also particularly appreciate the addition of the SPS data, which I think is a necessary contrast with the MPS to provide a rationale for why the MPS is needed or how it is different. One of my big concerns previously was that the procedure for inducing the MPS was not clearly described. Now that I understand it, I have some significant concerns about the design of the IFS/fear conditioning aspect. Those are detailed below. The manuscript also needs clarity throughout in the grammar and writing. I also think the discussion should more tightly summarize the key persistent changes in the different measures (when they peak), and better consider the limitations of the study design. My detailed concerns are described below.

1. There are methodological aspects of the experiment that I still cannot decipher between the main text and Methods. For example, the “fear conditioning” paradigm. It is described both as “fear conditioning” and as “IFS”. The authors should use one name throughout. It is also clearly backwards fear conditioning, which is very unusual: a compound light-sound stimulus is given “immediately following” each shock. What was the rationale for using backwards conditioning, rather than standard fear conditioning where predictive cues precede the shock? Backwards conditioning does induce fear to the cue (in this case, light and tone together), but it does so by causing retrieval of the context (an indirect way to retrieve a fear memory). Why did the authors design the IFS in this manner? Since contextual vs cued fear conditioning was a major comparison in the data analyses in this paper, it seems like forward conditioning, where the tone/light directly form an association with the footshock, would have been a better comparison. Backwards conditioning, I think, would minimize potential differences between the context and cue comparisons. This should at least be considered in the discussion. The methods specify that the unstressed control group received presentations of the tone, rather than the compound light-tone stimulus during the first 7 days of the experiment. Why? Was the light meant to serve as an additional US, or was it used as a CS? The Methods specify that fear retrieval was measured by “randomly conveyed 12 times of light and sound cues in a 10-min period.” Was it 12 presentations of the compound stimulus, or a subset of the 12 were light alone and the remainder were sound alone (it sounds like the latter)? What does “randomly” mean (different for each animal)? Without these details clearly stated, it is impossible for a reader to think about implementing this procedure in their own experiments. I also think part of this confusion arises because the manuscript is written by authors who are not native speakers of English. As I stated in my previous review, this manuscript has many places where details are not clear because the language used is ambiguous. I list just a few below, but there are many, many more. This manuscript needs to be thoroughly edited by a fluent English speaker to improve the clarity of the text—changing just the examples below is not sufficient.

Examples:

Line 91 “the single manipulation of SPS still has a limitation on the individual difference to generate constated symptoms in rodents” I have no idea what this means. “Constated” is not a word in English, but even that aside, I cannot decode this sentence. This is an important sentence because it explains the need for the MPS.

Line 558 “electrical stimuli... were randomly given in a 10-min period...with 2% of vinegar at the bottom

of the box ... as an unconditioned stimulus” The phrase “as unconditioned stimuli” should directly follow the words “randomly given” because it is the electrical stimuli that serve as the US, not the odorant used (vinegar at sufficiently high concentrations could be used as a US).

Line 588 “The long-term procedure was examined three weeks after the MPS stimulation,” Wasn’t it four weeks (i.e. 28 days) after the end of the MPS procedure?

Line 158 “we measured the neuronal activities 3-sec before (pre) and 3-sec after (post)” When you say “3-sec after” the tone cue, this means the 3 seconds that immediately follow the tone cue termination. However, I think what the authors mean is the first 3-sec after the tone cue onset? This needs to be clear. If the authors did not analyze theta during the tone, then statements like “significantly increased power during the 1-sec after cue onset” (line 161) don’t make sense.

Line 175 “after MPS strengthen the” should read “strengthens”

Line 294 “fear memory consolidation is intensively conducted with sleep-wake activity” Conducted is not the correct word to use—it doesn’t make sense in this context.

2. The IFS procedure seems a lot like the SEFL procedure first described by the Fanselow lab (except they didn’t have the backwards conditioning aspect included). Why is this chronic stress procedure not cited or compared to the findings here?

3. Line 40 “extended fear memory was estimated by fear retrieval induced-startle responses” I cannot find anything in the methods or main text that reports startle responses. I see only analysis of freezing behavior. Where is the startle data? Freezing is NOT the same thing as startle.

4. The authors note that the primary “new” thing about this manuscript is the assessment of PTSD-like changes longitudinally. Yet, the abstract does not describe this. The text about the theta coherence, anxiety, and sleep alterations does not describe when changes were observed. This should be clearly stated in the abstract so that the reader can see what is new in this manuscript.

5. Line 126 “the prolonged stress effect of MPS and the impairment of fear memory extinction.” The authors claim their procedure results in fear extinction impairment but I don’t see data to support this. There is no control group that shows faster fear extinction (the only control did not receive any fear conditioning, and thus has no fear memory to extinguish). The MPS group does show contextual extinction. They do not show extinction to the tone cues (or was it the compound tone-light cue?) but the behavior is at a ceiling level and so extinction have been present but not detected. To claim an “impairment” in fear extinction, one needs a group that shows what “normal” fear extinction should look like, since “impaired” is a relative term.

6. The process for measuring cue retrieval should be better described, since it was performed in the home cage. What was used to deliver the tones and lights? Was it the same equipment that was used for the fear conditioning in the boxes? How was the equipment secured to the home cage? Were the animals awakened prior to delivery of the cues? Line 69, Supp says “The percentage of immobility was

quantified as the rigid movement ceased over 3 sec” What does the 3sec refer to? Aren’t the reported freezing percentages for the entire 10 minute assessment period?

7. Freezing was scored manually, introducing the possibility of bias. Were experimenters blinded to which sessions were being scored? How many experimenters scored each video? What was the inter-individual variation in freezing score?

8. Figure 1. The design of this experiment was such that the same animals were repeatedly given “reminders” of the original MPS treatment, because they were repeatedly tested for fear memory retrieval. It is possible that this contributed to the persistence of the fear memory. To know that the fear memory truly persisted for this long, one would need a group of animals that underwent MPS and were tested for fear memory only once, at the 6 week time point. This limitation should at least be discussed.

9. For Figures 2 and 3, the correct statistical test is a repeated measures 2X2 ANOVA with factors of group and place (homecage vs retrieval box). These ANOVAs are not reported.

10. Line 137 “We next evaluated whether the neuronal activities during fear retrieval were correlated with the startle responses.” I do not see any data examining correlations between measures of oscillatory activity and fear retrieval (freezing).

11. Line 255 “Stress-induced anxiety commonly occurs when fear memory flashbacks,” Please provide citations for this.

12. Line 290 “the neuronal activities in the correlated fear-associated regions could be observed when fear memory was retrieved.” This statement is misleading. The regions aren’t “correlated” (wrong word), and the neuronal activity measured during the anxiety tests were not “observed when fear memory was retrieved” (it was measured AFTER fear memory retrieval).

13. In several places, it is stated that MPS effects are “more significant” than SPS effects. This is not an appropriate statement. There is no such thing as more/less significant. You can compare the size of an effect, by testing whether there is a group (MPS vs SPS) effect in a measure, but you do not compare the p-values of independent tests across the two groups. This should be corrected in multiple places.

14. The memory retrieval aspect of the MPS model is only retrieving one aspect of the stressful experience. The authors cannot know that this induces a memory retrieval of other aspects of the stressors (for example, the isoflurane exposures). The authors seem to imply that the “memory” being retrieved is that of the stressful experience in its totality.

15. I think that the finding that the MPS produces stronger effects than the SPS on multiple measures is a key finding of this paper. I think that the SPS data should be included in Figure 1 and discussed up front in the results.

Reviewer #2 (Remarks to the Author):

I am convinced that the paper will be of substantial interest to the community. Due to the lack of reliable stress protocol for modeling any stress-related disorders, including PTSD or anxiety disorders, the community demanded the another stress model. Through this revision process, the authors nicely improved their article by adding more data and more elaborate description to strengthen their argument.

- 1) The authors corrected their missing points.
- 2) The authors added the description in Introduction and Discussion and added another set of data which compare MPS model with SPS model to strengthen the necessity of MPS protocol instead of SPS protocol for modeling PTSD.
- 3) Also, calculating the difference between before-after (Supplementary Figure 2) was a nice idea to interpret the data in a simple way.

But, still, as previous comments, I am worried about the statistical analysis and suggest to reconsider. The selective choice of statistical analysis seems that the authors analyzed the data in a biased way. The authors should be stricter about the statistical analysis according to the rules.

Figure 1c-e: Here, the authors used unpaired-t-test just to analyze the group difference in various states. Since the same mice were used in different states, the authors should use Two-way ANOVA with posthoc test or at least, paired t-test.

Figure 2d-f: The authors compared two groups by One-way ANOVA with Tukey posthoc instead of t-test. This was unnecessary and no point of changing statistical method. In theory, this dataset should be compared to see the group effect and condition effect (home cage vs context), simultaneously. For this purpose, the authors should use Repeated Measures of ANOVA with posthoc test, if the same mice were used in homecage and context.

It was also surprising to see the huge group difference of theta power in home cage (Figure 2d) and pre-cue tone (Figure 2e), which may indicate the basal status. It seems that the theta power of MPS rats are higher in basal condition, which was not mentioned by authors. The authors should consider about discussing this difference.

We thank the editor and reviewers for reviewing this manuscript and providing constructive suggestions. According to the reviewer's comments, this manuscript has been revised. Furthermore, this manuscript has also been edited by a professional editor, Christina C., from a professional manuscript editing service, Wordvice Inc. The revised contents are written in red font with yellow highlight. The responses to each comment are answered point-by-point as follows.

Reviewer comments:

Reviewer: 1

Comments to the Corresponding Author:

I appreciate that the authors have extensively revised their manuscript. I also particularly appreciate the addition of the SPS data, which I think is a necessary contrast with the MPS to provide a rationale for why the MPS is needed or how it is different. One of my big concerns previously was that the procedure for inducing the MPS was not clearly described. Now that I understand it, I have some significant concerns about the design of the IFS/fear conditioning aspect. Those are detailed below. The manuscript also needs clarity throughout in the grammar and writing. I also think the discussion should more tightly summarize the key persistent changes in the different measures (when they peak), and better consider the limitations of the study design. My detailed concerns are described below.

Response: We thank the reviewer's comments for specifying the explanation of our findings in this manuscript. We understand your concerns about the IFS procedure; thus, we provided more information of the experimental materials and methods in the Methods section.

To present the main findings of this study, we added the discussion to summarize the significance and limitations of the MPS model in the Discussion section.

The summarized persistent alteration of MPS is displayed on page 21, lines 388-399, which complements the significantly prolonged period of each detected indicator.

The added limitations of our experimental results, including the fear extinction observation, variation of basal LFP power between the groups, and the mechanism of circuitry transmission in the MPS, are presented on page 30, lines 560-586.

We also apologize for the unclear writing and grammar errors in our manuscript. We have reconstructed the manuscript, and the article was edited through an English editing agency. The detailed responses to individual questions or concerns are explained as follows.

1. There are methodological aspects of the experiment that I still cannot decipher between the main text and Methods. For example, the “fear conditioning” paradigm. It is described both as “fear conditioning” and as “IFS”. The authors should use one name throughout. It is also clearly backwards fear conditioning, which is very unusual: a compound light-sound stimulus is given “immediately following” each shock. What was the rationale for using backwards conditioning, rather than standard fear conditioning where predictive cues precede the shock? Backwards conditioning does induce fear to the cue (in this case, light and tone together), but it does so by causing retrieval of the context (an indirect way to retrieve a fear memory). Why did the authors design the IFS in this manner? Since contextual vs cued fear conditioning was a major comparison in the data analyses in this paper, it seems like forward conditioning, where the tone/light directly form an association with the footshock, would have been a better comparison. Backwards conditioning, I think, would minimize potential differences between the context and cue comparisons. This should at least be considered in the discussion. The methods specify that the unstressed control group received presentations of the tone, rather than the compound light-tone stimulus during the first 7 days of the experiment. Why? Was the light meant to serve as an additional US, or was it used as a CS? The Methods specify

that fear retrieval was measured by “randomly conveyed 12 times of light and sound cues in a 10-min period.” Was it 12 presentations of the compound stimulus, or a subset of the 12 were light alone and the remainder were sound alone (it sounds like the latter)? What does “randomly” mean (different for each animal)? Without these details clearly stated, it is impossible for a reader to think about implementing this procedure in their own experiments. I also think part of this confusion arises because the manuscript is written by authors who are not native speakers of English. As I stated in my previous review, this manuscript has many places where details are not clear because the language used is ambiguous. I list just a few below, but there are many, many more. This manuscript needs to be thoroughly edited by a fluent English speaker to improve the clarity of the text—changing just the examples below is not sufficient.

Examples:

Line 91 “the single manipulation of SPS still has a limitation on the individual difference to generate constated symptoms in rodents” I have no idea what this means. “Constated” is not a word in English, but even that aside, I cannot decode this sentence. This is an important sentence because it explains the need for the MPS.

Line 558 “electrical stimuli... were randomly given in a 10-min period...with 2% of vinegar at the bottom of the box ... as an unconditioned stimulus” The phrase “as unconditioned stimuli” should directly follow the words “randomly given” because it is the electrical stimuli that serve as the US, not the odorant used (vinegar at sufficiently high concentrations could be used as a US).

Line 588 “The long-term procedure was examined three weeks after the MPS stimulation,” Wasn’t it four weeks (i.e. 28 days) after the end of the MPS procedure?

Line 158 “we measured the neuronal activities 3-sec before (pre) and 3-sec after (post)” When you say “3-sec after” the tone cue, this means the 3 seconds that immediately follow the tone cue termination. However, I think what the authors mean is the first 3-sec after the tone cue onset? This needs to be clear. If the authors did not analyze theta during the tone, then statements like “significantly increased power during the 1-sec after cue onset” (line 161) don’t make sense.

Line 175 “after MPS strengthen the” should read “strengthens”

Line 294 “fear memory consolidation is intensively conducted with sleep-wake activity” Conducted is not the correct word to use—it doesn’t make sense in this context.

Response: We are sorry for providing ambiguous descriptions in the article; we have revised the content of the main text and added more explanations in the Methods sections to clarify the experimental protocols.

The replies correspond to each question are described as follows.

(1) We apologize for referring to the primary stressor, IFS, with two separate nouns (IFS and fear conditioning). With the exception of the definition of IFS in the Methods section (page 33, lines 613-616), where we believe providing this interpretation would make the purpose of manipulating IFS more clearly stated, we have changed the expression of "fear conditioning" to "IFS" to help the readers better understand the content of this article.

The changed of the nouns are presented in the main text on page 7, lines 118-120, page 28, lines 508-511 and lines 517-518, page 29, lines 533-537, page 57, lines 1010-1012.

(2) We are sorry for our imprecise descriptions of IFS manipulating procedure in the Methods section that confused readers about the order of cue tone and electric shock. We used “forward” fear conditioning in our tests, where an electric shock was delivered right away following a 1-sec compounded light-sound cue tone. The sentence has been

rewritten as follows: “In addition to being designed the primary stressor to trigger PTSD-like behavior, the IFS was linked with particular forward fear conditioning context and cues to ensure fear memory manifestation throughout the experimental phase.” on page 33, lines 613-616 and “During IFS, a 1-second light (34 lux) and sound (2,000 Hz, 80 dB) compounded cue was delivered immediately before every electric stimulus as conditioned stimuli for further fear memory retrieval tasks.” on page 34, lines 625-627 in the Methods.

(3) Similar to the MPS group, the control group also received 12 compounded light-sound cue tones (referred to as CS) in the seven days protocol. The complementary details are presented as follows: “To compare the immobility with the MPS group, the control group was placed with the same footshock box settings with 2 % vinegar and 12 compounded light-sound cue tones (without IFS) for 10 minutes on odd days during the seven-day treatment (four times total).” in the Methods on page 34, lines 635-638.

(4) We apologize again for neglecting the central information of the IFS implementation process. The term “randomly” refers to the time points of 12 compounded (light-sound) cue tones used in conjunction with stimuli during a 10-min session. In other words, all shock intervals were uncertain, and each stimulation was administered at a random time point within a 10-min window. However, all the rats were applied to the same IFS treatment on the same day.

We have added more details to further clarify the random time point of each shock in the IFS as follows: “Twelve electrical stimulations (5 mA, 133 V, 0.5 sec for each stimulus; World Precision Instruments, FL, USA) were randomly distributed over 10 minutes. Each shock was administered at a random time point, and all the interstimulus intervals were uncertain in each IFS implementation process (all rats experienced the same IFS treatment on the same day) as an unconditioned stimulus. Control group rats were well-habituated through

gently handling using a towel for 10 minutes during the first seven-day procedure without MPS manipulation.” in the Methods on page 34, lines 628-635.

(5) We are sorry for the insufficient expression throughout the manuscript.

We have rewritten the ambiguous depictions, which native English speakers have further edited to improve legibility.

The partial sentences that the reviewer had pointed out are revised as follows:

Page 6, lines 91-96.

“However, the modified SPS procedure divergence in some research generated variant behavioral and neuronal consequences during different periods, indicating that SPS’ PTSD-like effects were time- and experience-dependent²⁰. Further research is necessary to determine long-lasting PTSD-like consequences since clinical patients experience significant fear extinction impairment⁴⁵⁻⁴⁸.”

Page 33, lines 617-620 and page 34, lines 628-633.

“IFS was conducted in a 28 x 22 x 40 cm³ footshock box with a stainless-steel rods board on the floor and 2% vinegar at the bottom of the box (background brightness: 3.5 lux) for conditioned contextual stimulation.”

“Twelve electrical stimulations (5 mA, 133 V, 0.5 sec for each stimulus; World Precision Instruments, FL, USA) were randomly distributed over 10 minutes. Each shock was administered at a random time point, and all the interstimulus intervals were uncertain in each IFS implementation process (all rats experienced the same IFS treatment on the same day) as an unconditioned stimulus.”

Page 37, lines 677-679.

“The long-term procedure was examined four weeks post-MPS stimulation, with context retrieval on Days 35 and 46 and cued retrieval on Days 37 and 44.”

Page 10, lines 172-174.

“To investigate theta power variation after cue retrieval, we measured three brain regions’ neuronal activity 3 seconds before (pre) and after (post) cue tone onset (present as 0) (Fig. 2c, e).”

Page 11, lines 192-193.

“Post-MPS fear retrieval strengthens PFC, BLA, and vHPC theta oscillation coherence and connectivity”

Page 17, lines 308-309.

“After a traumatic event, sleep-wake activity notably enhances fear memory consolidation^{54,55}.”

2. The IFS procedure seems a lot like the SEFL procedure first described by the Fanselow lab (except they didn't have the backwards conditioning aspect included). Why is this chronic stress procedure not cited or compared to the findings here?

Response: We thank reviewer for providing a considerable model, stress-induced enhancement of fear learning (SEFL), for us to discuss the chronic stress procedure and impact in our MPS model. We have complemented the operation of SEFL and the significant effects on PTSD-like-symptoms as follows: “Prior stress exposure enhanced the fear learning tasks also described in the stress-induced enhancement of fear learning (SEFL) model⁶⁸. SEFL utilized 15 unpredictable and inescapable electric footshock stimuli within a 90-minute experimental period before a single shock stimulation (1 mA, 1 sec for each shock) to increase the rat's sensitivity toward a subsequent mild stressor (single shock) and induced long-term PTSD-like symptoms^{68,69}. Previous studies verified that pre-exposure to repeated shocks enhances fear learning ability to the following fear-conditioned paired context or cue tone, which might be due to increased anxiety after shock stress^{68,70}. These results complement our findings that multiple stressors manipulation before the main shock (IFS)

intensified contextual and cue-paired fear learning efficacy in the rats ^{71,72}.”
on page 24, lines 445-456 in the Discussion.

3. Line 40 “extended fear memory was estimated by fear retrieval induced-startle responses” I cannot find anything in the methods or main text that reports startle responses. I see only analysis of freezing behavior. Where is the startle data? Freezing is NOT the same thing as startle.

Response: We apologize for using the incorrect term in the text to describe the freezing behavior; we have changed the startled reactions to “freezing behavior” or “immobilized behavior” instead.

The modified sentences are located at page 3, lines 43-45, page 7, lines 118-120, and page 23, lines 412-413, page 29, lines 530-533.

4. The authors note that the primary “new” thing about this manuscript is the assessment of PTSD-like changes longitudinally. Yet, the abstract does not describe this. The text about the theta coherence, anxiety, and sleep alterations does not describe when changes were observed. This should be clearly stated in the abstract so that the reader can see what is new in this manuscript.

Response: We thank the reviewer for the kind reminder for emphasized the longitudinal effects of the MPS model in each assessment experiment to clearly stated our key findings. We have added the main statement in the abstract as follows: “Behavioral and neural changes following MPS conveyed longitudinal PTSD-like effects in rats for six weeks.” on page 3, lines 41-43. We also change the descriptions of theta coherence, anxiety, and sleep alterations on page 3, lines 38-52 in the Abstract.

5. Line 126 “the prolonged stress effect of MPS and the impairment of fear memory extinction.” The authors claim their procedure results in fear extinction impairment but I don’t see data to support this. There is no control group that shows faster fear extinction (the only control did not receive any fear conditioning, and thus has no fear memory to extinguish). The MPS group does show contextual extinction. They do not show

extinction to the tone cues (or was it the compound tone-light cue?) but the behavior is at a ceiling level and so extinction have been present but not detected. To claim an “impairment” in fear extinction, one needs a group that shows what “normal” fear extinction should look like, since “impaired” is a relative term.

Response: We are sorry for using the wrong word to explain our results. The freezing behavior constantly presented a high percentage in the MPS group throughout six weeks process. Even though a decline during context recall has displayed from the fifth week (day 35), a significant increase still presented in the MPS group when compared to the control and SPS groups. Moreover, the elevated immobility (%) during cue retrieval persisted for six weeks in both SPS and MPS groups. Therefore, we did not observe a fear memory extinction in the MPS group in both contextual and cue retrieval in a six weeks experimental process. However, the SPS rats significantly decreased the immobility (%) during remote context recall (day 35), demonstrating the fear memory extinction after SPS with IFS administration. Thus, we changed the statement of fear memory retrieval in the MPS rats as follows: “These findings suggest that MPS has a stronger fear memory sustainability for contextually associated fear than SPS.” on page 8, lines 141-143.

6. The process for measuring cue retrieval should be better described, since it was performed in the home cage. What was used to deliver the tones and lights? Was it the same equipment that was used for the fear conditioning in the boxes? How was the equipment secured to the home cage? Were the animals awakened prior to delivery of the cues? Line 69, Supp says “The percentage of immobility was quantified as the rigid movement ceased over 3 sec” What does the 3sec refer to? Aren’t the reported freezing percentages for the entire 10 minutes assessment period?

Response: We thank the reviewer for reminding the missing details of the setup of cue tone delivering equipment and the analytic method of freezing

percentage. We have added additional illustrations to specify our methods. The replies corresponding to each question are described as follows.

- (1) We apologize for omitting the detailed devices and setup information for compounded cue tones. The descriptions of the lighting and audio sources, as well as the locations of the equipment used for IFS and cue retrieval, have been added as follows: "Additionally, to pair the electric shocks with compounded light and sound cue tones as conditioned stimulations, we positioned the light source (LED light bulb, white light, 1200 lm, 10 W; Everlight Electronics Co., Taipei, Taiwan), and sound source (SRS-XB13 speaker; Sony, Tokyo, Japan) 80 cm right above and directly above the footshock box, respectively." on page 34, lines 620-625 and "In addition, we validated the rats' immobility behavior in their home cages during cue retrieval to exclude novel context effects. Twelve compounded light and sound cues were randomly generated over 10 minutes. The same IFS lighting and audio equipment conditions were used in the cue retrieval test; thus, we placed a light bulb directly above the animal's home cage to provide the same brightness level (34 lux). We also positioned a speaker atop the home cage to produce the same sound cue volume (2,000 Hz, 80 dB) to recall the fear memory." on page 36, lines 657-665.
- (2) During the context and cue retrieval, all rats were awakened before each recall task to prevent the sleep behavior-induced bias on immobility percentage calculation. We have added this criterion to the Method section as follows: "Moreover, all rats needed to be awake during both fear retrieval processes to prevent sleep-caused deviation from affecting immobility calculation." on page 36, lines 665-667.
- (3) We appreciate the reviewer for indicating the insufficient analyzing details of the immobility behavior percentage. The 3-sec in our immobility behavior quantifying refers to the criteria of the freezing

behavior threshold that the recorded videos were artificial analyzed every three seconds. Once the rats performed absence of body, whisker and nose movement (apart from the respiration) at least persisted for three seconds or more would be scored as freezing and accumulated into the following immobilization duration. And finally, the accumulated immobility duration within the 10 mins period of each rat was further analyzed as percentage. The expansion of descriptions is shown on page 4, lines 65-81 in the Supplementary Methods.

7. Freezing was scored manually, introducing the possibility of bias. Were experimenters blinded to which sessions were being scored? How many experimenters scored each video? What was the inter-individual variation in freezing score?

Response: We understood the reviewer's concern about the scoring bias between different operators. However, all the recorded videos were blindly provided to one experimenter to analyze the freezing behavior of rats, which can prevent the inter-individual variation of the immobility scoring. This statement has been presented on page 4, lines 70-73 in the Supplementary Methods.

8. Figure 1. The design of this experiment was such that the same animals were repeatedly given "reminders" of the original MPS treatment, because they were repeatedly tested for fear memory retrieval. It is possible that this contributed to the persistence of the fear memory. To know that the fear memory truly persisted for this long, one would need a group of animals that underwent MPS and were tested for fear memory only once, at the 6-week time point. This limitation should at least be discussed.

Response: We recognize the reviewer's concern that the repeated retrieval may provide reminders for the MPS procedure and assist in long-term memory preservation. Therefore, we tested another MPS group (n=6) with the same stress-manipulated protocol on days 1-7 and housed in the home cage for six weeks without any disturbance, then detected single-time cue and contextual retrieval on days 44 and 46. The immobility (%)

result showed no differences between repeated and single retrieval examination groups, implying that the repeated cue and context exposure within six weeks did not affect extending fear memory preservation. Also, the results from other studies have suggested that the retrieval induced-memory labile is significant during recent, not remote consolidated memories (Alberini, 2011; Dudai, 2004; Milekic & Alberini, 2002). Thus, the memory retrieval process in our research consisted of six weeks and the possible impacts of fear retrieval on memory retention were limited. Yet, none of the extinction results have been displayed in the MPS group, which reflected an intense and prolonged stress effect of MPS. The added statistical method is displayed on page 10, lines 173-175 and the result is displayed on page 17, lines 242-256 of **Supplementary Figure 5** in the Supplementary Methods, Figures and Legends. The illustration is presented on page 23, lines 426-436 in the Discussion.

Additional references added are as follows:

Alberini, C. M. The Role of Reconsolidation and the Dynamic Process of Long-Term Memory Formation and Storage. *Front. Behav. Neurosci.* 5, (2011).

Dudai, Y. The Neurobiology of Consolidations, Or, How Stable is the Engram? *Annu. Rev. Psychol.* 55, 51–86 (2004).

Milekic, M. H. & Alberini, C. M. Temporally Graded Requirement for Protein Synthesis following Memory Reactivation. *Neuron* 36, 521–525 (2002).

9. For Figures 2 and 3, the correct statistical test is a repeated measures 2X2 ANOVA with factors of group and place (homecage vs retrieval box). These ANOVAs are not reported.

Response: We appreciate the reviewer's comments to include all the components of the within-group (home cage vs. context and pre-tone vs. post-tone) and between-group (control vs. MPS) comparison in Figures 2 and 3 using a two-way ANOVA. However, the basal power level between the control and MPS groups is too large to compare directly. Although we

have tried to identify all the surgical operations and eliminate the noise signal sources, we could not exclude the minor individual differences between rats that will affect the basal oscillation power. Therefore, we compared the within-group factors (home cage vs. context and pre-tone vs. post-tone) by t-test and plotted two groups in one graph to present the variation differences (within-group factors) between each group in Figure 2e and 2f of our first version of the manuscript. Nevertheless, with the consideration of the article's readability and the time factors (short- vs. long-term) during fear recall, we replotted the bar graphs to separate the control and MPS groups, and applied the statistical test of within-group repeat measure two-way ANOVA to include the treatment (home cage vs. context and pre-tone vs. post-tone) and time (short- vs. long-term) factors in Figure 2 and 3. We focused on the instant transformation after fear recall and marked the significant differences of the condition effect in the results, because the impact of time factor might affect the signal recording situation in each retrieval task. The recorded cell population during different time segments might be distinct from the original neurons and result in signal amplitude transformation. And the alteration trends were not consistent in each recording area, suggesting presented the significant alterations of the time-status effect might complex the main findings. Furthermore, in order to provide a direct comparison of neuronal oscillation power and coherence between control and MPS groups, we analyzed the variations by subtracting the pre-treated from the post-treated data in each group and used two-way ANOVA to analyze the differences from the between-group (control vs. MPS) and within-group (time sessions) analyses in Supplementary Figure 2.

The corrected statistical methods are shown on page 38, lines 710-713 in the Methods. The revised statistical descriptions of **Figure 2** are located on page 9, lines 160-166, page 10, lines 177-183 in the Results, and on page 59, lines 1037-1062 in the Figure legends.

And the corresponding descriptions of **Figure 3** are presented on page 12, lines 208-210 and lines 211-218, and page 13, lines 223-225 in the Results, and on page 61, lines 1064-1084 in the Figure legends.

The complementary contents of **Supplementary Figure 2** are displayed on page 9, lines 160-166, page 10, lines 183-188, page 12, lines 204-208 and lines 218-223 in the Results, and on page 10, lines 170-173 and page 13, lines 198-222 in the Supplementary Methods and Figures and Legends.

Moreover, the limitation of individual bias of baseline oscillation power between groups are complemented on page 31, lines 565-579 in the Discussion.

10. Line 137 “We next evaluated whether the neuronal activities during fear retrieval were correlated with the startle responses.” I do not see any data examining correlations between measures of oscillatory activity and fear retrieval (freezing).

Response: We are sorry for using the wrong explanation about the illustration of neuronal activities recording in the result. We have changed the statement of recording purpose as follows: “We next evaluated whether fear memory retrieval (context and cue retrieval) post-MPS manipulation would excite neuronal activity in fear encoding regions.” on page 9, lines 151-153 in the Results.

11. Line 255 “Stress-induced anxiety commonly occurs when fear memory flashbacks,” Please provide citations for this.

Response: We thank reviewer’s kindly reminder for adding the reference. We have added a corresponded citation and restructure the sentence as follows: “Stress-induced flashbacks provoked AMY, PFC, and HPC activation, which share the same hyperresponsivity regions that cause abnormal behaviors as anxiety⁵¹⁻⁵³.” on page 15, lines 269-271 in Result. Additional references added are as follows:

Bourne, C., Mackay, C. E. & Holmes, E. A. The neural basis of flashback formation: the impact of viewing trauma. *Psychol. Med.* 43, 1521–1532

(2013).

McGaugh, J. L. The Amygdala Modulates the Consolidation of Memories of Emotionally Arousing Experiences. *Annu. Rev. Neurosci.* 27, 1–28

(2004).

Sharp, B. M. Basolateral amygdala and stress-induced hyperexcitability affect motivated behaviors and addiction. *Transl. Psychiatry* 7, e1194–e1194 (2017).

12. Line 290 “the neuronal activities in the correlated fear-associated regions could be observed when fear memory was retrieved.” This statement is misleading. The regions aren’t “correlated” (wrong word), and the neuronal activity measured during the anxiety tests were not “observed when fear memory was retrieved” (it was measured AFTER fear memory retrieval).

Response: We thank reviewer for the correction our misleading statements in the result. We have revised the sentence as follows: “Together, the MPS protocol provoked extended anxiety, and raised neuronal activity in fear-associated regions (PFC, BLA, and vHPC) during behavior tasks, signifying rats exhibited hyper-responses after fear memory retrieval during the remote time segment.” on page 17, lines 302-305 in the Results.

13. In several places, it is stated that MPS effects are “more significant” than SPS effects. This is not an appropriate statement. There is no such thing as more/less significant. You can compare the size of an effect, by testing whether there is a group (MPS vs SPS) effect in a measure, but you do not compare the p-values of independent tests across the two groups. This should be corrected in multiple places.

Response: We apologize for using nonspecific statements in our manuscript. We have corrected and restructured all the imprecise comparison descriptions and added the statistical numbers in the required statements to support and clarify the main results.

The revised sentences are presented on page 11, lines 183-186 and page 12, lines 208-210.

14. The memory retrieval aspect of the MPS model is only retrieving one aspect of the stressful experience. The authors cannot know that this induces a memory retrieval of other aspects of the stressors (for example, the isoflurane exposures). The authors seem to imply that the “memory” being retrieved is that of the stressful experience in its totality.

Response: We are sorry for not explaining the implication of fear memory retrieval in recalling the IFS-induced-learned fear. Since stressful experiences in PTSD requires fear learning to pair the conditioned stimuli to trauma, the context and cue tone in the MPS process were only paired with the IFS stimulation; thus, the memory retrieval only recalled the IFS-induced stress. However, the multiple stressors that applied prior to IFS may increase the sensitivity of rats to the central IFS stimulation and induce longitudinal PTSD-like effects.

The complementary explanation is described on page 24, lines 438-445 in the Discussion.

15. I think that the finding that the MPS produces stronger effects than the SPS on multiple measures is a key finding of this paper. I think that the SPS data should be included in Figure 1 and discussed up front in the results.

Response: We thank reviewer’s suggestion. We have transferred the immobility behavior of the SPS from the supplementary data to Figure 1 to provide crucial evidence of that MPS produces stronger effects on remote fear memory retrieval when comparing with SPS and control groups in context recall. The comparison results of behavioral tasks and sleep-wake activity between SPS, MPS, and control groups were also discussed in the Discussion section to emphasize the major effects (theta oscillation activities, anxiety behavior, and sleep disruptions) of MPS in the main text.

The methods of SPS were described on page 35, lines 640-651 and lines 654-655 in the Methods. The revised results were discussed on page 7, lines 116-143 in the Results, and on page 56, lines 1002-1033 in **Figure 1** of Figures and Figure Legends. The remained behavioral tasks (page 27, lines

496-499) and sleep-wake activity (page 28, lines 508-511 and lines 521-527 and page 29, lines 530-537, page 30, lines 543-547) were discussed in the Discussion.

Reviewer: 2

Comments to the Corresponding Author:

I am convinced that the paper will be of substantial interest to the community. Due to the lack of reliable stress protocol for modeling any stress-related disorders, including PTSD or anxiety disorders, the community demanded the another's stress model. Through this revision process, the authors nicely improved their article by adding more data and more elaborate description to strengthen their argument.

- 1) The authors corrected their missing points.
- 2) The authors added the description in Introduction and Discussion and added another set of data which compare MPS model with SPS model to strengthen the necessity of MPS protocol instead of SPS protocol for modeling PTSD.
- 3) Also, calculating the difference between before-after (Supplementary Figure 2) was a nice idea to interpret the data in a simple way.

But, still, as previous comments, I am worried about the statistical analysis and suggest to reconsider. The selective choice of statistical analysis seems that the authors analyzed the data in a biased way. The authors should be stricter about the statistical analysis according to the rules.

Response: We appreciate reviewer's suggestions. We understood your concern about the statistical analysis method applied in the main results. Therefore, we have made corrections according to the bellowed commons in each questionable figure. The detailed responses to individual questions or concerns are explained as follows.

1. Figure 1c-e: Here, the authors used unpaired-t-test just to analyze the group difference in various states. Since the same mice were used in different states, the authors should use Two-way ANOVA with posthoc test or at least, paired t-test.

Response: We thank the reviewer for reminding us of the time factors (different states) of the immobility behavior in each comparison group. We

have changed the statistical method into two-way ANOVA with Bonferroni pairwise post hoc comparison to include the factors of time processing (within-group) that caused behavior alteration and different manipulations (between-group) induced memory preservation diversity. However, due to the data loss in the memory retrieval process, we excluded the inappropriate behavior results (e.g., sleep during the cued recall) during fear recall tasks. Thus, the collected sample number was inconsistent, which the repeated measures of two-way ANOVA were unable to apply in Figure 1c-e. Also, according to reviewer #1's suggestion on providing evidence of superior longitudinal effects of MPS over SPS, we replaced the original Figure 1c-e with the comparison of immobility results between control, SPS, and MPS groups during the MPS and during fear memory retrieval period.

The statistical methods have been revised on page 38, lines 707-710 in the Methods. The added descriptions of SPS methods were located on page 35, lines 640-651 and lines 654-655 in the Methods. The corresponding results were displayed on page 7, lines 116-143 in the Results. The replaced figure and legends were presented on page 56, lines 1002-1033 in the Figures and Legends.

Furthermore, considering state alteration in behavioral tasks (EPM and OFT), we also changed the statistical analysis method in Figure 5c and 5e and Supplementary Figure 6b and 6c. Two-way ANOVA with Bonferroni pairwise post hoc comparison was used to analyze the within-group difference of short-term and long-term variations after either context or cue recall, respectively, and the between-group difference in different manipulations.

The statistical methods have been revised on page 38, lines 707-710 in the Methods, and on page 10, lines 173-175 in the Supplementary Methods. The corresponding results were displayed on page 15, lines 275-278 and page 16, lines 292-293 in the Results, and on page 27, lines 496-502 in the Discussion. The replaced figures and legends were presented on page 64, lines 1100-1127 (Figure 5c and 5e) in the Figures and

Legends, and on page 18, lines 258-277 (**Supplementary Figure 6b and 6c**) in the Supplementary Figures and Legends.

2. Figure 2d-f: The authors compared two groups by One-way ANOVA with Tukey posthoc instead of t-test. This was unnecessary and no point of changing statistical method. In theory, this dataset should be compared to see the group effect and condition effect (home cage vs context), simultaneously. For this purpose, the authors should use Repeated Measures of ANOVA with posthoc test, if the same mice were used in homecage and context.

Response: We apologize for using the biased statistical methods in analyzing the LFP power intensity in Figure 2. To clarify the main comparison factors of pre- and post-retrieval alterations of theta power, we separated the original merged-group liner plot into individual bar graphs of control and MPS groups in **Figures 2d and 2f**. Moreover, repeated measures of two-way ANOVA with Bonferroni post hoc comparison were used to measure the pre- and post-condition effect (home cage vs. context or pre- vs. post-cue) and time status effect (short- vs. long-term).

Nonetheless, the effect of time processing-caused variant status might alter the signal recording situation in each retrieval task. The recorded cell population during different time segments might be distinct from the original neurons and result in signal amplitude transformation. And the alteration trends were not consistent in each recording area, suggesting presented the significant alterations of the time-status effect might complex the main findings. Therefore, we focused on the instant transformation after fear recall, and marked the significant differences of the condition effect in the figure.

The statistical methods have been revised on page 38, lines 710-713 in the Methods. The corresponding results were displayed on page 9, lines 160-166, page 10, lines 177-183 in the Results. The replaced figure and legends were presented on page 59, lines 1037-1062 in the Figures and Legends.

By using the same method, we also changed the statistical analysis in **Figure 3b and 3c** by repeated measures of two-way ANOVA with Bonferroni post hoc comparison to analyze the pre- and post-condition effect and time status effect.

The statistical methods have been revised on page 38, lines 710-713 in the Methods. The corresponding results were displayed on page 12, lines 208-210 and lines 211-218, page 13, lines 223-225 in the Results. The replaced figure and legends were presented on page 61, lines 1064-1084 in the Figures and Legends.

We also replaced the statistical method in the **Supplementary Figure 2** by repeated measures of two-way ANOVA with Bonferroni post hoc comparison to include the factors from different manipulations (control vs. MPS; between-group) and time status (within-group) effects.

The statistical methods have been revised on page 10, lines 170-173 in the Supplementary Methods. The corresponding results were displayed on page 9, lines 160-166, page 10, lines 183-188, page 12, lines 204-208 and lines 218-223 in the Results. The replaced figure and legends were presented on page 13, lines 198-222 in the Supplementary Figures and Legends.

3. It was also surprising to see the huge group difference of theta power in home cage (Figure 2d) and pre-cue tone (Figure 2e), which may indicate the basal status. It seems that the theta power of MPS rats are higher in basal condition, which was not mentioned by authors. The authors should consider about discussing this difference.

Response: We understood the reviewer's concern about the significant variation of basal power intensity between groups. We have added an explanation of the possible reason in the Discussion section that "Furthermore, the considerable control and MPS group basal neuronal oscillation power variations in the PFC, BLA, and vHPC regions may be due to the surgical deviation of LFP electrode stability, begetting signal discrepancies."

The added descriptions were presented on page 31, lines 565-568 in the Discussion.

Reviewers' comments:

Reviewer #1 (Remarks to the Author):

The authors have done a lot to improve the paper. Comparing SPS and MPS is important, and some of the methods are much more clear. The biggest remaining problem seems to be the use of incorrect statistics. In their response, the authors indicate that t-tests and similar were used because of missing data. However, the figure legends and methods don't indicate exclusions that would support missing data, and the information in the ANOVAs included in the current figure legends also appear to have the same number of subjects across different time points. I think part of our (the reviewer's) concerns about this arise because it appears there is no missing data, so we don't understand why repeated measures ANOVAs are not being used to make across-group comparisons. If there is missing data, then there are better ways to deal with that here, but my recommendations would depend on how much data is missing (and I can't tell that because the authors don't report this). Please see below.

It is concerning that despite comments from both reviewers about the statistics used, the statistics reported are still not the correct ones. Perhaps the authors should consult with a statistician. Figure 1 reports two-way ANOVAs for each pair of time points in panels C,D, and E. This is largely not correct. For example, to support the claim that MPS "immobility (%) continuously increased during the IFS procedure", one needs a one-way ANOVA with a significant effect of time for the MPS group, or a significant time X group interaction in a two-way repeated measures ANOVA examining time and group (MPS vs control). In Figure 1D, to show significant extinction in contextual fear, one needs a significant main effect of time in the repeated measures ANOVA (group X time), and to show differences in extinction rate across groups, a significant group X time effect. Same for Figure 1E. In the response, the authors state, "due to the data loss in the memory retrieval process, we excluded the inappropriate behavior results (e.g., sleep during the cued recall)". However, methods don't describe any criteria for exclusions nor is it indicated how many rats were excluded at each data point. The figure legend seems to indicate that the MPS group always has 18 and the control and SPS 12 each. In addition, the reported degrees of freedom are the same for days 3,5, and 7, which suggests that the same number of data points are present at the three time points. If there are truly different numbers of rats represented on days 3,5, and 7, then the Ns clearly need to be indicated, the criteria for exclusion need to be described clearly in the methods (how would the authors know the animal was sleeping versus merely freezing?), and the degrees of freedom reported should be checked because they are most likely wrong. If there is missing data, then there are specific statistical models that deal with this, and those should be implemented here.

In Figures 2D and F, two-way repeated measures ANOVA are correctly reported, but the authors don't indicate what effect is being reported (main effect? Interaction?) Also, as the other reviewer noted, the most important comparison is the across-group comparison. The fact that the basal oscillations differ across groups is fine (one would report a main effect of group). But, one must, for each brain area, run a repeated measures ANOVA with factors of group, testing environment (home cage vs context) and time. If the groups differ in terms of how oscillatory activity is changed depending on context, there should be a significant group X environment X time interaction (or perhaps a group X environment interaction, depending on how you want to make the comparison).

The authors say that the basal differences between MPS and control groups can be accounted for by “surgical deviation of LFP electrode stability”. I don’t know what this means. It seems far more logical to me that the difference is because MPS suppresses oscillatory activity. Why do the authors not consider this the most straightforward explanation?

p.11 “We found that MPS protocol strengthened PFC-BLA coherence during context retrieval, but only increased it in the control group during long-term retrieval” Aren’t all the time points shown “long-term retrieval”? This statement does not accurately describe the findings.

Figure 4 has similar statistical concerns as described above. To make group comparisons, t-tests are reported. This is not appropriate. ANOVA should be used. In all Figures, including this one, the Ns for each group need to be clearly reported and any “missing data” needs to be clearly indicated, with the reasons well-described in the methods.

Reviewer #2 (Remarks to the Author):

The authors tried to implement my comments, especially about the statistical analysis. I have no other comments.

We thank the editor and reviewers for reviewing this manuscript and providing constructive suggestions. According to the reviewer's comments, this manuscript has been revised. The revised contents are written in red font with yellow highlight. The responses to each comment are answered point-by-point as follows.

Reviewer comments:

Reviewer: 1

Comments to the Corresponding Author:

The authors have done a lot to improve the paper. Comparing SPS and MPS is important, and some of the methods are much more clear. The biggest remaining problem seems to be the use of incorrect statistics. In their response, the authors indicate that t-tests and similar were used because of missing data. However, the figure legends and methods don't indicate exclusions that would support missing data, and the information in the ANOVAs included in the current figure legends also appear to have the same number of subjects across different time points. I think part of our (the reviewer's) concerns about this arise because it appears there is no missing data, so we don't understand why repeated measures ANOVAs are not being used to make across-group comparisons. If there is missing data, then there are better ways to deal with that here, but my recommendations would depend on how much data is missing (and I can't tell that because the authors don't report this). Please see below.

Response: We thank the reviewer's comments for pointing out the statistic problems in this manuscript. We are sorry for our loss of not reporting clear information about the missing data in the experiment and using the wrong statistical methods to analyze our results. Thus, we have corrected the analyzed methods that were aroused by the reviewer and provided clear statistical values in the corresponding figure legends. The detailed responses to individual questions or concerns are explained as follows.

1. It is concerning that despite comments from both reviewers about the statistics used, the statistics reported are still not the correct ones. Perhaps the authors should consult with a statistician. Figure 1 reports two-way ANOVAs for each pair of time points in panels C,D, and E. This is largely not correct. For example, to support the claim that MPS “immobility (%) continuously increased during the IFS procedure”, one needs a one-way ANOVA with a significant effect of time for the MPS group, or a significant time X group interaction in a two-way repeated measures ANOVA examining time and group (MPS vs control). In Figure 1D, to show significant extinction in contextual fear, one needs a significant main effect of time in the repeated measures ANOVA (group X time), and to show differences in extinction rate across groups, a significant group X time effect. Same for Figure 1E. In the response, the authors state, “due to the data loss in the memory retrieval process, we excluded the inappropriate behavior results (e.g., sleep during the cued recall)”. However, methods don’t describe any criteria for exclusions nor is it indicated how many rats were excluded at each data point. The figure legend seems to indicate that the MPS group always has 18 and the control and SPS 12 each. In addition, the reported degrees of freedom are the same for days 3,5, and 7, which suggests that the same number of data points are present at the three time points. If there are truly different numbers of rats represented on days 3,5, and 7, then the Ns clearly need to be indicated, the criteria for exclusion need to be described clearly in the methods (how would the authors know the animal was sleeping versus merely freezing?), and the degrees of freedom reported should be checked because they are most likely wrong. If there is missing data, then there are specific statistical models that deal with this, and those should be implemented here.

Response: We apologize for using the incorrect method in analyzing immobility (%); we have re-analyzed the data and added the corresponding statistical results, including the sample size (n) on each day and interaction of each comparison, in the figure legends of **Figure 1** and **Supplementary Figure 5**. We also added the complementary details

of exclusion criteria during immobility (%) data collection (sleep during the cued recall) that sleep behavior refers to rats presented with eyes closed and motionless for over 12 seconds (equal to the epoch length in our sleep-wake activity scoring) during fear memory retrieval.

In **Figure 1c**, due to the SPS protocol only having single-time IFS process manipulation that cannot be directly compared with the four-time IFS protocol in the MPS group; therefore, we separated the fear acquisition immobility (%) in the SPS group and used one-way ANOVA to compare the freezing behavior between control, SPS, and MPS-Day 1 group (right upper panel).

In **Figure 1d and 1e**, as described in the Supplementary Methods, some of the immobility (%) data on each day were excluded because of the sleep behavior. Furthermore, we also found that some of the video results from the control group were lost due to machine malfunction; hence, we could not apply repeat-measure ANOVA in our data analysis. However, to include the group (between group) x time (within group) factors together in our result comparison, we used generalized estimating equations (GEE) with exchangeable correlation structure and Bonferroni pairwise comparison model to process our unequal data collections on each day. And we also presented the sample size (n) on each day and the interaction of each comparison in the figure legends.

The same analyzing model (GEE) was applied and reported in **Supplementary Figure 5** to compare the group x time difference between the single and repeated memory retrieval group.

The added data exclusion illustration is described on page 5, lines 77-79 in the Supplementary Methods.

The revised analysis models were presented on page 40, lines 746-752 in the Methods and page 10, lines 174-177 in the Supplementary Methods.

The changed figures and legends are displayed on page 58, line 1031 and page 59, lines 1042-1066 of **Figure 1** in the Figures and Legends and page 19, lines 263-267 of **Supplementary Figure 5** in the Supplementary Figures and Legends.

2. In Figures 2D and F, two-way repeated measures ANOVA are correctly reported, but the authors don't indicate what effect is being reported (main effect? Interaction?) Also, as the other reviewer noted, the most important comparison is the across-group comparison. The fact that the basal oscillations differ across groups is fine (one would report a main effect of group). But, one must, for each brain area, run a repeated measures ANOVA with factors of group, testing environment (home cage vs context) and time. If the groups differ in terms of how oscillatory activity is changed depending on context, there should be a significant group X environment X time interaction (or perhaps a group X environment interaction, depending on how you want to make the comparison).

Response: We are sorry for not concluding the across-group comparison and reporting the interaction and main effect significant values in **Figure 2** and **Figure 3**. As the reviewer mentioned above, the power and coherence alteration's primary effect must include the group and environment (home cage vs. context)/period (1 second pre- and post-cue onset) factors. Nevertheless, we focus on comparing the memory retrieval effect in each detected day, and our immobility (%) data also indicated that repeated memory recall has no difference in contrast to the single-time recall group (Supplementary Figure 5), which multiple examinations may not be the main effect dominate power alteration. Thus, we did not include the time differences interaction; instead, we used group x environment/period two-way repeated measure ANOVA for analysis. Furthermore, we added the corresponding statistical report of sample size (n) in each group and interaction and main effects (group and environment/period) value in the figure legends to present our results.

The changed statistical methods are shown on page 41, lines 752-755 in the Methods.

The revised result descriptions are located at page 9, lines 164-178 and page 11, lines 190-200 (**Figure 2**) and at page 12, lines 208-213 and lines 217-224, page 13, lines 229-241 (**Figure 3**) in the Results.

The replaced figures and legends are displayed on page 61, line 1071 and lines 1078-1094, and page 63, lines 1098-1117 (**Figure 2**) and page 64, line 1120 and lines 1124-1158 (**Figure 3**) in the Figures and Legends.

3. The authors say that the basal differences between MPS and control groups can be accounted for by “surgical deviation of LFP electrode stability”. I don’t know what this means. It seems far more logical to me that the difference is because MPS suppresses oscillatory activity. Why do the authors not consider this the most straightforward explanation?

Response: We thank the reviewer for providing an explanation to illustrate the basal oscillation power differences between the control and MPS groups. We used “surgical deviation of LFP electrode stability” as the possible rationale because we observed an overall high basal power in the control group when we separated and compared the power intensity in the control and MPS group in our last version of the manuscript. Therefore, we concluded that the variation might result from the surgical deviation caused by the slight difference when we coordinated the brain regions or implanted the electrodes in each rat. However, when we applied group x environment/ period repeated comparison, we found that the basal power variation between groups was not consistently presented in each area and period, which indicates less correlation with the surgical deviation caused the difference. Thus, we revised the explanation into “Furthermore, the considerable basal neuronal oscillation power variations in the PFC, BLA, and vHPC regions between control and MPS group were discovered in short-term memory retrieval (Days 21 and 19; Fig. 2d, f), and was specifically stronger at PFC region throughout the long-term period (Fig. 2d, f). One possible

reason might be that the stress, caused by experiment manipulations (e.g., changing rats from animal housing room to insulation LFP recording space and electrodes implantation), increases the control rat's oscillation power during short-term memory retrievals. Another possibility is that PFC basal oscillation variation may result from the nuclei's functional specificity. The PFC activation is crucial for limbic system connection and processes the safe signal 91. Our results exhibited that control rats enhanced their theta power intensity, specifically in the medial prefrontal cortex (mPFC) region, and its coherency from mPFC to BLA and vHPC transiting the safety direction, which may refer to the continuous dominance of PFC when environment shifting. Moreover, other research has also indicated that a population of PFC neurons is less responsive when repeated exposure to similar stimulation that caused a slow adaptive effect 92, which multiple times of context or cue memory recall may reduce the reaction of PFC in the MPS rats and resulted in signal discrepancies contrast to the control group."

The altered explanation is presented on page 32, lines 588-607 in the Discussion.

Additional reference added is as follows:

Lesting, J. et al. Directional Theta Coherence in Prefrontal Cortical to Amygdalo-Hippocampal Pathways Signals Fear Extinction. PLoS ONE 8, e77707 (2013).

Jackson, M. E. & Moghaddam, B. Distinct patterns of plasticity in prefrontal cortex neurons that encode slow and fast responses to stress. Eur. J. Neurosci. 24, 1702–1710 (2006)

4. p.11 "We found that MPS protocol strengthened PFC-BLA coherence during context retrieval, but only increased it in the control group during long-term retrieval" Aren't all the time points shown "long-term retrieval"? This statement does not accurately describe the findings.

Response: We are sorry for our misleading description in the Result section. The time points in our experimental design displayed Days 10-21 as short-term and Days 35-46 as long-term memory retrieval periods,

which the illustration is located at page 38, lines 709-718 in the Methods. However, the results of coherence intensity significance between brain regions were changed after we altered the statistical method to group x environment/period two-way repeated measure ANOVA; therefore, the original statement was not correlated with the new statistical result that both control and MPS groups were strengthened in long-term PFC-BLA coherence.

The changed result depiction is shown on page 12, lines 208-210 in the Results

5. Figure 4 has similar statistical concerns as described above. To make group comparisons, t-tests are reported. This is not appropriate. ANOVA should be used. In all Figures, including this one, the Ns for each group need to be clearly reported and any “missing data” needs to be clearly indicated, with the reasons well-described in the methods.

Response: We appreciate the reviewer’s reminder on our inappropriate statistical methods in **Figure 4**; we have revised the method from the t-test into a two-way repeated measure ANOVA that included environment/period x direction factors.

Furthermore, considering the repeated examination factor in anxiety behavior tasks (EPM and OFT) in **Figure 5c and 5e** and **Supplementary Figure 6b and 6c**, we re-analyzed the results by group x time two-way repeated measure ANOVA.

To clearly reported our analyzed data, we complement the sample size (n) of each group in all the figure legends.

The corrected statistical methods are presented on page 41, lines 752-758 in the Methods and page 10, lines 177-179 in the Supplementary Methods.

The modified corresponding result statements are located at page 14, lines 256-275, page 15 and lines 278-288 (Figure 4) in the Result. Page 27, lines 502-508 (Figure 4); Page 28, lines 519-525 (Supplementary Figure 6) in the Discussion.

The replaced figures and legends are displayed on page 67, line 1161 and lines 1164-1196 (**Figure 4**); page 70, line 1199, page 71, lines 1208-1218 and lines 1221-1234 (Figure 5). Page 19, line 270 and lines 274-290 (Supplementary Figure 6) in the Supplementary information.

Reviewer #1 (Remarks to the Author):

I appreciate that the authors have added more information about the data exclusions. I am clear with the GEE modeling of Figure 1. I also think Figure 5 stats are fine. However, for the other figures (2- 4), the authors have reported many interactions in the figure legends, but it still isn't always clear what is being reported. For example, let's look at Figure 2. It says "Two-way repeated measures ANOVA: environment x group interaction FDay 21(1,10)=4.802, p=0.053" This is clear. The environment is a two level repeated measures factor (home cage, context) and the group is a two level factor (MPS or control). The "FDay21" is a strange way of reporting this, but I understand this means the ANOVA was run just on the Day 21 data.

But then, in the long list of F-statistics after "environment X group interaction", after the 3 days of environment X group F-statistics are reported for each of the 3 days, the next text reads "Fenvironment Day 21(1,10)=1.101, p=0.319". I don't know what this is. What does the "environment Day" after the "F" signify? Whatever that statistic is, it is reported for Day 21 and Day 46, but not Day 35. Why? Is this supposed to be the main effect of environment on Days 21 and 46? If so, I have never seen this reported in this way. I have the same question for the "F group Day" statistics. What does "group Day" mean? And why is this reported for Day 35 and 46, but not Day 21? Whatever the authors are trying to report here, it is largely unclear and it is not a conventional way of reporting these statistics. I suggest that a table embedded within the figure (or provided as a supplement) would be clearer. The first column would read "main effect of group" "main effect of environment" and "group X environment interaction" from top to bottom. The top row would read Day 21, Day 35 and Day 46 across 3 columns. For Figure 2, there would be 3 tables, one each for PFC, BLA, and vHPC. The authors could then report the F-statistic and p-value for each square in the resulting table. They could make the shading in all square with significant p-values a grayscale value to further draw attention to the significant effects. There would be parallel tables for the other figures, but with the relevant factors in the two-way ANOVAs (it isn't always group and environment).

We thank the editor and reviewers for reviewing this manuscript and providing constructive suggestions. According to the reviewer's comments, this manuscript has been revised. The revised contents are written in red font with yellow highlight. The responses to each comment are answered point-by-point as follows.

Reviewer comments:

Reviewer: 1

Comments to the Corresponding Author:

I appreciate that the authors have added more information about the data exclusions. I am clear with the GEE modeling of Figure 1. I also think Figure 5 stats are fine. However, for the other figures (2- 4), the authors have reported many interactions in the figure legends, but it still isn't always clear what is being reported. For example, let's look at Figure 2. It says "Two-way repeated measures ANOVA: environment x group interaction $F_{Day\ 21(1,10)}=4.802$, $p=0.053$ " This is clear. The environment is a two level repeated measures factor (home cage, context) and the group is a two level factor (MPS or control). The "FDay21" is a strange way of reporting this, but I understand this means the ANOVA was run just on the Day 21 data.

But then, in the long list of F-statistics after "environment X group interaction", after the 3 days of environment X group F-statistics are reported for each of the 3 days, the next text reads "Fenvironment Day 21(1,10)=1.101, $p=0.319$ ". I don't know what this is. What does the "environment Day" after the "F" signify? Whatever that statistic is, it is reported for Day 21 and Day 46, but not Day 35. Why? Is this supposed to be the main effect of environment on Days 21 and 46? If so, I have never seen this reported in this way. I have the same question for the "F group Day" statistics. What does "group Day" mean? And why is this reported for Day 35 and 46, but not Day 21? Whatever the authors are trying to report here, it is largely unclear and it is not a conventional way of reporting these statistics. I suggest that a table embedded within the figure (or provided as a supplement) would be clearer. The first column would read "main effect of group" "main effect of environment" and "group X environment interaction" from top to bottom. The top row would read Day 21, Day 35 and

Day 46 across 3 columns. For Figure 2, there would be 3 tables, one each for PFC, BLA, and vHPC. The authors could then report the F-statistic and p-value for each square in the resulting table. They could make the shading in all square with significant p-values a grayscale value to further draw attention to the significant effects. There would be parallel tables for the other figures, but with the relevant factors in the two-way ANOVAs (it isn't always group and environment).

Response: We thank the reviewer's comments for pointing out the statistic reporting problems and providing construct suggestions for this manuscript. We understand the reviewer's concern that the statistical description of two-way repeated measures ANOVA is unclear; however, to display all the interactions, between, and within factor statistics of the significant data, we used three types of subscripts in our figure legends. The explanation of each subscript type is presented below.

In the two-way repeated measures ANOVA statistical value presentation, we first reported the **two-factor interaction values** in each analyzing condition, then we reported the **main effect's** statistical values ("environment/period" and "group" effects) that have post-hoc analyzed significant differences.

Take Figure 2d PFC theta power comparison as an example:

- The **interaction** *F*-statistics and *p*-values in each experimental day was displayed.

Environment x group interaction: on Day 21 $F_{(1,10)}=4.802$, $p=0.053$, on Day 35 $F_{(1,10)}=0.129$, $p=0.727$, on Day 46 $F_{(1,10)}=1.527$, $p=0.245$.

- Main effects of **environment**:

Only Days 21 and 46 presented theta power differences when changing rats from home cage to context in the MPS group.

On Day 21 $F_{\text{environment}(1,10)}=1.101$, $p=0.319$, on Day 46 $F_{\text{environment}(1,10)}=7.804$, $p<0.05$.

- Main effects of **group**:

Only Days 35 and 46 presented theta power differences in either the home cage or context when compared control with MPS rats.

On Day 35 $F_{\text{group}(1,10)}=11.223$, $p<0.05$, on Day 46 $F_{\text{group}(1,10)}=13.795$, $p<0.01$.

The reported method is referred to “Oesch, L. T. et al. REM sleep stabilizes hypothalamic representation of feeding behavior. Proc. Natl. Acad. Sci. 117, 19590–19598 (2020)”.

Nonetheless, owing to the defects of not providing complete statistical results of each main effect and the unclear presented values that may confuse the readers, we provide six extra supplementary tables that concluded the statistical values of “interaction” and “main effects” of all the two-way repeated measures ANOVA analysis results to clearly exhibit our data.

The corrected figure legends are displayed on page 63, lines 1103-1148 of **Figure 2**; page 66, lines 1157-1197 of **Figure 3**; page 70, lines 1206-1242 of **Figure 4**; page 73, lines 1255-1285 of **Figure 5** in the Figures and Legends. And page 14, lines 207-236 of **Supplementary Figure 2**; page 20, lines 282-296 of **Supplementary Figure 6** in the Supplementary Figures and Legends.

The added statistical tables are shown on page 26, lines 357-361 of **Supplementary Table 1** (corresponding with Figure 2); on page 27, lines 362-366 of **Supplementary Table 2** (corresponding with Figure 3); on page 28, lines 368-372 of **Supplementary Table 3** (corresponding with Figure 4); on page 29, lines 374-378 of **Supplementary Table 4** (corresponding with Figure 5); on page 30, lines 380-384 of **Supplementary Table 5** (corresponding with Supplementary Figure 2); on page 31, lines 386-390 of **Supplementary Table 6** (corresponding with Supplementary Figure 6) in the Supplementary Tables and Legends.